# Distributed Least Squares in Small Space via Sketching and Bias Reduction

**Sachin Garg**
Computer Science & Engineering
University of Michigan
sachg@umich.edu

**Kevin Tan**
Department of Statistics
University of Pennsylvania
kevtan@umich.edu

**Michał Dereziński**
Computer Science & Engineering
University of Michigan
derezin@umich.edu

## Abstract

Matrix sketching is a powerful tool for reducing the size of large data matrices. Yet there are fundamental limitations to this size reduction when we want to recover an accurate estimator for a task such as least square regression. We show that these limitations can be circumvented in the distributed setting by designing sketching methods that minimize the bias of the estimator, rather than its error. In particular, we give a sparse sketching method running in optimal space and current matrix multiplication time, which recovers a nearly-unbiased least squares estimator using two passes over the data. This leads to new communication-efficient distributed averaging algorithms for least squares and related tasks, which directly improve on several prior approaches. Our key novelty is a new bias analysis for sketched least squares, giving a sharp characterization of its dependence on the sketch sparsity. The techniques include new higher-moment restricted Bai-Silverstein inequalities, which are of independent interest to the non-asymptotic analysis of deterministic equivalents for random matrices that arise from sketching.

## 1 Introduction

Matrix sketching is a powerful collection of randomized techniques for compressing large data matrices, developed over a long line of works as part of Randomized Numerical Linear Algebra [RandNLA, e.g., 45, 26, 37, 39, 21]. Sketching can be used to reduce the large dimension $n$ of a data matrix $\mathbf{A} \in \mathbb{R}^{n \times d}$ by applying a random sketching matrix (operator) $\mathbf{S} \in \mathbb{R}^{m \times n}$ to obtain the sketch $\tilde{\mathbf{A}} = \mathbf{SA} \in \mathbb{R}^{m \times d}$ where $m \ll n$. For example, sketching can be used to approximate the solution to the least squares problem, $\mathbf{x}^* = \operatorname{argmin}_{\mathbf{x}} L(\mathbf{x})$ where $L(\mathbf{x}) = \|\mathbf{Ax} - \mathbf{b}\|^2$, by using a sketched estimator $\tilde{\mathbf{x}} = \operatorname{argmin}_{\mathbf{x}} \|\tilde{\mathbf{A}}\mathbf{x} - \tilde{\mathbf{b}}\|^2$, where $\tilde{\mathbf{A}} = \mathbf{SA}$ and $\tilde{\mathbf{b}} = \mathbf{Sb}$.

Perhaps the simplest form of sketching is subsampling, where the sketching operator $\mathbf{S}$ selects a random sample of the rows of matrix $\mathbf{A}$. However, the real advantage of sketching as a framework emerges as we consider more complex operators $\mathbf{S}$, such as sub-Gaussian matrices [1], randomized Hadamard transforms [3], and sparse random matrices [12]. These approaches ensure higher quality and more robust compression of the data matrix, e.g., leading to provable $\epsilon$-approximation guarantees for the estimate $\tilde{\mathbf{x}}$ in the least squares task, i.e., $L(\tilde{\mathbf{x}}) \leq (1 + \epsilon)L(\mathbf{x}^*)$. Nevertheless, there are fundamental limitations to how far we can compress a data matrix using sketching while ensuring an $\epsilon$-approximation. These limitations pose a challenge particularly in space-limited computing

38th Conference on Neural Information Processing Systems (NeurIPS 2024).

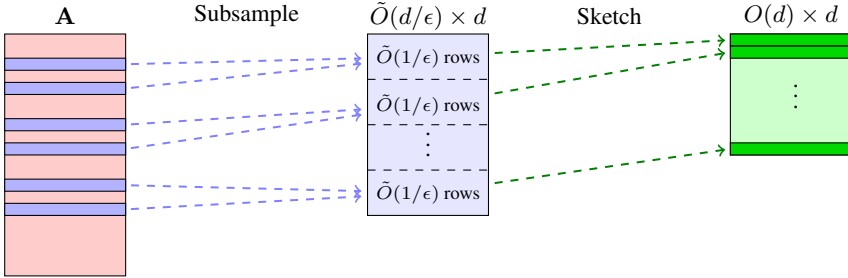

Figure 1: Illustration of the leverage score sparsification algorithm used in Theorem 1. Each row of the sketch mixes $\tilde{O}(1/\epsilon)$ leverage score samples from $\mathbf{A}$. Remarkably, the $\epsilon$-error guarantee of the subsampled estimator is retained as $\epsilon$-bias of the sketched estimator.

environments, such as for streaming algorithms where we observe the matrix $\mathbf{A}$, say, one row at a time, and we have limited space for storing the sketch [11].

One strategy for overcoming the fundamental limitations of sketching as a compression tool is to look beyond the single approximation guarantee provided by a sketching-based estimator $\tilde{\mathbf{x}}$, and consider how its broader statistical properties can be leveraged in a given computing environment. To that end, many recent works have demonstrated both theoretically and empirically that sketching-based estimators often exhibit not only approximation robustness but also statistical robustness, for instance enjoying sharp confidence intervals, effectiveness of statistical inference tools such as bootstrap and cross-validation, as well as accuracy boosting techniques such as distributed averaging [e.g., 35, 25, 32, 33]. Yet, these results have had limited impact on the traditional computational complexity analysis in RandNLA and sketching literature, as many of them either impose additional assumptions, focus on sharpening the constant factors, or require using more expensive sketching techniques. In this work, we demonstrate that the statistical properties of sketching-based estimators can in fact have a substantial impact on the computational trade-offs that arise in RandNLA.

Our key motivating example is the above mentioned least squares regression task. It is well understood that for an $n \times d$ least squares task, to recover an $\epsilon$-approximate solution out of an $m \times d$ sketch, we need sketch size at least $m = \Omega(d/\epsilon)$. This has been formalized in the streaming setting with a lower bound of $\Omega(\epsilon^{-1}d^2 \log(nd))$ bits of space required, when all of the input numbers use $O(\log(nd))$ bits of precision [11]. One setting where this can be circumvented is in the distributed computing model where the bits can be spread out across many machines, so that the per-machine space can be smaller. Here, one could for instance hope that we can maintain small $O(d) \times d$ sketches in $q = O(1/\epsilon)$ machines and then combine their estimates to recover an $\epsilon$-approximate solution. A simple and attractive approach is to average the estimates $\tilde{\mathbf{x}}_i$ produced by the individual machines, returning $\hat{\mathbf{x}} = \frac{1}{q} \sum_{i=1}^{q} \tilde{\mathbf{x}}_i$, as this only requires each machine to communicate $O(d \log(nd))$ bits of information about its sketch. This approach requires the sketching-based estimates $\tilde{\mathbf{x}}_i$ to have sufficiently small bias for the averaging scheme to be effective. While this has been demonstrated empirically in many cases, existing theoretical results still require relatively expensive sketching methods to recover low-bias estimators, leading to an unfortunate trade-off in the distributed averaging scheme between the time and space complexity required.

In this work, we address the time-space trade-off in distributed averaging of sketching-based estimators, by giving a sharp characterization of how their bias depends on the sparsity of the sketching matrix. Remarkably, we show that in the distributed streaming environment one can compress the data down to the minimum size of $O(d^2 \log(nd))$ bits at no extra computational cost, while still being able to recover an $\epsilon$-approximate solution for least squares and related problems. Importantly, our results require the sketching matrix to be slightly denser than is necessary for obtaining approximation guarantees on a single estimate, and thus, cannot be recovered by standard RandNLA sampling methods such as approximate leverage score sampling [27].

Before we state the full result in the distributed setting, we give our main technical contribution. which is the following efficient construction of a low-bias least squares estimator in a single pass using only $O(d^2 \log(nd))$ bits of space, assuming all numbers use $O(\log(nd))$ bits of precision. Below, $\gamma > 0$ denotes an arbitrarily small constant.

| Reference | Method | Total runtime | Parallel passes |
|---|---|---|---|
| Folklore | Gaussian sketch | $O(nd^{\omega-1})$ | 1 |
| [43] | Leverage Score Sampling | $\mathrm{nnz}(\mathbf{A}) + \tilde{O}(d^\omega/\sqrt{\epsilon})$ | 1 |
| [5] | Determinantal Point Process | $\mathrm{nnz}(\mathbf{A}) + \tilde{O}(d^\omega)$ | $\log^3(n/\epsilon)$ |
| [9] | Weighted Mb-SGD (sequential) | $\mathrm{nnz}(\mathbf{A}) + \tilde{O}(d^2/\epsilon)$ | $1/\epsilon$ |
| **This work** (Thm. 1) | Leverage Score Sparsification | $\mathrm{nnz}(\mathbf{A}) + \tilde{O}(d^2/\epsilon)$ | 1 |

Table 1: Comparison of time complexities and parallel passes over the data required for different methods to obtain a $(1 + \epsilon)$-approximation in $O(d^2 \log(nd))$ bits of space for an $n \times d$ least squares problem $(\mathbf{A}, \mathbf{b})$, given a preconditioner $\mathbf{P}$ such that $\kappa(\mathbf{AP}) = O(1)$ (see Section 3 for our computational model). We include the fully sequential Weighted Mb-SGD as a reference.

**Theorem 1.** *Given streaming access to* $\mathbf{A} \in \mathbb{R}^{n \times d}$ *and* $\mathbf{b} \in \mathbb{R}^n$, *and direct access to a preconditioner matrix* $\mathbf{P} \in \mathbb{R}^{d \times d}$ *such that* $\kappa(\mathbf{AP}) \leq \alpha$, *within a single pass over* $(\mathbf{A}, \mathbf{b})$, *in* $O(\gamma^{-1}\mathrm{nnz}(\mathbf{A}) + \epsilon^{-1}\alpha d^{2+\gamma}\mathrm{polylog}(d))$ *time and* $O(d^2 \log(nd))$ *bits of space, we can construct a randomized estimator* $\tilde{\mathbf{x}}$ *for the least squares solution* $\mathbf{x}^* = \mathrm{argmin}_{\mathbf{x}} \|\mathbf{Ax} - \mathbf{b}\|^2$ *such that:*

$$\text{(Bias)} \quad \|\mathbf{A}\mathbb{E}[\tilde{\mathbf{x}}] - \mathbf{b}\|^2 \leq (1 + \epsilon)\|\mathbf{Ax}^* - \mathbf{b}\|^2,$$
$$\text{(Variance)} \quad \mathbb{E}[\|\mathbf{A}\tilde{\mathbf{x}} - \mathbf{b}\|^2] \leq 2\|\mathbf{Ax}^* - \mathbf{b}\|^2.$$

**Remark 1.** *The above construction assumes access to a preconditioner matrix* $\mathbf{P}$ *with* $\kappa(\mathbf{AP}) \leq \alpha$ *(where* $\kappa$ *denotes the condition number). Such matrix can be obtained efficiently with* $\alpha = O(1)$ *in a separate single pass, leading to a two-pass algorithm described later in Theorem 2.*

Our estimator $\tilde{\mathbf{x}}$ is constructed at the end of the data pass from a sketch $(\tilde{\mathbf{A}}, \tilde{\mathbf{b}})$ where $\tilde{\mathbf{A}} = \mathbf{SA}$ and $\tilde{\mathbf{b}} = \mathbf{Sb}$, by minimizing $\|\tilde{\mathbf{A}}\mathbf{x} - \tilde{\mathbf{b}}\|^2$ using preconditioned conjugate gradient. Here, $\mathbf{S}$ is a carefully constructed sparse sketching matrix which is inspired by the so-called leverage score sparsified (LESS) embeddings [18]. Leverage scores represent the relative importances of the rows of $\mathbf{A}$ which are commonly used for subsampling in least squares (see Definition 1), and their estimates can be easily obtained in a single pass by using the preconditioner matrix $\mathbf{P}$.

Our time complexity bound of $\tilde{O}(\mathrm{nnz}(\mathbf{A}) + d^2/\epsilon)$ matches the time it would take (for a single machine) to subsample $\tilde{O}(d/\epsilon)$ rows of $\mathbf{A}$ according to the approximate leverage scores and produce an estimator $\tilde{\mathbf{x}}$ that achieves the $\epsilon$-error bound $\|\mathbf{A}\tilde{\mathbf{x}} - \mathbf{b}\|^2 \leq (1 + \epsilon)\|\mathbf{Ax}^* - \mathbf{b}\|^2$. However, this strategy requires either maintaining $\tilde{O}(d^2/\epsilon)$ bits of space for the sketch, or computing $\tilde{\mathbf{x}}$ directly along the way, blowing up the runtime to $\tilde{O}(d^\omega/\epsilon)$. Since approximate leverage score sampling leads to significant least squares bias, averaging can only improve this to $\tilde{O}(d^\omega/\sqrt{\epsilon})$ (see Table 1). An alternate strategy would be to combine leverage score sampling with a preconditioned mini-batch stochastic gradient descent (Weighted Mb-SGD), with mini-batches chosen so that they fit in $\tilde{O}(d^2)$ space. This achieves the same time and space complexity as our method, but due to the streaming access to $\mathbf{A}$ and the sequential nature of SGD, it requires $O(1/\epsilon)$ data passes.

Instead, our algorithm essentially mixes an $\tilde{O}(d/\epsilon)$ size leverage score sample into an $O(d)$ size sketch, merging $\tilde{O}(1/\epsilon)$ rows of $\mathbf{A}$ into a single row of the sketch (see Figure 1). This results in better data compression compared to direct leverage score sampling, with only $\tilde{O}(d^2)$ bits of space, while retaining the same $\tilde{O}(\mathrm{nnz}(\mathbf{A}) + d^2/\epsilon)$ runtime complexity as the above approaches and requiring only a single data pass. The resulting estimator $\tilde{\mathbf{x}}$ can no longer recover the $\epsilon$-error bound, but remarkably, its expectation $\mathbb{E}[\tilde{\mathbf{x}}]$ still does. To turn this into an improved estimator in a distributed model, we can simply average $q = 1/\epsilon$ such estimators, i.e., $\hat{\mathbf{x}} = \frac{1}{q} \sum_{i=1}^{q} \tilde{\mathbf{x}}_i$, obtaining $\mathbb{E}\|\mathbf{A}\hat{\mathbf{x}} - \mathbf{b}\|^2 \leq (1 + 2\epsilon)\|\mathbf{Ax}^* - \mathbf{b}\|^2$. As shown in Table 1, ours is the first result in this model to achieve $\tilde{O}(d^2)$ space in a single pass and faster than current matrix multiplication time $O(d^\omega)$.

Finally, by incorporating a preconditioning scheme, we illustrate how our construction can be used to design the first algorithm that solves least squares in current matrix multiplication time, constant parallel passes and $O(d^2 \log(nd))$ bits of space. We note that the $O(d^\omega)$ cost comes only from the worst-case complexity of constructing the preconditioner $\mathbf{P}$, which can often be accelerated in practice. The computational model used in Theorem 2 is described in detail in Section 3.

**Theorem 2.** *Given* $\mathbf{A} \in \mathbb{R}^{n \times d}$ *and* $\mathbf{b} \in \mathbb{R}^n$ *in the parallel computing model, using two parallel passes with* $q$ *machines, we can compute* $\tilde{\mathbf{x}}$ *such that with probability* $0.9$

$$\|\mathbf{A}\tilde{\mathbf{x}} - \mathbf{b}\| \leq \Big(1 + \epsilon + O(1/q)\Big)\|\mathbf{A}\mathbf{x}^* - \mathbf{b}\|$$

*in* $O(\gamma^{-1}\mathrm{nnz}(\mathbf{A}) + d^\omega + \epsilon^{-1}d^{2+\gamma}\mathrm{polylog}(d))$ *time,* $O(d^2\log(nd))$ *bits of space and* $O(d\log(nd))$ *bits of communication. In particular, choosing* $q = 1/\epsilon$ *we recover an* $O(\epsilon)$-*approximation.*

**Remark 2.** *While the parallel computing model in Theorem 2 assumes that all machines have streaming access to the entire data matrix, this result can be easily extended to the setting where* $\mathbf{A}$ *has been randomly down-sampled or partitioned into separate size* $n$ *chunks* $\mathbf{A}_1, ..., \mathbf{A}_q$, *and each machine constructs an estimate* $\tilde{\mathbf{x}}_i$ *based on a sketch of its own chunk. Then, with the same computational guarantees as in Theorem 2, the averaged estimator* $\tilde{\mathbf{x}} = \frac{1}{q}\sum_{i=1}^{q}\tilde{\mathbf{x}}_i$ *with probability* $0.9$ *enjoys a guarantee of:*

$$\|\mathbf{A}\tilde{\mathbf{x}} - \mathbf{b}\| \leq \Big(1 + O(\epsilon + 1/q + B_{\mathrm{chunk}})\Big)\|\mathbf{A}\mathbf{x}^* - \mathbf{b}\|,$$

*where* $B_{\mathrm{chunk}}$ *is the bias that would be incurred if we solved each chunk exactly and averaged those solutions. Using existing guarantees for uniform down-sampling of least squares [Theorem 20, 43], one can bound this bias as* $B_{\mathrm{chunk}} = \tilde{O}\big(\frac{\mu}{qn} + (\frac{\mu}{n})^2\big)$, *where* $\mu$ *is the coherence of the data matrix and* $n$ *is the chunk size. For sufficiently large chunks, this bias is negligible compared to the error* $\epsilon$. *We prove the above high probability guarantee in Theorem 6 in Appendix E.*

**Further applications.** Our least squares analysis can be extended to other settings where prior works [e.g., 17, 18, 16, 22] have analyzed randomized estimators based on sparse sketching via techniques from asymptotic random matrix theory. The primary and most direct application involves correcting *inversion bias* in the so-called sketched inverse covariance estimate $(\tilde{\mathbf{A}}^\top\tilde{\mathbf{A}})^{-1}$, which was the motivating task of [18], with applications including distributed second-order optimization and statistical uncertainty quantification, where quantities like $(\tilde{\mathbf{A}}^\top\tilde{\mathbf{A}})^{-1}\mathbf{x}$ are approximated.

**Theorem 3** (informal Theorem 5). *Given* $\mathbf{A} \in \mathbb{R}^{n \times d}$ *and its LESS embedding* $\mathbf{S}$ *with sketch size* $m \geq Cd$ *and* $s$ *non-zeros per row, the inverse covariance sketch* $(\frac{m}{m-d}\mathbf{A}^\top\mathbf{S}^\top\mathbf{S}\mathbf{A})^{-1}$ *is an* $(\epsilon, \delta)$-*unbiased estimator of* $(\mathbf{A}^\top\mathbf{A})^{-1}$ *(see Definition 3) for* $\epsilon = \tilde{O}\big((1 + \sqrt{d/s})\frac{\sqrt{d}}{m}\big)$ *and* $\delta = 1/\mathrm{poly}(d)$.

This result should be compared with $\epsilon = \tilde{O}\big((1 + d/s)\frac{\sqrt{d}}{m}\big)$ obtained by [18]. Thus, we get a direct improvement for very sparse sketches, i.e., $s = o(d)$. This can be immediately translated into an improved local convergence guarantee for Distributed Newton Sketch which is a second-order convex minimization algorithm used in settings where the Hessian matrix can be expressed as $\mathbf{A}^\top\mathbf{A}$ for a tall matrix $\mathbf{A}$, e.g., in generalized linear models like logistic regression. In this method, following Corollary 16 of [18] as well as related results [16, 44, 19], we use sketching and averaging to estimate a Newton step:

$$\mathbf{x}_{t+1} = \mathbf{x}_t - \frac{1}{q}\sum_{i=1}^{q}\tilde{\mathbf{H}}_i^{-1}\mathbf{g}_t,$$

where $\mathbf{g}_t$ is the gradient at $\mathbf{x}_t$ and $\tilde{\mathbf{H}}_1, ..., \tilde{\mathbf{H}}_q$ are Hessian sketches constructed by independent machines. The following corollary, which is a direct improvement over Corollary 16 of [18], shows that Distributed Newton Sketch on a generalized linear model task can achieve a fast local convergence rate of the form $\frac{f(\mathbf{x}_{t+1}) - f(\mathbf{x}^*)}{f(\mathbf{x}_t) - f(\mathbf{x}^*)} = o(1)$ with $O(d^\omega)$ time per iteration (see Appendix B.1 for details).

**Corollary 1** (Distributed Newton Sketch). *Consider* $f(\mathbf{x}) = \frac{1}{n}\sum_{i=1}^{n}\ell_i(\mathbf{x}^\top\phi_i) + \frac{\lambda}{2}\|\mathbf{x}\|^2$, *where* $\ell_i$ *are convex twice continuously differentiable functions, such that* $f$ *has a Lipschitz Hessian,* $\lambda > 0$, *and* $\phi_i^\top$ *is the* $i$th *row of an* $n \times d$ *data matrix* $\mathbf{\Phi}$. *Given* $\epsilon > 0$, *there is a neighborhood* $U_\epsilon$ *around the minimizer* $\mathbf{x}^* = \arg\min_{\mathbf{x}} f(\mathbf{x})$ *such that, for any* $\mathbf{x}_t \in U_\epsilon$, *using two parallel passes with* $\tilde{O}(1/\epsilon)$ *machines, we can compute a Distributed Newton Sketch update* $\mathbf{x}_{t+1}$ *such that*

$$f(\mathbf{x}_{t+1}) - f(\mathbf{x}^*) \leq \epsilon \cdot \big[f(\mathbf{x}_t) - f(\mathbf{x}^*)\big],$$

*in* $O(\gamma^{-1}\mathrm{nnz}(\mathbf{\Phi}) + d^\omega + \epsilon^{-1}d^{2+\gamma}\mathrm{polylog}(d))$ *time,* $O(d^2\log(nd))$ *bits of space and* $O(d\log(nd))$ *bits of communication.*

**Our Techniques.** At the core of our analysis are techniques inspired by asymptotic random matrix theory (RMT) in the proportional limit [e.g., see 6]. Here, in order to establish the limiting spectral distribution (such as the Marchenko-Pastur law) of a random matrix $\tilde{\mathbf{A}}^\top \tilde{\mathbf{A}}$ whose dimensions diverge to infinity, one aims to show the convergence of the Stieltjes transform of its resolvent matrix $(\tilde{\mathbf{A}}^\top \tilde{\mathbf{A}} - z\mathbf{I})^{-1}$. Recently, [18] showed that these techniques can be adapted to sparse sketching matrices (via leverage score sparsification) in order to characterize the bias of the sketched inverse covariance $(\tilde{\mathbf{A}}^\top \tilde{\mathbf{A}})^{-1}$, where $\tilde{\mathbf{A}} = \mathbf{S}\mathbf{A}$.

Our main contribution is two-fold. First, we show that a similar argument can also be applied to analyze the bias of the least squares estimator, $\tilde{\mathbf{x}} = (\tilde{\mathbf{A}}^\top \tilde{\mathbf{A}})^{-1}\tilde{\mathbf{A}}^\top \tilde{\mathbf{b}}$. Unlike the inverse covariance, this estimator no longer takes the form of a resolvent matrix, but its bias is also associated with the inverse, which means that we can use a leave-one-out argument to characterize the effect of removing a single row of the sketch on the estimation bias. Our second main contribution is to improve the sharpness of the bounds relative to the sparsity of the sketching matrix by combining a careful application of Hölder's inequality with a higher moments analysis of the restricted Bai-Silverstein inequality for quadratic forms. Those improvements are not only applicable to the least squares analysis, but also to all existing RMT-style results for LESS embeddings, including the aforementioned inverse covariance estimation, as well as applications in stochastic optimization, resulting in the sketching cost of LESS embeddings dropping below matrix multiplication time.

## 2 Related Work

**Randomized numerical linear algebra.** RandNLA sketching techniques have been developed over a long line of works, starting from fast least squares approximations of [42]; for an overview, see [45, 26, 37, 39, 21] among others. Since then, these methods have been used in designing fast algorithms not only for least squares but also many other fundamental problems in numerical linear algebra and optimization including low-rank approximation [13, 34], $l_p$ regression [14], solving linear systems [28, 24] and more. Using sparse random matrices for matrix sketching also has a long history, including data-oblivious sketching methods such as CountSketch [12], OSNAP [40], and more [38]. Leverage score sparsification (LESS) was introduced by [18] as a data-dependent sparse sketching method to enable RMT-style analysis for sketching (see below).

**Unbiased estimators for least squares.** To put our results in a proper context, let us consider other approaches for producing near-unbiased estimators for least squares, see also Table 1. First, a well known folklore result states that the least squares estimator computed from a dense Gaussian sketching matrix is unbiased. The bias of other sketching methods, including leverage score sampling and OSNAP, has been studied by [43], showing that these methods need a $\sqrt{\epsilon}$-error guarantee to achieve an $\epsilon$-bias which leads to little improvement unless $\epsilon$ is extremely small and the sketch size is sufficiently large. Another approach of constructing unbiased estimators for least squares, first proposed by [23], is based on subsampling with a non-i.i.d. importance sampling distribution based on Determinantal Point Processes [DPPs, 31, 20]. However, despite significant efforts [30, 7, 5], sampling from DPPs remains quite expensive: the fastest known algorithm requires running a Markov chain for $\text{polylog}(n/\epsilon)$ many steps, each of which requires a separate data pass and takes $O(d^\omega)$ time. Other approaches have also been considered which provide partial bias reduction for i.i.d. RandNLA subsampling schemes in various regimes that are are either much more expensive or not directly comparable to ours [2, 44].

**Statistical and RMT analysis of sketching.** Recently, there has been significant interest in statistical and random matrix theory (RMT) analysis of matrix sketching; see [21] for an overview. These approaches include both asymptotic analysis via limiting spectral distributions and deterministic equivalents [35, 25, 32, 33], and non-asymptotic analysis under statistical assumptions [36, 41, 4]. A number of works have shown that the RMT-style techniques based on deterministic equivalents can be made rigorously non-asymptotic for certain sketching methods such as dense sub-Gaussian [17], LESS matrices [16, 18], and other sparse matrices [9], which has been applied to low-rank approximation, fast subspace embeddings and stochastic optimization. Our new analysis can be viewed as a general strategy for directly improving the sparsity required by LESS embeddings (and thereby, the sketching time complexity) in many of these applications, specifically those that rely on analysis inspired by the calculus of deterministic equivalents via generalized Stieltjes transforms.

# 3 Preliminaries

**Notations.** In all our results, we use lowercase letters to denote scalars, lowercase boldface for vectors, and uppercase boldface for matrices. The norm $\|\cdot\|$ denotes the spectral norm for matrices and the Euclidean norm for vectors, whereas $\|\cdot\|_F$ denotes the Frobenius norm for matrices. We use $\preceq$ to denote the p.s.d. ordering of matrices.

**Computational model.** We next clarify the computational model that is used in Theorem 2. We consider a central data server storing $(\mathbf{A}, \mathbf{b})$, and $q$ machines. The $j$th machine has a handle $\text{Stream}(j)$, which can be used to *open* a stream and to *read* the next row/label pair $(\mathbf{a}_i, b_i)$ in the stream. After a full pass, the machine can re-open the handle and begin another pass over the data. The machines can operate their streams entirely asynchronously, and each has its own limited local storage space, e.g., in Theorem 2 we use $O(d^2 \log(nd))$ bits of space per machine. At the end, they can communicate some information back to the server, e.g., in Theorem 2, they communicate their final estimate vectors $\tilde{\mathbf{x}}_j$, using $O(d \log(nd))$ bits of communication. Then, the server computes the final estimate, in our case via averaging, $\tilde{\mathbf{x}} = \frac{1}{q} \sum_{i=1}^{q} \tilde{\mathbf{x}}_i$, which can be done either directly or via a map-reduce type architecture.

We define the *parallel passes* required by such an algorithm as the maximum number of times the stream is opened by any single machine. We analogously define time/space/communication costs by taking a maximum over the costs required by any single machine (for communication, this refers only to the number of bits sent from the machine back to the server).

**Definitions and useful lemmas.** In our framework, we construct a sparse sketching matrix $\mathbf{S}$ where sparsification is achieved using a probability distribution over rows of data matrix $\mathbf{A}$, that is proportional to the leverage scores of $\mathbf{A}$. The next definition [following, e.g., 9] provides the explicit definition of exact and approximate leverage scores for our setting.

**Definition 1** (($\beta_1, \beta_2$)-approximate leverage scores). *Fix a matrix $\mathbf{A} \in \mathbb{R}^{n \times d}$ and consider matrix $\mathbf{U} \in \mathbb{R}^{n \times d}$ with orthonormal columns spanning the column space of $\mathbf{A}$. Then, the leverage scores $l_i, 1 \leq i \leq n$ are defined as the row norms squared of $\mathbf{U}$, i.e., $l_i = \|\mathbf{u}_i\|^2$, where $\mathbf{u}_i^\top$ is the $i$th row of $\mathbf{U}$. Furthermore, consider fixed $\beta_1, \beta_2 > 1$. Then $\tilde{l}_i$ are called ($\beta_1, \beta_2$)-approximate leverage scores for $\mathbf{A}$ if the following holds for all $i$*

$$\frac{l_i}{\beta_1} \leq \tilde{l}_i \quad \text{and} \quad \sum_{i=1}^{n} \tilde{l}_i \leq \beta_2 \cdot d.$$

The approximate leverage scores can be computed by first constructing a preconditioner matrix $\mathbf{P} \in \mathbb{R}^{d \times d}$ such that $\kappa(\mathbf{AP}) = O(1)$, which takes $O(\text{nnz}(\mathbf{A}) + d^\omega)$ in a single pass, and then relying on the following norm approximation scheme.

**Lemma 1** (Based on Lemma 7.2 from [10]). *Given $\mathbf{A} \in \mathbb{R}^{n \times d}$ and $\mathbf{P} \in \mathbb{R}^{d \times d}$, using a single pass over $\mathbf{A}$ in time $O(\gamma^{-1}(\text{nnz}(\mathbf{A}) + d^2))$ for small constant $\gamma > 0$, we can compute estimates $\tilde{l}_1, ..., \tilde{l}_n$ such that with probability $\geq 0.95$:*

$$n^{-\gamma} \|e_i^\top \mathbf{AP}\|^2 \leq \tilde{l}_i \leq O(\log(n)) \|\mathbf{e}_i^\top \mathbf{AP}\|^2 \quad \forall i \qquad \text{and} \qquad \sum_i \tilde{l}_i \leq O(1) \cdot \|\mathbf{AP}\|_F^2.$$

In the next definition, we give the sparse sketching strategy used in our analysis. This approach is similar to the original leverage score sparsification proposed by [18], except: 1) we adapted it so that it can be implemented effectively in a single pass, and 2) we use it in a much sparser regime (fewer non-zeros per row).

**Definition 2** (($s, \beta_1, \beta_2$)-LESS embedding). *Fix a matrix $\mathbf{A} \in \mathbb{R}^{n \times d}$ and some $s \geq 0$. Let the tuple $(\tilde{l}_1, \cdots, \tilde{l}_n)$ denote ($\beta_1, \beta_2$)-approximate leverage scores for $\mathbf{A}$. Let $p_i = \min\{1, \frac{s\beta_1 \tilde{l}_i}{d}\}$. We define a ($s, \beta_1, \beta_2$)-approximate leverage score sparsifier $\boldsymbol{\xi}$ as follows.*

$$\boldsymbol{\xi} = \left( \frac{b_1}{\sqrt{p_1}}, \cdots, \frac{b_n}{\sqrt{p_n}} \right) \quad \text{where} \quad b_i \sim \text{Bernoulli}(p_i).$$

*Moreover, we define the ($s, \beta_1, \beta_2$)-leverage score sparsified (LESS) embedding of size $m$ as matrix $\mathbf{S} \in \mathbb{R}^{m \times d}$ with i.i.d. rows $\frac{1}{\sqrt{m}} \mathbf{x}_i$ such that $\mathbf{x}_i = \text{diag}(\boldsymbol{\xi}_i) \mathbf{y}_i$ where $\boldsymbol{\xi}_i$ denotes a randomly generated ($\beta_1, \beta_2$)-approximate leverage score sparsifier and $\mathbf{y}_i \in \mathbb{R}^n$ consist of random $\pm 1$ entries.*

A key property of a sketching matrix is the subspace embedding property, defined below. It was recently shown by [9] that LESS embeddings require only polylogarithmically many non-zeros per row of $\mathbf{S}$ to prove that $\mathbf{S}$ is a subspace embedding for the data matrix $\mathbf{A}$ with the optimal $m = O(d)$ sketching dimension. For our main results, it is sufficient to use $\eta = O(1)$ below.

**Lemma 2** (Subspace embedding for LESS, Theorem 1.3, [9]). *Fix $\eta, \delta > 0$. Consider $\beta_1, \beta_2 > 1$ and a full rank matrix $\mathbf{A} \in \mathbb{R}^{n \times d}$. Then for a $(\beta_1, \beta_2)$-leverage score sparsified embedding $\mathbf{S} \in \mathbb{R}^{m \times n}$ with $s \geq O(\log^4(d/\delta)/\eta^4)$ and $m = \mathcal{O}((d + \log 1/\delta)/\eta^2)$, with probability $1 - \delta$ we have*

$$\frac{1}{1 + \eta} \cdot \mathbf{A}^\top \mathbf{A} \preceq \mathbf{A}^\top \mathbf{S}^\top \mathbf{S} \mathbf{A} \preceq (1 + \eta) \cdot \mathbf{A}^\top \mathbf{A}. \tag{1}$$

# 4 Least Squares Bias Analysis

In this section we provide an outline of the bias analysis for the sketched least squares estimator constructed using a LESS embedding, leading to the proofs of our main results, Theorems 1 and 2. In particular, we prove the following main technical result (detailed proof in Appendix C).

**Theorem 4** (Bias of LESS-sketched least squares). *Fix $\mathbf{A} \in \mathbb{R}^{n \times d}$ and let $\mathbf{S}$ be an $(s, \beta_1, \beta_2)$-LESS embedding of size $m$ for $\mathbf{A}$. Let $\mathbf{S}$ satisfy (1) with $\eta = \frac{1}{2}$ and probability $1 - \delta$ where $\delta < \frac{1}{m^4}$. Then there exists an event $\mathcal{E}$ with probability at least $1 - \delta$ such that*

$$L(\mathbb{E}_\mathcal{E}[\tilde{\mathbf{x}}]) - L(\mathbf{x}^*) = \mathcal{O}\left(\frac{d}{m^2}\left(1 + \frac{d}{s}\right)\log^9(n/\delta)\right) \cdot L(\mathbf{x}^*).$$

**Remark 3.** *Thus, the bias $L(\mathbb{E}_\mathcal{E}[\tilde{\mathbf{x}}]) - L(\mathbf{x}^*)$ of the LESS estimator using $O(\beta_1 \beta_2 s) = \tilde{O}(s)$ non-zeros per row of $\mathbf{S}$ is of the order $\tilde{O}(\frac{d^2}{sm^2} + \frac{d}{m^2}) \cdot L(\mathbf{x}^*)$. By comparison, the standard expected loss bound which holds for sketched least squares (including this estimator) is $\mathbb{E}[L(\tilde{\mathbf{x}})] - L(\mathbf{x}^*) \leq \tilde{O}(\frac{d}{m})L(\mathbf{x}^*)$, and the best known bound on the bias of most standard sketched estimators (e.g., leverage score sampling) is $\tilde{O}(\frac{d^2}{m^2})L(\mathbf{x}^*)$, given by [43]. So, our result recovers the standard bias bound for $s = 1$ and improves on it for $s \gg 1$ by a factor of $\min\{s, d\}$. At the end of the section, we discuss how to deal with the lower order term $\tilde{O}(\frac{d}{m^2})$ to reduce the bias further.*

**Proof sketch.** Using a standard argument, we can replace the matrix $\mathbf{A}$ with the matrix $\mathbf{U} \in \mathbb{R}^{n \times d}$ consisting of orthonormal columns spanning the column space of $\mathbf{A}$, and assume that $n = \text{poly}(d)$. Let $\mathbf{S}$ be an $(s, \beta_1, \beta_2)$-LESS embedding for $\mathbf{U}$. Also, let $\mathbf{b} \in \mathbb{R}^n$ be a vector of responses/labels corresponding to $n$ rows in $\mathbf{U}$. Let $\tilde{\mathbf{x}} = \text{argmin}_\mathbf{x} \|\mathbf{SUx} - \mathbf{Sb}\|^2$. Furthermore for any $\mathbf{x} \in \mathbb{R}^d$ we can find the loss at $\mathbf{x}$ as $L(\mathbf{x}) = \|\mathbf{Ux} - \mathbf{b}\|^2$. Additionally, we use $\mathbf{r}$ to denote the residual $\mathbf{b} - \mathbf{Ux}^*$. We also define $\mathbf{Q} = (\gamma \mathbf{U}^\top \mathbf{S}^\top \mathbf{SU})^{-1}$ as the sketched inverse covariance matrix with scaling $\gamma = \frac{m}{m-d}$ representing the standard correction accounting for inversion bias. We condition on the high probability event $\mathcal{E}$ guaranteed in Lemma 2 and consider $L(\mathbb{E}_\mathcal{E}(\tilde{\mathbf{x}})) - L(\mathbf{x}^*)$. By Pythagorean theorem, we have $L(\mathbb{E}_\mathcal{E}[\tilde{\mathbf{x}}]) - L(\mathbf{x}^*) = \|\mathbf{U}(\mathbb{E}_\mathcal{E}[\tilde{\mathbf{x}}]) - \mathbf{Ux}^*\|^2$. Note that by the normal equations we have $\tilde{\mathbf{x}} = (\mathbf{U}^\top \mathbf{S}^\top \mathbf{SU})^{-1} \mathbf{U}^\top \mathbf{S}^\top \mathbf{Sb} = \gamma \mathbf{QU}^\top \mathbf{S}^\top \mathbf{Sb}$, and also $\mathbf{S}^\top \mathbf{S} = \frac{1}{m} \sum_{i=1}^m \mathbf{x}_i \mathbf{x}_i^\top$. These two facts lead to writing the bias as follows:

$$L(\mathbb{E}_\mathcal{E}[\tilde{\mathbf{x}}]) - L(\mathbf{x}^*) = \|\gamma \cdot \mathbb{E}_\mathcal{E}[\mathbf{QU}^\top \mathbf{x}_i \mathbf{x}_i^\top \mathbf{r}]\|^2.$$

Using a leave-one-out technique, we replace $\mathbf{Q}$ with $\mathbf{Q}_{-i} = (\gamma \mathbf{U}^\top \mathbf{S}_{-i}^\top \mathbf{S}_{-i} \mathbf{U})^{-1}$, where $\mathbf{S}_{-i}$ denotes matrix $\mathbf{S}$ without the $i$th row, by noting that $\mathbf{Q} = (\gamma \mathbf{U}^\top \mathbf{S}_{-i}^\top \mathbf{S}_{-i} \mathbf{U} + \frac{\gamma}{m} \mathbf{U}^\top \mathbf{x}_i \mathbf{x}_i^\top \mathbf{U})^{-1}$ and applying the Sherman-Morrison formula. This leads to the following relation:

$$L(\mathbb{E}_\mathcal{E}[\tilde{\mathbf{x}}]) - L(\mathbf{x}^*) \leq 2 \underbrace{\|\mathbb{E}_\mathcal{E}[\mathbf{Q}_{-i} \mathbf{U}^\top \mathbf{x}_i \mathbf{x}_i^\top \mathbf{r}]\|^2}_{\|\mathbf{Z}_0 \mathbf{r}\|^2} + 2 \underbrace{\left\|\mathbb{E}_\mathcal{E}\left[\left(\frac{\gamma}{\gamma_i} - 1\right) \mathbf{Q}_{-i} \mathbf{U}^\top \mathbf{x}_i \mathbf{x}_i^\top \mathbf{r}\right]\right\|^2}_{\|\mathbf{Z}_2 \mathbf{r}\|^2}$$

where $\gamma_i = 1 + \frac{\gamma}{m} \mathbf{x}_i^\top \mathbf{U} \mathbf{Q}_{-i} \mathbf{U}^\top \mathbf{x}_i$. Due to the subspace embedding assumption and assuming $m$ large enough, we have $\|\mathbf{Q}\| = O(1)$ and also $\|\mathbf{Q}_{-i}\| = O(1)$. The first term $\|\mathbf{Z}_0 \mathbf{r}\|^2$ is quite straightforward to bound since, if not for the conditioning on the high probability event $\mathcal{E}$, we would have $\mathbb{E}[\mathbf{Q}_{-i} \mathbf{U}^\top \mathbf{x}_i \mathbf{x}_i^\top \mathbf{r}] = \mathbb{E}[\mathbf{Q}_{-i} \mathbf{U}^\top \mathbf{r}] = \mathbf{0}$, which follows from $\mathbf{U}^\top (\mathbf{b} - \mathbf{Ux}^*) = \mathbf{0}$. We get an upper bound on $\|\mathbf{Z}_0 \mathbf{r}\|^2$ as $O\left(\frac{d^2 \log(d/\delta)}{sm^2} + \frac{d}{m^2}\right) \cdot \|\mathbf{r}\|^2$, which is sufficient for us.

The central novelty of our analysis lies in bounding $\|\mathbf{Z}_2\mathbf{r}\|^2$ for $(s, \beta_1, \beta_2)$-LESS embeddings, which is the dominant term. Our key observation is that, when examining a random variable of the form $\mathbf{x}_i^\top \mathbf{v}$ for some vector $\mathbf{v}$, the dependence on the sparsity of row $\mathbf{x}_i$ only arises when considering moments higher than $2 + \frac{1}{O(\log(n))}$, because otherwise we can simply rely on the fact that $\mathbb{E}[\mathbf{x}_i\mathbf{x}_i^\top] = \mathbf{I}$. Thus, when decomposing $\|\mathbf{Z}_2\mathbf{r}\|^2$, we must carefully separate the contribution of near-second moments vs the contribution of higher moments to the overall bound.

To obtain this separation, we start by applying Hölder's inequality on $\|\mathbf{Z}_2\mathbf{r}\|$ with $p = O(\log(n))$ and $q = 1 + \frac{1}{O(\log(n))}$ to get

$$\|\mathbf{Z}_2\mathbf{r}\| \leq \left(\mathbb{E}_\mathcal{E}[|\frac{\gamma}{\gamma_i} - 1|^p]\right)^{1/p} \cdot \left(\sup_{\|\mathbf{v}\|=1} \mathbb{E}_\mathcal{E}\left[\mathbf{v}^\top \mathbf{Q}_{-i}\mathbf{U}^\top \mathbf{x}_i\mathbf{x}_i^\top \mathbf{r}\right]^q\right)^{1/q}.$$

Furthermore applying Cauchy-Schwarz inequality on the second term leads to

$$\|\mathbf{Z}_2\mathbf{r}\| \leq \left(\mathbb{E}_\mathcal{E}[|\frac{\gamma}{\gamma_i} - 1|^p]\right)^{1/p} \cdot \left(\mathbb{E}_\mathcal{E}\|\mathbf{Q}_{-i}\mathbf{U}^\top \mathbf{x}_i\|^{2q}\right)^{1/2q} \cdot \left(\mathbb{E}_\mathcal{E}\|\mathbf{x}_i^\top \mathbf{r}\|^{2q}\right)^{1/2q}.$$

Unlike [18], we exploit the fact that $\|\mathbf{x}_i\|^{1/O(\log(n))} = O(1)$ and get a constant upper bound on $(\mathbb{E}_\mathcal{E}\|\mathbf{Q}_{-i}\mathbf{U}^\top \mathbf{x}_i\|^{2q})$. However, this results in a much more careful argument, requiring now an upper bound on $(\mathbb{E}_\mathcal{E}[|\frac{\gamma}{\gamma_i} - 1|^p])^{1/p}$ for $p = O(\log(n))$. First, we observe that

$$\left(\mathbb{E}_\mathcal{E}[|\frac{\gamma}{\gamma_i} - 1|^p]\right)^{1/p} \leq |\gamma - \bar\gamma| + \left(\mathbb{E}_\mathcal{E}\left[(\gamma_i - \bar\gamma)^p\right]\right)^{1/p} \tag{2}$$

where $\bar\gamma = 1 + \frac{\gamma}{m}\mathbb{E}_\mathcal{E}\left(\mathbf{x}_i^\top \mathbf{U}\mathbf{Q}_{-i}\mathbf{U}^\top \mathbf{x}_i\right)$. In particular, for the second term, we have

$$\left(\mathbb{E}_\mathcal{E}\left[(\gamma_i - \bar\gamma)^p\right]\right)^{1/p} \leq \left(\frac{\gamma}{m}\right) \cdot \left[\left(\mathbb{E}_\mathcal{E}\left[(\mathrm{tr}(\mathbf{Q}_{-i}) - \mathbf{x}_i^\top \mathbf{U}\mathbf{Q}_{-i}\mathbf{U}^\top \mathbf{x}_i)^p\right]\right)^{1/p} + \left(\mathbb{E}_\mathcal{E}\left[\mathrm{tr}(\mathbf{Q}_{-i}) - \mathbb{E}_\mathcal{E}\mathrm{tr}(\mathbf{Q}_{-i})\right]^p\right)^{1/p}\right]. \tag{3}$$

To bound the first of these two terms, we prove a new version of the Restricted Bai-Silverstein inequality (Lemma 3) for $(s, \beta_1, \beta_2)$-LESS embeddings. Unlike [18], we provide a proof with any $p$ and any $(\beta_1, \beta_2)$ values. Furthermore, utilizing the subspace embedding guarantee from Lemma 2, we prove a much more general result where the number of non-zeros in the approximate leverage score sparsifier $\boldsymbol\xi$ can be much smaller than $d$ (proof in Appendix D).

**Lemma 3** (Restricted Bai-Silverstein for $(s, \beta_1, \beta_2)$-LESS embeddings). *Let $p \in \mathbb{N}$ be fixed and $\mathbf{U} \in \mathbb{R}^{n \times d}$ be such that $\mathbf{U}^\top \mathbf{U} = \mathbf{I}$. Let $\mathbf{x}_i = \mathrm{diag}(\boldsymbol\xi)\mathbf{y}_i$ where $\mathbf{y}_i \in \mathbb{R}^n$ has independent $\pm 1$ entries and $\boldsymbol\xi$ is an $(s, \beta_1, \beta_2)$-approximate leverage score sparsifier for $\mathbf{U}$. Then for any matrix with $0 \preceq \mathbf{C} \preceq \mathcal{O}(1) \cdot \mathbf{I}$ and any $\delta > 0$ we have for an absolute constant $c > 0$.*

$$\left(\mathbb{E}\left[\mathrm{tr}(\mathbf{C}) - \mathbf{x}_i^\top \mathbf{U}\mathbf{C}\mathbf{U}^\top \mathbf{x}_i\right]^p\right)^{1/p} < c \cdot \sqrt{d}p^3 \cdot \left(1 + \sqrt{\frac{dp\log(d/\delta)}{s}}\right).$$

Using Lemma 3, we upper bound the first term squared in (3) as $\tilde{O}\left(\frac{d}{m^2}\left(1 + \frac{d}{s}\right)\right)$. Moreover, also using Lemma 3, we get a matching upper bound on $|\gamma - \bar\gamma|$. We then design a martingale concentration argument to prove a high probability upper bound on the last remaining term, $|\mathrm{tr}(\mathbf{Q}_{-i}) - \mathbb{E}_\mathcal{E}\mathrm{tr}(\mathbf{Q}_{-i})|$, which implies the desired moment bound (proof in Appendix B), concluding the proof of Theorem 4.

**Lemma 4.** *For given $\delta > 0$ and matrix $\mathbf{Q}_{-i}$ we have with probability $1 - \delta$:*

$$|\mathrm{tr}(\mathbf{Q}_{-i}) - \mathbb{E}\mathrm{tr}(\mathbf{Q}_{-i})| \leq c'\gamma \cdot \frac{d}{\sqrt{m}}\log^{4.5}(m/\delta).$$

**Completing the proof of Theorem 1.** First, suppose that $\epsilon \geq O(\mathrm{polylog}(d)/d)$ so that the bias bound can be achieved from Theorem 4. Our implementation is mainly based on the online construction of approximate leverage scores, given the preconditioner $\mathbf{P}$, using Lemma 1. Briefly, this construction proceeds by first sketching $\mathbf{P}$ using a $d \times O(1/\gamma)$ Gaussian matrix $\mathbf{G}$ to produce the matrix $\tilde{\mathbf{P}} = \mathbf{P}\mathbf{G}$, and then, for each observed row $\mathbf{a}_i$ of $\mathbf{A}$, we compute $\tilde{l}_i = \|\mathbf{a}_i^\top \tilde{\mathbf{P}}\|^2$. Assuming without loss of generality that $d = \mathrm{poly}(n)$ and adjusting $\gamma$, the estimates satisfy $\beta_1\beta_2 = O(\alpha d^\gamma)$.

Next, we sample the non-zero entries of $\mathbf{S}$ corresponding to the observed row $\mathbf{a}_i$, i.e., the $i$-th column of $\mathbf{S}$. Note that for this we only need to know the single leverage score estimate $\tilde{l}_i$. Crucially for our analysis, the entries of this column need to be sampled i.i.d., which can be done in time proportional to the number of non-zeros in that column by first sampling a corresponding Binomial distribution to determine how many non-zeros we need, then picking a random subset of that size, and then sampling the random $\pm 1$ values. Altogether, the cost of constructing the sketch is $O(\gamma^{-1}\mathrm{nnz}(\mathbf{A}) + \beta_1\beta_2 sd^2) = O(\gamma^{-1}\mathrm{nnz}(\mathbf{A}) + \alpha\epsilon^{-1}d^{2+\gamma}\mathrm{polylog}(d))$ by setting $s = O(\mathrm{polylog}(d)/\epsilon)$. Finally, once we construct the sketch, at the end of the pass we can run conjugate gradient preconditioned with $\mathbf{P}$ on the sketched problem, which takes $\tilde{O}(\alpha d^2)$.

We note that in the (somewhat artificial) regime where we require extremely small bias, i.e., $\epsilon = o(\mathrm{polylog}(d)/d)$, the bound claimed in Theorem 1 can still be obtained, since in this case for small enough $\gamma$ we have $d^{2+\gamma}/\epsilon = O(d^\omega/\sqrt{\epsilon})$ with $\omega < 2.5$, so we can rely on direct leverage score sampling (which corresponds to $s = 1$), and instead of maintaining the sketch, we compute the estimator $\tilde{\mathbf{x}} = (\tilde{\mathbf{A}}^\top\tilde{\mathbf{A}})^{-1}\tilde{\mathbf{A}}^\top\mathbf{b}$ directly along the way. This involves performing a separate $d \times d$ matrix multiplication after collecting each $d$ leverage score samples, to gradually compute $\tilde{\mathbf{A}}^\top\tilde{\mathbf{A}}$, and then inverting the matrix at the end. From Theorem 4, we see that it suffices to set sketch size $m = \tilde{O}(d/\sqrt{\epsilon})$, which leads to the desired runtime.

**Completing the proof of Theorem 2.**   Here, we use a slightly modified variant of Lemma 2, given as Theorem 1.4 in [9], which shows that using a single pass we can compute a sketch $\tilde{\mathbf{A}}$ in time $O(\mathrm{nnz}(\mathbf{A}) + d^\omega)$, which satisfies the subspace embedding property (1) with $\eta = \frac{1}{2}$. Then, we can perform the QR decomposition $\tilde{\mathbf{A}} = \mathbf{Q}\mathbf{R}$ and set $\mathbf{P} = \mathbf{R}^{-1}$ in additional time $O(d^\omega)$ to obtain the desired preconditioner. Next, we use Theorem 1 to construct $q$ i.i.d. estimators $\tilde{\mathbf{x}}_i$ in a second parallel pass, and finally, the estimators are aggregated to compute $\hat{\mathbf{x}} = \frac{1}{q}\sum_{i=1}^{q}\tilde{\mathbf{x}}_i$, which satisfies $\mathbb{E}\|\mathbf{A}\hat{\mathbf{x}} - \mathbf{b}\|^2 \leq \left(1 + \epsilon + O(1/q)\right)\|\mathbf{A}\mathbf{x}^* - \mathbf{b}\|^2$. Applying Markov's inequality concludes the proof.

# 5   Experiments

In this section, we illustrate empirically how our results point to a practical free lunch phenomenon in distributed averaging of sketching-based estimators. As mentioned in Section 1, our construction from Theorem 1 essentially works by taking a subsample of the data and then mixing groups of those rows together to produce an even smaller sketch (see Figure 1). According to our theory, while the small sketch does not recover the same $\epsilon$-small error as the larger subsample, it does recover an $\epsilon$-small bias. Moreover, this happens without incurring any additional computational cost, as the cost of the sketching is proportional to the cost of simply reading the subsampled rows. This suggests that we can use sparse sketching to compress a data subsample down to a small size while retaining the least squares performance in a distributed averaging environment.

To verify this, we evaluate the effectiveness of distributed averaging of sketched least squares estimators on several benchmark datasets. Specifically, we visualize the relative error of the averaged sketch-and-solve estimator $\frac{L(\hat{\mathbf{x}}) - L(\mathbf{x}^*)}{L(\mathbf{x}^*)}$, against the number of machines $q$ used to generate the estimate $\hat{\mathbf{x}} = \frac{1}{q}\sum_{i=1}^{q}\tilde{\mathbf{x}}_i$. Each estimate $\tilde{\mathbf{x}}_i$ is constructed with the same sparsification strategy used by LESS, except that instead of sparsifying the sketch with leverage scores, we instead sparsify them with uniform probabilities (which is often sufficient in practice). Following [16], we call the resulting method LESSUniform. Within each dataset, we perform four simulations, designed so that the total sketching cost stays the same for all four test cases, by simultaneously changing sketch size and sparsity. Concretely, we vary these so that the product (sketch size $\times$ nnz per row) stays the same in each case, so as to ensure that the total cost of sketching is fixed in each plot.

In Figure 2, on the X-axis we plot the number $q$ of estimators being averaged, so that the bias of a single estimator appears on the right-hand side of the plot (large $q$), whereas the variance (error) appears on the left-hand side ($q = 1$). In each plot, the line with nnz per row = 1 (and large sketch size) corresponds to uniform subsampling, whereas the remaining ones are sketches produced by compressing that subsample. The plot shows that decreasing the sketch size (i.e., compressing the sample) does increase the error of a single estimator (as expected), however it also shows that the bias of these estimators remains essentially unchanged regardless of the sketch size (since all lines

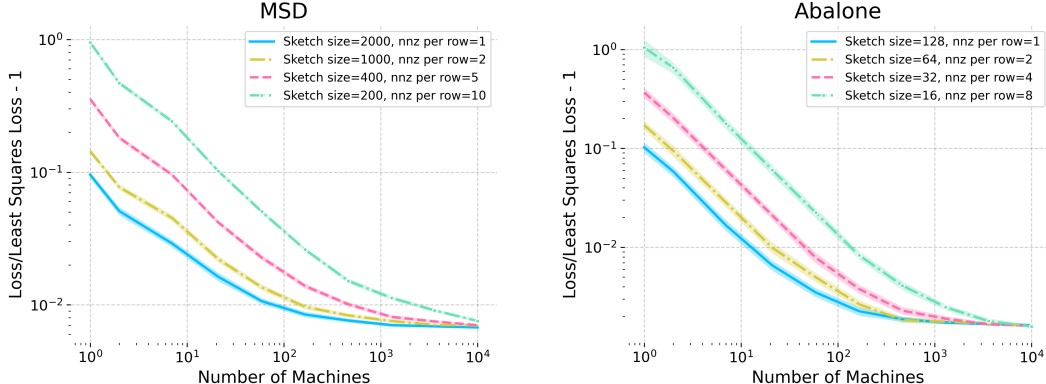

Figure 2: Distributed averaging experiment on `YearPredictionMSD` and `Abalone` datasets [8], showing that sparse sketching can be used to compress the data while preserving near-unbiasedness without increasing the estimation cost (see Appendix F for results on the `Boston` dataset).

meet as $q \to \infty$), confirming that suitable sparse sketches that "compress" rows of data can preserve near-unbiasedness without increasing the cost.

This phenomenon may not occur for all sketching methods. Figure 4 within Appendix F showcases a few more interesting results. We first further demonstrate that suitable sketches that "compress" rows of data $\mathbf{A}$ into a single row of $\tilde{\mathbf{A}}$ can preserve near-unbiasedness without increasing the cost. In particular, this desirable phenomenon that LESSUniform enjoys also extends to LESS proper, as well as the Gaussian and Subgaussian (Rademacher) sketches. In fact, we observe that LESS enjoys similar desirable performance as the Gaussian and Subgaussian sketches and virtually no least squares bias, while retaining the computational speedups of sparse sketching, suggesting that it attains the best of both worlds.

However, when we decrease the number of subsamples within leverage score subsampling, the bias introduced by subsampling increases as expected. This happens as the number of subsamples is reduced without increasing the amount of "compression" as one would with LESS or LESSUniform.[1] We also show that the subsampled randomized Hadamard transform (SRHT) can exhibit some amount of least squares bias as the sketch size decreases. The numerical results shown for the bias introduced by leverage score subsampling and the SRHT complement the lower bounds established in [18].

## 6 Conclusions

We gave a new sparse sketching method that, using two passes over the data, produces a nearly-unbiased least squares estimator, which can be used to improve upon the space-time trade-offs of solving least squares in parallel or distributed environments via simple averaging. In particular, our algorithm is the first to require only $O(d^2 \log(nd))$ bits of space and current matrix multiplication time $O(d^\omega)$ while obtaining an $\epsilon = o(1)$ approximation in few passes. Our techniques are of broader interest to sketching-based optimization algorithms, including Distributed Newton Sketch.

## Acknowledgments

This work was partially supported by NSF CAREER CCF-2338655.

---

[1]One can think of LESS and LESSUniform as generalizations of leverage score subsampling and uniform subsampling respectively, where we mix $\tilde{O}(1/\epsilon)$ subsamples into a single row of the sketch. This suggests that sparse sketching, as an extension of subsampling, yields desirable bias reduction without significantly increasing computational costs.

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

# A Detailed preliminaries

We start by providing several classical results, used in our analysis. The following formula provides a way to compute the inverse of matrix $\mathbf{A}$ after a rank-1 update, given the inverse before the update.

**Lemma 5** (Sherman-Morrison formula). *For an invertible matrix $\mathbf{A} \in \mathbb{R}^{d \times d}$ and vector $\mathbf{u}, \mathbf{v} \in \mathbb{R}^d$, $\mathbf{A} + \mathbf{u}\mathbf{v}^\top$ is invertible if and only if $1 + \mathbf{v}^\top \mathbf{A}^{-1} \mathbf{u} \neq 0$. If this holds then,*

$$(\mathbf{A} + \mathbf{u}\mathbf{v}^\top)^{-1} = \mathbf{A}^{-1} - \frac{\mathbf{A}^{-1}\mathbf{u}\mathbf{v}^\top \mathbf{A}^{-1}}{1 + \mathbf{v}^\top \mathbf{A}^{-1} \mathbf{u}}.$$

*In particular,*

$$(\mathbf{A} + \mathbf{u}\mathbf{v}^\top)^{-1}\mathbf{u} = \frac{\mathbf{A}^{-1}\mathbf{u}}{1 + \mathbf{v}^\top \mathbf{A}^{-1} \mathbf{u}}.$$

The following inequality provides a crucial tool for writing expectation of the product of two random variables as the product of higher individual moments.

**Lemma 6** (Hölder's inequality). *For real-valued random variables $X$ and $Y$,*

$$\mathbb{E}[|XY|] \leq (\mathbb{E}[|X|^p])^{1/p} \cdot (\mathbb{E}[|Y|^q])^{1/q}$$

*where $p, q > 0$ are Hölder's conjugates, i.e. $\frac{1}{p} + \frac{1}{q} = 1$.*

The following technical lemmas provide concentration results for the sum of random quantities. We collect these results here and then refer to them while using in our analysis.

**Lemma 7** (Matrix Chernoff Inequality). *For $i = 1, 2, \cdots, n$ consider a sequence $\mathbf{Z}_i$ of $d \times d$ positive semi-definite random matrices such that $\mathbb{E}[\frac{1}{n} \sum_i \mathbf{Z}_i] = \mathbf{I}_d$ and $\|\mathbf{Z}_i\| \leq R$. Then for any $\epsilon > 0$, we have*

$$\Pr\left(\lambda_{\max}\left(\frac{1}{n}\sum_{i=1}^n \mathbf{Z}_i\right) \geq (1+\epsilon)\right) \leq d \cdot \exp\left(-\frac{n\epsilon^2}{(2+\epsilon)R}\right).$$

*where $\lambda_{max}$ denotes taking the maximum eigenvalue.*

**Lemma 8** (Azuma's inequality). *If $\{Y_0, Y_1, Y_2, \cdots\}$ is a martingale with $|Y_j - Y_{j-1}| \leq c_j$ then for any $m, \lambda > 0$ we have*

$$\Pr\left(|Y_m - Y_0| \geq \lambda\right) \leq 2 \cdot \exp\left(-\frac{\lambda^2}{2\sum_{j=1}^m c_j^2}\right).$$

**Lemma 9** (Rosenthal's inequality ([29], Theorem 2.5 and Corollary 2.6)). *Let $1 \leq p < \infty$ and $X_1, X_2, \cdots, X_n$ are nonnegative, independent random variables with finite $p^{th}$ moments then,*

$$\left(\mathbb{E}\left[\sum_i X_i\right]^p\right)^{1/p} \leq \frac{2p}{\log(p)} \cdot \max\left\{\sum_i \mathbb{E}[X_i], \left(\sum_i \mathbb{E}[X_i^p]\right)^{1/p}\right\}.$$

*Furthermore, for mean-zero independent and symmetric random variables we have*

$$\left(\mathbb{E}\left[\sum_i X_i\right]^p\right)^{1/p} \leq \frac{2p}{\sqrt{\log(p)}} \cdot \max\left\{\left(\sum_i \mathbb{E}[X_i^2]\right)^{1/2}, \left(\sum_i \mathbb{E}[X_i^p]\right)^{1/p}\right\}.$$

**Lemma 10** (Bai-Silverstein's Inequality Lemma B.26 from [6]). *Let $\mathbf{B}$ be a $d \times d$ be a fixed matrix and $\mathbf{x}$ be a random vector of independent entries. Let $\mathbb{E}[x_i] = 0$ and $\mathbb{E}[x_i^2] = 1$, and $\mathbb{E}|x_j|^l \leq \nu_l$. Then for any $p \geq 1$,*

$$\mathbb{E}|\mathbf{x}^\top \mathbf{B}\mathbf{x} - \mathrm{tr}(\mathbf{B})|^p \leq (2p)^p \cdot \left((\nu_4 \mathrm{tr}(\mathbf{B}\mathbf{B}^\top))^{p/2} + \nu_{2p}\mathrm{tr}(\mathbf{B}\mathbf{B}^\top)^{p/2}\right).$$

# B   Inversion bias analysis

We use the notion of an $(\epsilon, \delta)$ unbiased estimator of $\mathbf{A}$ as defined in [18].

**Definition 3** (($\epsilon, \delta$)-unbiased estimator)**.** *For $\epsilon, \delta > 0$, a random positive definite matrix $\mathbf{B} \in \mathbb{R}^{d \times d}$ is called an $(\epsilon, \delta)$ unbiased estimator of $\mathbf{A}$ if there exists an event $\mathcal{E}$ with $\Pr(\mathcal{E}) \geq 1 - \delta$ such that,*

$$\frac{1}{1+\epsilon}\mathbf{A} \preceq \mathbb{E}_\mathcal{E}[\mathbf{B}] \preceq (1+\epsilon)\mathbf{A} \quad and, \quad \mathbf{B} \preceq O(1) \cdot \mathbf{A},$$

*when conditioned on the event $\mathcal{E}$.*

In this section, we give a formal statement and proof for Theorem 3, which is then used in the proof of Theorem 4. We replace $\mathbf{A}$ with $\mathbf{U}$ such that $\mathbf{U}$ consists of $d$ orthonormal columns spanning the column space of $\mathbf{A}$. Here $\mathbf{S}_m \in \mathbb{R}^{m \times n}$ denotes a LESS sketching matrix with independent rows $\frac{1}{\sqrt{m}}\mathbf{x}^\top$, $\mathbf{x}^\top = \mathbf{y}^\top \cdot \mathrm{diag}(\boldsymbol{\xi})$ where $\mathbf{y}$ consists of $\pm 1$ Rademacher entries and $\boldsymbol{\xi}$ is an $(s, \beta_1, \beta_2)$-approximate leverage score sparsifier. Note that $\mathbb{E}[\mathbf{x}\mathbf{x}^\top] = \mathbf{I}_n$. We assume that the sketching matrix $\mathbf{S}_m$ consists of $m \geq 10d$ i.i.d rows and 3 divides $m$. Also, we assume that the $\mathbf{S}_m$ satisfies the subspace embedding condition for $\mathbf{U}$ (Theorem 2) with $\eta = \frac{1}{2}$. Let $\mathbf{Q} = (\gamma \mathbf{U}^\top \mathbf{S}_m^\top \mathbf{S}_m \mathbf{U})^{-1}$ where $\gamma = \frac{m}{m-d}$.

**Theorem 5** (Small inversion bias for $(s, \beta_1, \beta_2)$-LESS embeddings)**.** *Let $\delta > 0$ satisfy $\delta < \frac{1}{m^4}$ and $m \geq O(d)$. Let $\mathbf{S}_m \in \mathbb{R}^{m \times n}$ be an $(s, \beta_1, \beta_2)$-LESS embedding for data matrix $\mathbf{U} \in \mathbb{R}^{n \times d}$ such that $\mathbf{U}^\top \mathbf{U} = \mathbf{I}$. Then there exists an event $\mathcal{E}$ with $\Pr(\mathcal{E}) \geq 1 - \delta$ such that,*

$$\frac{1}{1+\epsilon} \cdot \mathbf{I} \preceq \mathbb{E}_\mathcal{E}[\mathbf{Q}] \preceq (1+\epsilon) \cdot \mathbf{I} \quad and \quad \frac{1}{2}\mathbf{I} \preceq \mathbf{Q} \preceq 2\mathbf{I} \quad when \ conditioned \ on \ \mathcal{E}$$

*where $\epsilon = O\left(\frac{\sqrt{d}\log^{4.5}(n/\delta)}{m}\left(1 + \sqrt{\frac{d}{s}}\right)\right)$.*

*Proof.* Let $\mathbf{S}_{-i}$ denote $\mathbf{S}_m$ without the $i^{th}$ row, and $\mathbf{S}_{-ij}$ denote $\mathbf{S}_m$ with the $i^{th}$ and $j^{th}$ rows removed. Let $\mathbf{Q}_{-i} = (\gamma \mathbf{U}^\top \mathbf{S}_{-i}^\top \mathbf{S}_{-i}\mathbf{U})^{-1}$ and $\mathbf{Q}_{-ij} = (\gamma \mathbf{U}^\top \mathbf{S}_{-ij}^\top \mathbf{S}_{-ij}\mathbf{U})^{-1}$. We proceed with the same proof strategy as adopted in [18]. We define the events $\mathcal{E}_j$ as follows:

$$\mathcal{E}_j = \frac{3}{m}\mathbf{U}^\top \left(\sum_{i=t(j-1)+1}^{tj} \mathbf{x}_i \mathbf{x}_i^\top\right)\mathbf{U} \succeq \frac{1}{2}\mathbf{I}, \ j = 1, 2, 3 \quad, \mathcal{E} = \wedge_{j=1}^3 \mathcal{E}_j.$$

Note that event $\mathcal{E}_j$ means that the sketching matrix with just $(1/3)^{rd}$ rows (scaled to maintain unbiasedness of the sketch) from $\mathbf{S}_m$ satisfies a lower spectral approximation of $\mathbf{U}^\top \mathbf{U} = \mathbf{I}$. Also we notice that events $\mathcal{E}_1, \mathcal{E}_2, \mathcal{E}_3$ are independent, and for any pair $(i, j)$ there exists at least one event $\mathcal{E}_k$, $k \in \{1, 2, 3\}$ such that $\mathcal{E}_k$ is independent of both $\mathbf{x}_i$ and $\mathbf{x}_j$. Furthermore conditioned on $\mathcal{E}_k$ we have

$$\mathbf{Q}_{-i} \preceq 6 \cdot \mathbf{I}_d \ \text{ and } \ \mathbf{Q}_{-ij} \preceq 6 \cdot \mathbf{I}_d.$$

Note that as guaranteed in Theorem 2, we have $\Pr(\mathcal{E}_k) \geq 1 - \delta'$ for all $k$ and therefore $\Pr(\mathcal{E}) \geq 1 - \delta$ with $\delta' = \delta/3$. Let $\mathbb{E}_\mathcal{E}$ denote the expectation conditioned on the event $\mathcal{E}$.

$$\begin{aligned}
\mathbf{I} - \mathbb{E}_\mathcal{E}[\mathbf{Q}] &= -\mathbb{E}_\mathcal{E}[\mathbf{Q}] + \gamma \mathbb{E}_\mathcal{E}[\mathbf{Q}\mathbf{U}^\top \mathbf{S}_m^\top \mathbf{S}_m \mathbf{U}] \\
&= -\mathbb{E}_\mathcal{E}[\mathbf{Q}] + \gamma \mathbb{E}_\mathcal{E}[\mathbf{Q}\mathbf{U}^\top \mathbf{x}_i \mathbf{x}_i^\top \mathbf{U}] \\
&= -\mathbb{E}_\mathcal{E}[\mathbf{Q}] + \gamma \mathbb{E}_\mathcal{E}\Big[\frac{\mathbf{Q}_{-i}\mathbf{U}^\top \mathbf{x}_i \mathbf{x}_i^\top \mathbf{U}}{1 + \frac{\gamma}{m}\mathbf{x}_i^\top \mathbf{U}\mathbf{Q}_{-i}\mathbf{U}^\top \mathbf{x}_i}\Big] \\
&= \underbrace{\mathbb{E}_\mathcal{E}[\mathbf{Q}_{-i}\mathbf{U}^\top(\mathbf{x}_i \mathbf{x}_i^\top - \mathbf{I})\mathbf{U}]}_{\mathbf{Z}_0} + \underbrace{\mathbb{E}_\mathcal{E}[\mathbf{Q}_{-i} - \mathbf{Q}]}_{\mathbf{Z}_1} + \underbrace{\mathbb{E}_\mathcal{E}\Big[\big(\frac{\gamma}{\gamma_i} - 1\big)\mathbf{Q}_{-i}\mathbf{U}^\top \mathbf{x}_i \mathbf{x}_i^\top \mathbf{U}\Big]}_{\mathbf{Z}_2}
\end{aligned}$$

where $\gamma_i = 1 + \frac{\gamma}{m}\mathbf{x}_i^\top \mathbf{U}\mathbf{Q}_{-i}\mathbf{U}^\top \mathbf{x}_i$. The second equality follows by noting that $\mathbf{S}_m^\top \mathbf{S}_m = \frac{1}{m}\sum_{i=1}^m \mathbf{x}_i \mathbf{x}_i^\top$ and using linearity of expectation. The third equality holds due to the application of Sherman Morrison's (Lemma 5) formula on $\mathbf{Q} = (\gamma \mathbf{U}^\top \mathbf{S}_{-i}^\top \mathbf{S}_{-i}\mathbf{U} + \frac{\gamma}{m}\mathbf{U}^\top \mathbf{x}_i \mathbf{x}_i^\top \mathbf{U})^{-1}$. We start by upper bounding $\mathbf{Z}_0$ and use the following from [18].

**Lemma 11** (Upper bound on $\|\mathbf{Z}_0\|$). *For any $k > 0$ we have,*

$$\|\mathbb{E}_{\mathcal{E}}[\mathbf{Q}_{-i}\mathbf{U}^\top(\mathbf{x}_i\mathbf{x}_i^\top - \mathbf{I})\mathbf{U}]\| \leq 12\left(k\delta' + \int_k^\infty \Pr(\mathbf{x}_i^\top\mathbf{U}\mathbf{U}^\top\mathbf{x}_i \geq x)dx\right).$$

Using Chebyshev's inequality we have, $\Pr(\mathbf{x}_i^\top\mathbf{U}\mathbf{U}^\top\mathbf{x}_i \geq x) \leq \frac{\text{Var}(\mathbf{x}_i^\top\mathbf{U}\mathbf{U}^\top\mathbf{x}_i)}{x^2}$. Using Restricted Bai-Silverstein inequality for $(s, \beta_1, \beta_2)$-approximate LESS embeddings i.e., Lemma 3 (proved in Lemma 16) with $p = 2$, we have $\text{Var}(\mathbf{x}_i^\top\mathbf{U}\mathbf{U}^\top\mathbf{x}_i) \leq cd \cdot \left(1 + \frac{d\log(d/\delta)}{s}\right)$ for some absolute constant $c$. Let $k = m^2$ and $\delta' < \frac{1}{m^4}$ we get,

$$\|\mathbf{Z}_0\| = \mathcal{O}\left(\frac{1}{m^2} + \frac{d}{m^2} + \frac{d^2\log(d/\delta)}{sm^2}\right) = \mathcal{O}\left(\frac{d}{m^2} + \frac{d^2\log(d/\delta)}{sm^2}\right). \tag{4}$$

We use the bound on the term $\|\mathbf{Z}_1\|$ directly from [18], provided below as a Lemma.

**Lemma 12** (Upper bound on $\|\mathbf{Z}_1\|$, [18]).

$$\|\mathbb{E}_{\mathcal{E}'}[\mathbf{Q}_{-i} - \mathbf{Q}]\| = \mathcal{O}(1/m). \tag{5}$$

It remains to upper bound $\|\mathbf{Z}_2\|$.

$$\|\mathbf{Z}_2\| = \|\mathbb{E}_{\mathcal{E}}\left[(\frac{\gamma}{\gamma_i} - 1)\mathbf{Q}_{-i}\mathbf{U}^\top\mathbf{x}_i\mathbf{x}_i^\top\mathbf{U}\right]\| \leq \sup_{\|\mathbf{v}\|=1,\,\|\mathbf{z}\|=1} \mathbb{E}_{\mathcal{E}}[|\frac{\gamma}{\gamma_i} - 1| \cdot |\mathbf{v}^\top\mathbf{Q}_{-i}\mathbf{U}^\top\mathbf{x}_i\mathbf{x}_i^\top\mathbf{U}\mathbf{z}|].$$

Applying Hölder's inequality with $p = \mathcal{O}(\log(n))$ and $q = \frac{p}{p-1} = 1 + \Delta$ for $\Delta = \frac{1}{\mathcal{O}(\log(n))}$, we get,

$$\|\mathbf{Z}_2\| \leq \sup_{\|\mathbf{v}\|=1,\,\|\mathbf{z}\|=1} \left(\mathbb{E}_{\mathcal{E}}\left[|\frac{\gamma}{\gamma_i} - 1|^p\right]\right)^{1/p} \cdot \left(\mathbb{E}_{\mathcal{E}}\left[|\mathbf{v}^\top\mathbf{Q}_{-i}\mathbf{U}^\top\mathbf{x}_i\mathbf{x}_i^\top\mathbf{U}\mathbf{z}|^q\right]\right)^{1/q}$$

$$\leq \left(\mathbb{E}_{\mathcal{E}}\left[|\frac{\gamma}{\gamma_i} - 1|^p\right]\right)^{1/p} \cdot \underbrace{\sup_{\|\mathbf{v}\|=1}\left(\mathbb{E}_{\mathcal{E}}\left[|\mathbf{v}^\top\mathbf{Q}_{-i}\mathbf{U}^\top\mathbf{x}_i|^{2q}\right]\right)^{1/2q}}_{\mathcal{O}(1)} \cdot \underbrace{\sup_{\|\mathbf{z}\|=1}\left(\mathbb{E}_{\mathcal{E}}\left[|\mathbf{x}_i^\top\mathbf{U}\mathbf{z}|^{2q}\right]\right)^{1/2q}}_{\mathcal{O}(1)}$$

$$\tag{6}$$

where we used Cauchy-Schwarz inequality on $\mathbb{E}_{\mathcal{E}}\left[|\mathbf{v}^\top\mathbf{Q}_{-i}\mathbf{U}^\top\mathbf{x}_i\mathbf{x}_i^\top\mathbf{U}\mathbf{z}|^q\right]$. Now note that $\mathbf{x}_i = \text{diag}(\boldsymbol{\xi}) \cdot \mathbf{y}_i$, we have $\|\mathbf{x}_i\| = \text{poly}(n)$ and therefore $\|\mathbf{x}_i\|^\Delta = \mathcal{O}(1)$. We now show the terms involving exponents depending on $q$ are $\mathcal{O}(1)$ as highlighted in the inequality (6).

$$\left(\sup_{\|\mathbf{v}\|=1} \mathbb{E}_{\mathcal{E}}\left[|\mathbf{v}^\top\mathbf{Q}_{-i}\mathbf{U}^\top\mathbf{x}_i|^{2q}\right]\right)^{1/2q} \leq \left(\sup_{\|\mathbf{v}\|=1} \mathbb{E}_{\mathcal{E}}\left[|\mathbf{v}^\top\mathbf{Q}_{-i}\mathbf{U}^\top\mathbf{x}_i|^2 \cdot |\mathbf{v}^\top\mathbf{Q}_{-i}\mathbf{U}^\top\mathbf{x}_i|^{2\Delta}\right]\right)^{1/2q}$$

$$\leq \left(\sup_{\|\mathbf{v}\|=1} \mathbb{E}_{\mathcal{E}}\left[|\mathbf{v}^\top\mathbf{Q}_{-i}\mathbf{U}^\top\mathbf{x}_i|^2 \cdot \|\mathbf{v}^\top\mathbf{Q}_{-i}\mathbf{U}^\top\|^{2\Delta} \cdot \|\mathbf{x}_i\|^{2\Delta}\right]\right)^{1/2q}$$

$$\leq \mathcal{O}(1) \cdot \left(\mathbb{E}_{\mathcal{E}}\left[\|\mathbf{Q}_{-i}\mathbf{U}^\top\mathbf{x}_i\|^2 \cdot \|\mathbf{Q}_{-i}\mathbf{U}^\top\|^{2\Delta}\right]\right)^{1/2q}$$

$$\leq \mathcal{O}(1) \cdot \left(2 \cdot \mathbb{E}_{\mathcal{E}'}\left[\|\mathbf{Q}_{-i}\mathbf{U}^\top\mathbf{x}_i\|^2 \cdot \|\mathbf{Q}_{-i}\mathbf{U}^\top\|^{2\Delta}\right]\right)^{1/2q}.$$

Here $\mathcal{E}'$ is an event independent of $\mathbf{x}_i$. Without loss of generality, we can assume that $\mathcal{E}' = \mathcal{E}_1 \wedge \mathcal{E}_2$. We first condition on $\mathbf{Q}_{-i}$ and take expectation over $\mathbf{x}_i$. Also note that $\mathbf{Q}_{-i}$ and $\mathbf{x}_i$ are independent and event $\mathcal{E}'$ is independent of $\mathbf{x}_i$, and furthermore $\mathbb{E}_{\mathcal{E}'}[\mathbf{x}_i^\top\mathbf{x}_i] = \mathbb{E}[\mathbf{x}_i^\top\mathbf{x}_i] = 1$. We get,

$$\left(\sup_{\|\mathbf{v}\|=1} \mathbb{E}_{\mathcal{E}}\left[|\mathbf{v}^\top\mathbf{Q}_{-i}\mathbf{U}^\top\mathbf{x}_i|^{2q}\right]\right)^{1/2q} \leq \mathcal{O}(1) \cdot \left(\mathbb{E}_{\mathcal{E}'}[\|\mathbf{Q}_{-i}\mathbf{U}^\top\|^{2q}]\right)^{1/2q}.$$

Now we use that conditioned on $\mathcal{E}'$, $\|\mathbf{Q}_{-i}\| \leq 6$ and $\|\mathbf{U}^\top\| = 1$, we get,

$$\left(\sup_{\|\mathbf{v}\|=1} \mathbb{E}_{\mathcal{E}}\left[|\mathbf{v}^\top\mathbf{Q}_{-i}\mathbf{U}^\top\mathbf{x}_i|^{2q}\right]\right)^{1/2q} = \mathcal{O}(1).$$

Similarly, using $\|\mathbf{x}_i\|^\Delta = \mathcal{O}(1)$ and $\mathbb{E}_{\mathcal{E}'}[\mathbf{x}_i^\top \mathbf{x}_i] = \mathbb{E}[\mathbf{x}_i^\top \mathbf{x}_i] = 1$,

$$\sup_{\|\mathbf{z}\|=1} \left(\mathbb{E}_{\mathcal{E}}\left[|\mathbf{x}_i^\top \mathbf{U}\mathbf{z}|^{2q}\right]\right)^{1/2q} = \mathcal{O}(1).$$

Now we prove an upper bound on $\left(\mathbb{E}_{\mathcal{E}}\left[\left(\frac{\gamma}{\gamma_i} - 1\right)^p\right]\right)^{1/p}$. Without loss of generality, we assume that $p$ is even. We have,

$$\mathbb{E}_{\mathcal{E}}\left[\left(\frac{\gamma}{\gamma_i} - 1\right)^p\right] \leq 2 \cdot \mathbb{E}_{\mathcal{E}'}\left[\left(\frac{\gamma}{\gamma_i} - 1\right)^p\right]$$

where $\mathcal{E}'$ is event independent of $\mathbf{x}_i$. The above can be upper bounded as,

$$\left(\mathbb{E}_{\mathcal{E}'}\left[\left(\frac{\gamma}{\gamma_i} - 1\right)^p\right]\right)^{1/p} \leq \left(\mathbb{E}_{\mathcal{E}'}\left[(\gamma - \gamma_i)^p\right]\right)^{1/p} = \left(\mathbb{E}_{\mathcal{E}}\left[(\gamma - \bar{\gamma} + \bar{\gamma} - \gamma_i)^p\right]\right)^{1/p}$$

$$\leq |\gamma - \bar{\gamma}| + \left(\mathbb{E}_{\mathcal{E}'}\left[(\gamma_i - \bar{\gamma})^p\right]\right)^{1/p} \tag{7}$$

where $\bar{\gamma} = 1 + \frac{\gamma}{m}\mathbb{E}_{\mathcal{E}'}\left(\mathbf{x}_i^\top \mathbf{U}\mathbf{Q}_{-i}\mathbf{U}^\top \mathbf{x}_i\right)$. As $\mathcal{E}'$ is independent of $\mathbf{x}_i$ and $\mathbf{Q}_{-i}$ is independent of $\mathbf{x}_i$, we get $\mathbb{E}_{\mathcal{E}'}\left(\mathbf{x}_i^\top \mathbf{U}\mathbf{Q}_{-i}\mathbf{U}^\top \mathbf{x}_i\right) = \mathbb{E}_{\mathcal{E}'}\operatorname{tr}(\mathbf{Q}_{-i})$. Therefore, $\bar{\gamma} = 1 + \frac{\gamma}{m}\mathbb{E}_{\mathcal{E}'}\operatorname{tr}(\mathbf{Q}_{-i})$. We now aim to upper bound $\left(\mathbb{E}_{\mathcal{E}'}\left[(\gamma_i - \bar{\gamma})^p\right]\right)^{1/p}$ as,

$$\left(\mathbb{E}_{\mathcal{E}'}\left[(\gamma_i - \bar{\gamma})^p\right]\right)^{1/p} \leq \left(\frac{\gamma}{m}\right) \cdot \left[\left(\mathbb{E}_{\mathcal{E}'}\left[(\operatorname{tr}(\mathbf{Q}_{-i}) - \mathbf{x}_i^\top \mathbf{U}\mathbf{Q}_{-i}\mathbf{U}^\top \mathbf{x}_i)^p\right]\right)^{1/p} + \left(\mathbb{E}_{\mathcal{E}'}\left[(\operatorname{tr}(\mathbf{Q}_{-i}) - \mathbb{E}_{\mathcal{E}'}\operatorname{tr}(\mathbf{Q}_{-i}))^p\right]\right)^{1/p}\right].$$

Using our new Restricted Bai-Silverstein inequality from Lemma 3 (restated as Lemma 16 and proven in Appendix D), we have

$$\left(\mathbb{E}_{\mathcal{E}'}\left[(\operatorname{tr}(\mathbf{Q}_{-i}) - \mathbf{x}_i^\top \mathbf{U}\mathbf{Q}_{-i}\mathbf{U}^\top \mathbf{x}_i)^p\right]\right)^{1/p} < c \cdot p^3\sqrt{d} \cdot \left(1 + \sqrt{\frac{dp\log(d/\delta)}{s}}\right).$$

We now consider $\left(\mathbb{E}_{\mathcal{E}'}\left[\operatorname{tr}(\mathbf{Q}_{-i}) - \mathbb{E}_{\mathcal{E}'}\operatorname{tr}(\mathbf{Q}_{-i})\right]^p\right)^{1/p}$. In Lemma 4 (restated below as Lemma 13), we show that $|\operatorname{tr}(\mathbf{Q}_{-i}) - \mathbb{E}_{\mathcal{E}'}\operatorname{tr}(\mathbf{Q}_{-i})| \leq \frac{c'\gamma}{\sqrt{m}} \cdot d\log^{4.5}(m/\delta)$ with probability at least $1 - \delta$. Conditioned on this high-probability event we have,

$$\left(\mathbb{E}_{\mathcal{E}'}\left[\operatorname{tr}(\mathbf{Q}_{-i}) - \mathbb{E}_{\mathcal{E}'}\operatorname{tr}(\mathbf{Q}_{-i})\right]^p\right)^{1/p} \leq \frac{c'\gamma}{\sqrt{m}} \cdot d\log^{4.5}(m/\delta)$$

for an absolute constant $c' > 0$. Therefore we get with probability at least $1 - \delta$,

$$\left(\mathbb{E}_{\mathcal{E}'}\left[(\gamma_i - \bar{\gamma})^p\right]\right)^{1/p} \leq \frac{\gamma}{m}\left[c \cdot p^3\sqrt{d} \cdot \left(1 + \sqrt{\frac{dp\log(d/\delta)}{s}}\right) + \frac{c'\gamma}{\sqrt{m}} \cdot d\log^{4.5}(m/\delta)\right].$$

As $m > d$, we get $\left(\mathbb{E}_{\mathcal{E}'}\left[(\gamma_i - \bar{\gamma})^p\right]\right)^{1/p} = \mathcal{O}\left(\frac{\sqrt{d}\log^{4.5}(n/\delta)}{m} \cdot \left(1 + \sqrt{\frac{d}{s}}\right)\right)$. Also using the analysis in [18] for upper bounding $|\gamma - \bar{\gamma}|$, we get a matching upper bound on $|\gamma - \bar{\gamma}|$ as follows:

$$|\gamma - \bar{\gamma}| = \mathcal{O}\left(\frac{\sqrt{d}\log^{4.5}(n/\delta)}{m} \cdot \left(1 + \sqrt{\frac{d}{s}}\right)\right).$$

Substituting these bounds in (7) and then in (6) we get,

$$\|\mathbf{Z}_2\| = \mathcal{O}\left(\frac{\sqrt{d}\log^{4.5}(n/\delta)}{m} \cdot \left(1 + \sqrt{\frac{d}{s}}\right)\right). \tag{8}$$

Combining the upper bounds for $\mathbf{Z}_0, \mathbf{Z}_1$ and $\mathbf{Z}_2$ using relations (4,5,8) we conclude our proof. $\qquad\square$

We now provide the proof of Lemma 4, which we restate in the following Lemma.

**Lemma 13.** *For given $\delta > 0$ and matrix $\mathbf{Q}_{-i}$ we have with probability $1 - \delta$:*

$$|\mathrm{tr}(\mathbf{Q}_{-i}) - \mathbb{E}_{\mathcal{E}'}\mathrm{tr}(\mathbf{Q}_{-i})| \leq \frac{c'\gamma}{\sqrt{m}} \cdot d\log^{4.5}(m/\delta)$$

*for an absolute constant $c' > 0$.*

*Proof.* Writing $\mathrm{tr}(\mathbf{Q}_{-i}) - \mathbb{E}_{\mathcal{E}'}\mathrm{tr}(\mathbf{Q}_{-i})$ as a finite sum, we have,

$$\mathrm{tr}(\mathbf{Q}_{-i}) - \mathbb{E}_{\mathcal{E}'}\mathrm{tr}(\mathbf{Q}_{-i}) = \sum_{j=1}^{m} \mathbb{E}_{\mathcal{E}',j}\mathrm{tr}(\mathbf{Q}_{-i}) - \mathbb{E}_{\mathcal{E}',j-1}\mathrm{tr}(\mathbf{Q}_{-i}).$$

Denoting $X_j = \mathbb{E}_{\mathcal{E}',j}\mathrm{tr}(\mathbf{Q}_{-i})$ with $X_0 = \mathbb{E}_{\mathcal{E}'}\mathrm{tr}(\mathbf{Q}_{-i})$, we have the following formulation

$$\mathrm{tr}(\mathbf{Q}_{-i}) - \mathbb{E}_{\mathcal{E}'}\mathrm{tr}(\mathbf{Q}_{-i}) = \sum_{j=1}^{m} X_j - X_{j-1}$$

with $\mathbb{E}_{\mathcal{E}',j-1}[X_j] = X_{j-1}$. The random sequence $X_j$ forms a martingale and $\mathrm{tr}(\mathbf{Q}_{-i}) - \mathbb{E}_{\mathcal{E}'}\mathrm{tr}(\mathbf{Q}_{-i}) = X_m - X_0$. We find an upper bound on $|X_j - X_{j-1}|$. To achieve that we note,

$$X_j - X_{j-1} = \mathbb{E}_{\mathcal{E}',j}\mathrm{tr}(\mathbf{Q}_{-i}) - \mathbb{E}_{\mathcal{E}',j-1}\mathrm{tr}(\mathbf{Q}_{-i}) = -(\mathbb{E}_{\mathcal{E}',j} - \mathbb{E}_{\mathcal{E}',j-1})(\mathrm{tr}(\mathbf{Q}_{-ij} - \mathbf{Q}_{-i}) - \mathrm{tr}(\mathbf{Q}_{-ij})).$$

Therefore with $\psi_j = (\mathbb{E}_{\mathcal{E}',j} - \mathbb{E}_{\mathcal{E}',j-1})\mathrm{tr}(\mathbf{Q}_{-ij} - \mathbf{Q}_{-i})$ and $\chi_j = -(\mathbb{E}_{\mathcal{E}',j} - \mathbb{E}_{\mathcal{E}',j-1})\mathrm{tr}(\mathbf{Q}_{-ij})$, we have,

$$|X_j - X_{j-1}| \leq |\psi_j + \chi_j| \leq |\psi_j| + |\chi_j|.$$

From [18], we have $|\chi_j| \leq \frac{1}{m}$. We now prove an upper bound on $\psi_j$.

$$0 \leq \mathrm{tr}(\mathbf{Q}_{-ij}) - \mathrm{tr}(\mathbf{Q}_{-i}) = \mathrm{tr}\left( \frac{\frac{\gamma}{m}\mathbf{Q}_{-ij}\mathbf{U}^\top\mathbf{x}_j\mathbf{x}_j^\top\mathbf{U}\mathbf{Q}_{-ij}\mathbf{U}^\top\mathbf{U}}{1 + \frac{\gamma}{m}\mathbf{x}_j^\top\mathbf{U}\mathbf{Q}_{-ij}\mathbf{U}^\top\mathbf{x}_j} \right)$$

$$= \frac{\frac{\gamma}{m}\mathbf{x}_j^\top(\mathbf{U}\mathbf{Q}_{-ij}\mathbf{U}^\top)^2\mathbf{x}_j}{1 + \frac{\gamma}{m}\mathbf{x}_j^\top\mathbf{U}\mathbf{Q}_{-ij}\mathbf{U}^\top\mathbf{x}_j} \leq \frac{\gamma}{m}\mathbf{x}_j^\top\mathbf{U}\mathbf{Q}_{-ij}^2\mathbf{U}^\top\mathbf{x}_j.$$

Now look at the term $\mathbf{x}_j^\top\mathbf{U}\mathbf{Q}_{-ij}^2\mathbf{U}^\top\mathbf{x}_j$. For any $a > 0$ and any $k > 0$, by Markov's inequality we have,

$$\Pr\left(\mathbf{x}_j^\top\mathbf{U}\mathbf{Q}_{-ij}^2\mathbf{U}^\top\mathbf{x}_j \geq a\right) \leq \frac{\mathbb{E}[|\mathbf{x}_j^\top\mathbf{U}\mathbf{Q}_{-ij}^2\mathbf{U}^\top\mathbf{x}_j|^k]}{a^k}$$

$$\leq \frac{2^{k-1} \cdot \mathbb{E}[|\mathbf{x}_j^\top\mathbf{U}\mathbf{Q}_{-ij}^2\mathbf{U}^\top\mathbf{x}_j - \mathrm{tr}(\mathbf{Q}_{-ij}^2)|^k}{a^k} + \frac{2^{k-1} \cdot \mathbb{E}[(\mathrm{tr}(\mathbf{Q}_{-ij}^2))^k]}{a^k}.$$

Let $\mathcal{E}_1$ be an event independent of both $\mathbf{x}_i$ and $\mathbf{x}_j$ and have probability at least $1 - \delta'$. Therefore we have $\mathbb{E}[\cdot] \leq 2 \cdot \mathbb{E}_{\mathcal{E}_1}[\cdot]$. We get,

$$\Pr\left(\mathbf{x}_j^\top\mathbf{U}\mathbf{Q}_{-ij}^2\mathbf{U}^\top\mathbf{x}_j \geq a\right) \leq \frac{2^k \cdot \mathbb{E}_{\mathcal{E}_1}[|\mathbf{x}_j^\top\mathbf{U}\mathbf{Q}_{-ij}^2\mathbf{U}^\top\mathbf{x}_j - \mathrm{tr}(\mathbf{Q}_{-ij}^2)|^k}{a^k} + \frac{2^k \cdot \mathbb{E}_{\mathcal{E}_1}[(\mathrm{tr}(\mathbf{Q}_{-ij}^2))^k]}{a^k}. \tag{9}$$

We now upper bound both terms on the right-hand side separately. Considering the term $\mathbb{E}_{\mathcal{E}_1}[|\mathbf{x}_j^\top\mathbf{U}\mathbf{Q}_{-ij}^2\mathbf{U}^\top\mathbf{x}_j - \mathrm{tr}(\mathbf{Q}_{-ij}^2)|^k$ and using Lemma 3 we get,

$$\mathbb{E}_{\mathcal{E}_1}[|\mathbf{x}_j^\top\mathbf{U}\mathbf{Q}_{-ij}^2\mathbf{U}^\top\mathbf{x}_j - \mathrm{tr}(\mathbf{Q}_{-ij}^2)|^k] \leq c^k \cdot k^{3k} \cdot \left( \frac{d^2 k\log(d/\delta)}{s} + d \right)^{k/2}. \tag{10}$$

Now considering the second term in (9), i.e., $\mathbb{E}_{\mathcal{E}_1}[(\mathrm{tr}(\mathbf{Q}_{-ij}^2))^k]$. We use that conditioned on $\mathcal{E}_1$, we have $\mathbf{Q}_{-ij} \preceq 6\mathbf{I}_d$. Therefore,

$$\mathbb{E}_{\mathcal{E}_1}[(\mathrm{tr}(\mathbf{Q}_{-ij}^2))^k] \leq 6^k \cdot d^k. \tag{11}$$

Substituting (10) and (11) in (9),

$$\Pr\left(\mathbf{x}_j^\top \mathbf{U}\mathbf{Q}_{-ij}^2\mathbf{U}^\top \mathbf{x}_j \geq a\right) \leq \frac{2^k \cdot c^k \cdot k^{3k} \cdot \left(\frac{d^2 k \log(d/\delta)}{s} + d\right)^{k/2}}{a^k} + \frac{2^k \cdot 6^k \cdot d^k}{a^k}$$

$$\Pr\left(\mathbf{x}_j^\top \mathbf{U}\mathbf{Q}_{-ij}^2\mathbf{U}^\top \mathbf{x}_j \geq a\right) \leq \frac{2^k \cdot c^k \cdot k^{3k} \cdot d^k \cdot (k \log(d/\delta))^{k/2}}{a^k}$$

for some potentially different constant $c$. Consider $k = \left\lceil \frac{\log(m/\delta)}{\log(2)} \right\rceil$ and $a = 4 \cdot c \cdot k^3 \cdot d \cdot \sqrt{k \log(d/\delta)}$ and we have with probability at least $1 - \delta/m$,

$$\mathbf{x}_j^\top \mathbf{U}\mathbf{Q}_{-ij}^2\mathbf{U}^\top \mathbf{x}_j \leq 4 \cdot c \cdot k^3 \cdot d \cdot \sqrt{\log(kd/\delta)}.$$

This implies that for an absolute constant $c'$ we have,

$$|\mathrm{tr}(\mathbf{Q}_{-ij}) - \mathrm{tr}(\mathbf{Q}_{-i})| \leq c' \cdot \frac{\gamma}{m} \cdot d \cdot \log^{3.5}(m/\delta).$$

Therefore we now have an upper bound for $|\psi_j|$

$$|\psi_j| = |\mathbb{E}_{\mathcal{E}',j} - \mathbb{E}_{\mathcal{E}',j-1}(\mathrm{tr}(\mathbf{Q}_{-ij} - \mathbf{Q}_{-i}))| \leq 2c' \cdot \frac{\gamma}{m} \cdot d \cdot \log^{3.5}(m/\delta).$$

This means for all $j$, we have,

$$|X_j - X_{j-1}| \leq 4c' \cdot \frac{\gamma}{m} \cdot d \cdot \log^{3.5}(m/\delta)$$

with probability at least $1 - \delta$. Consider $c_j = 4c' \cdot \frac{\gamma}{m} \cdot d \cdot \log^{3.5}(m/\delta)$. Then $\sum_{j=1}^m c_j^2 = \frac{\gamma^2}{m} \cdot 16c'^2 \cdot d^2 \log^7(m/\delta)$. Applying Azuma's inequality (Lemma 8) with $\lambda = \frac{\gamma}{\sqrt{m}} \cdot 4c' \cdot d \cdot \log^{3.5}(m/\delta)$. We get with probability at least $1 - \delta$ and for potentially different absolute constant $c' > 0$:

$$|X_m - X_0| \leq \frac{c'\gamma}{\sqrt{m}} \cdot d \log^{4.5}(m/\delta).$$

This concludes our proof. $\qquad\square$

## B.1 Proof of Corollary 1

We consider the following Distributed Newton Sketch method:

$$\mathbf{x}_{t+1} = \mathbf{x}_t - \mathbf{p}_t, \qquad \mathbf{p}_t = \frac{1}{q}\sum_{i=1}^q \tilde{\mathbf{H}}_i^{-1}\mathbf{g}_t,$$

where $\mathbf{g}_t = \nabla f(\mathbf{x}_t)$ and $\tilde{\mathbf{H}}_i = \frac{m}{m-d}\mathbf{A}_t^\top \mathbf{S}_i^\top \mathbf{S}_i \mathbf{A}_t$ is a sketched estimate of the Hessian $\mathbf{H}_t = \nabla^2 f(\mathbf{x}_t)$, when expressed as the matrix product $\mathbf{H}_t = \mathbf{A}_t^\top \mathbf{A}_t$ for the appropriately chosen $n \times d$ matrix $\mathbf{A}_t$ (as can be done for any generalized linear model). This update can be viewed as an approximate Newton step with the Hessian inverse estimate $\hat{\mathbf{H}}_t^{-1} = \frac{1}{q}\sum_{i=1}^q \tilde{\mathbf{H}}_i^{-1}$. As long as each $\tilde{\mathbf{H}}_t^{-1}$ is an $(\sqrt{\epsilon}, \delta/2q)$-unbiased estimator of $\mathbf{H}_t^{-1}$, then using Lemma 34 from [18] we get that the averaged Hessian inverse with $q = \tilde{O}(1/\epsilon)$ satisfies:

$$\frac{1}{1+\tilde{\epsilon}}\mathbf{H}_t^{-1} \preceq \hat{\mathbf{H}}_t^{-1} \preceq (1+\tilde{\epsilon})\mathbf{H}_t^{-1}, \quad \tilde{\epsilon} = \sqrt{\epsilon} + \tilde{O}(1/\sqrt{q}) = O(\sqrt{\epsilon}).$$

Using standard approximate Newton analysis (e.g., see Lemmas 1 and 3, and the related discussion in [15]), this implies that when $\mathbf{x}_t$ is in a sufficiently small neighborhood around $\mathbf{x}^*$ (determined solely by the strong convexity and Lipschitz constants of the Hessian of $f$), we have:

$$\frac{f(\mathbf{x}_{t+1}) - f(\mathbf{x}^*)}{f(\mathbf{x}_t) - f(\mathbf{x}^*)} = O(\tilde{\epsilon}^2) = O(\epsilon).$$

Adjusting the constants appropriately, relying on Theorem 3 for constructing $\tilde{\mathbf{H}}_i$, and following the complexity analysis from Theorem 2, we recover the claim.

# C  Least squares bias analysis: Proof of Theorem 4

In this section, we aim to prove Theorem 4. Let $\mathbf{x}^* = \operatorname{argmin}_{\mathbf{x}} \|\mathbf{U}\mathbf{x} - \mathbf{b}\|^2$ where $\mathbf{U} \in \mathbb{R}^{n \times d}$ is the data matrix containing $n$ data points and $\mathbf{b} \in \mathbb{R}^n$ is a vector containing labels corresponding to $n$ data points. We adopt the same notations as used in the proof of Theorem 5. Let $\tilde{\mathbf{x}} = \operatorname{argmin}_{\mathbf{x}} \|\mathbf{S}_m\mathbf{U}\mathbf{x} - \mathbf{S}_m\mathbf{b}\|^2$. Furthermore for any $\mathbf{x} \in \mathbb{R}^d$ we can find the loss at $\mathbf{x}$ as $L(\mathbf{x}) = \|\mathbf{U}\mathbf{x} - \mathbf{b}\|^2$. Additionally, we use $\mathbf{r}$ to denote the residual $\mathbf{b} - \mathbf{U}\mathbf{x}^*$. We aim to provide an upper bound on the bias introduced due to this sketch and solve paradigm, i.e. $L(\mathbb{E}(\tilde{\mathbf{x}})) - L(\mathbf{x}^*)$. Similar to Theorem 4 we condition on the high probability event $\mathcal{E}$ and consider $L(\mathbb{E}_{\mathcal{E}}(\tilde{\mathbf{x}})) - L(\mathbf{x}^*)$. By Pythagorean theorem, we have $L(\mathbb{E}_{\mathcal{E}}[\tilde{\mathbf{x}}]) - L(\mathbf{x}^*) = \|\mathbf{U}(\mathbb{E}_{\mathcal{E}}[\tilde{\mathbf{x}}]) - \mathbf{U}\mathbf{x}^*\|^2$. Also,

$$
\begin{aligned}
L(\mathbb{E}_{\mathcal{E}}[\tilde{\mathbf{x}}]) - L(\mathbf{x}^*) &= \|\mathbf{U}(\mathbb{E}_{\mathcal{E}}[\tilde{\mathbf{x}}]) - \mathbf{U}\mathbf{x}^*\|^2 = \|\mathbb{E}_{\mathcal{E}}[\tilde{\mathbf{x}}] - \mathbf{x}^*\|^2 \\
&= \left\|\mathbb{E}_{\mathcal{E}}[(\mathbf{U}^\top\mathbf{S}_m^\top\mathbf{S}_m\mathbf{U})^{-1}\mathbf{U}^\top\mathbf{S}_m^\top\mathbf{S}_m\mathbf{b}] - \mathbf{U}^\top\mathbf{b}\right\|^2 \\
&= \left\|\mathbb{E}_{\mathcal{E}}[(\mathbf{U}^\top\mathbf{S}_m^\top\mathbf{S}_m\mathbf{U})^{-1}\mathbf{U}^\top\mathbf{S}_m^\top\mathbf{S}_m](\mathbf{b} - \mathbf{U}\mathbf{U}^\top\mathbf{b})\right\|^2 .
\end{aligned}
$$

Note that $\mathbf{b} - \mathbf{U}\mathbf{U}^\top\mathbf{b} = \mathbf{b} - \mathbf{U}\mathbf{x}^* = \mathbf{r}$. We get,

$$
L(\mathbb{E}_{\mathcal{E}}[\tilde{\mathbf{x}}]) - L(\mathbf{x}^*) = \left\|\mathbb{E}_{\mathcal{E}}[(\mathbf{U}^\top\mathbf{S}_m^\top\mathbf{S}_m\mathbf{U})^{-1}\mathbf{U}^\top\mathbf{S}_m^\top\mathbf{S}_m]\mathbf{r}\right\|^2 .
$$

Consider $\mathbf{Q} = (\gamma\mathbf{U}^\top\mathbf{S}_m^\top\mathbf{S}_m\mathbf{U})^{-1}$, $\mathbf{Q}_{-i} = (\gamma\mathbf{U}^\top\mathbf{S}_{-i}^\top\mathbf{S}_{-i}\mathbf{U})^{-1}$ and $\gamma = \frac{m}{m-d}$,

$$
\begin{aligned}
\mathbb{E}_{\mathcal{E}}[(\mathbf{U}^\top\mathbf{S}_m^\top\mathbf{S}_m\mathbf{U})^{-1}\mathbf{U}^\top\mathbf{S}_m^\top\mathbf{S}_m\mathbf{r}] &= \mathbb{E}_{\mathcal{E}}[\gamma(\gamma\mathbf{U}^\top\mathbf{S}_m^\top\mathbf{S}_m\mathbf{U})^{-1}\mathbf{U}^\top\mathbf{S}_m^\top\mathbf{S}_m\mathbf{r}] \\
&= \gamma\mathbb{E}_{\mathcal{E}}[\mathbf{Q}\mathbf{U}^\top\mathbf{S}_m^\top\mathbf{S}_m\mathbf{r}] \\
&= \gamma\mathbb{E}_{\mathcal{E}}[\mathbf{Q}\mathbf{U}^\top\mathbf{x}_i\mathbf{x}_i^\top\mathbf{r}]
\end{aligned}
$$

where we used linearity of expectation in the last line combined with $\mathbf{S}_m^\top\mathbf{S}_m = \frac{1}{m}\sum_{i=1}^m \mathbf{x}_i\mathbf{x}_i^\top$. Using Sherman-Morrison formula (Lemma 5) we have $\mathbf{Q}\mathbf{U}^\top\mathbf{x}_i = \frac{\mathbf{Q}_{-i}\mathbf{U}^\top\mathbf{x}_i}{1+\frac{\gamma}{m}\mathbf{x}_i^\top\mathbf{U}\mathbf{Q}_{-i}\mathbf{U}^\top\mathbf{x}_i}$. Denote $\gamma_i = 1 + \frac{\gamma}{m}\mathbf{x}_i^\top\mathbf{U}\mathbf{Q}_{-i}\mathbf{U}^\top\mathbf{x}_i$ and substitute we get,

$$
\mathbb{E}_{\mathcal{E}}[(\mathbf{U}^\top\mathbf{S}_m^\top\mathbf{S}_m\mathbf{U})^{-1}\mathbf{U}^\top\mathbf{S}_m^\top\mathbf{S}_m\mathbf{r}] = \mathbb{E}_{\mathcal{E}}\left[\left(\frac{\gamma}{\gamma_i}\right)\mathbf{Q}_{-i}\mathbf{U}^\top\mathbf{x}_i\mathbf{x}_i^\top\mathbf{r}\right]
$$

$$
\mathbb{E}_{\mathcal{E}}[(\mathbf{U}^\top\mathbf{S}_m^\top\mathbf{S}_m\mathbf{U})^{-1}\mathbf{U}^\top\mathbf{S}_m^\top\mathbf{S}_m\mathbf{r}] = \mathbb{E}_{\mathcal{E}}[\mathbf{Q}_{-i}\mathbf{U}^\top\mathbf{x}_i\mathbf{x}_i^\top\mathbf{r}] + \mathbb{E}_{\mathcal{E}}\left[\left(\frac{\gamma}{\gamma_i} - 1\right)\mathbf{Q}_{-i}\mathbf{U}^\top\mathbf{x}_i\mathbf{x}_i^\top\mathbf{r}\right].
$$

So we get the following decomposition:

$$
\begin{aligned}
L(\mathbb{E}_{\mathcal{E}}[\tilde{\mathbf{x}}]) - L(\mathbf{x}^*) &= \left\|\mathbb{E}_{\mathcal{E}}[(\mathbf{U}^\top\mathbf{S}_m^\top\mathbf{S}_m\mathbf{U})^{-1}\mathbf{U}^\top\mathbf{S}_m^\top\mathbf{S}_m]\mathbf{r}\right\|^2 \\
&= \left\|\mathbb{E}_{\mathcal{E}}[\mathbf{Q}_{-i}\mathbf{U}^\top\mathbf{x}_i\mathbf{x}_i^\top\mathbf{r}] + \mathbb{E}_{\mathcal{E}}\left[\left(\frac{\gamma}{\gamma_i} - 1\right)\mathbf{Q}_{-i}\mathbf{U}^\top\mathbf{x}_i\mathbf{x}_i^\top\mathbf{r}\right]\right\|^2 \\
&\leq 2\underbrace{\|\mathbb{E}_{\mathcal{E}}[\mathbf{Q}_{-i}\mathbf{U}^\top\mathbf{x}_i\mathbf{x}_i^\top\mathbf{r}]\|^2}_{\|\mathbf{Z}_0\mathbf{r}\|^2} + 2\underbrace{\left\|\mathbb{E}_{\mathcal{E}}\left[\left(\frac{\gamma}{\gamma_i} - 1\right)\mathbf{Q}_{-i}\mathbf{U}^\top\mathbf{x}_i\mathbf{x}_i^\top\mathbf{r}\right]\right\|^2}_{\|\mathbf{Z}_2\mathbf{r}\|^2}. \quad (12)
\end{aligned}
$$

Note that a similar decomposition was considered in the proof of Theorem 5 (see Appendix B) with slightly different $\mathbf{Z}_0$ and $\mathbf{Z}_2$. We first bound $\|\mathbf{Z}_0\mathbf{r}\|^2$ in the following argument. Without loss of generality, we assume that events $\mathcal{E}_1$ and $\mathcal{E}_2$ are independent of $\mathbf{x}_i$ and $\mathcal{E}' = \mathcal{E}_1 \wedge \mathcal{E}_2$.

$$
\begin{aligned}
\mathbf{Z}_0\mathbf{r} = \mathbb{E}_{\mathcal{E}}[\mathbf{Q}_{-i}\mathbf{U}^\top\mathbf{x}_i\mathbf{x}_i^\top\mathbf{r}] &= \frac{\mathbb{E}[\mathbf{Q}_{-i}\mathbf{U}^\top\mathbf{x}_i\mathbf{x}_i^\top\mathbf{r} \cdot \mathbf{1}_{\mathcal{E}}]}{\Pr(\mathcal{E})} \\
&= \frac{\mathbb{E}[\mathbf{Q}_{-i}\mathbf{U}^\top\mathbf{x}_i\mathbf{x}_i^\top\mathbf{r} \cdot \mathbf{1}_{\mathcal{E}_1} \cdot \mathbf{1}_{\mathcal{E}_2} \cdot (1 - \mathbf{1}_{\neg\mathcal{E}_3})]}{\Pr(\mathcal{E})} \\
&= \frac{\mathbb{E}[\mathbf{Q}_{-i}\mathbf{U}^\top\mathbf{x}_i\mathbf{x}_i^\top\mathbf{r} \cdot \mathbf{1}_{\mathcal{E}_1} \cdot \mathbf{1}_{\mathcal{E}_2}]}{\Pr(\mathcal{E})} - \frac{\mathbb{E}[\mathbf{Q}_{-i}\mathbf{U}^\top\mathbf{x}_i\mathbf{x}_i^\top\mathbf{r} \cdot \mathbf{1}_{\mathcal{E}_1} \cdot \mathbf{1}_{\mathcal{E}_2} \cdot \mathbf{1}_{\neg\mathcal{E}_3}]}{\Pr(\mathcal{E})} \\
&= \frac{1}{1 - \delta'}\left(\mathbb{E}_{\mathcal{E}'}[\mathbf{Q}_{-i}\mathbf{U}^\top\mathbf{x}_i\mathbf{x}_i^\top\mathbf{r}] - \mathbb{E}_{\mathcal{E}'}[\mathbf{Q}_{-i}\mathbf{U}^\top\mathbf{x}_i\mathbf{x}_i^\top\mathbf{r} \cdot \mathbf{1}_{\neg\mathcal{E}_3}]\right).
\end{aligned}
$$

Now note that in the first term $\mathbf{Q}_{-i}$ is independent of $\mathbf{x}_i$ and $\mathbf{x}_i$ is also independent of the event $\mathcal{E}'$. Using this with the fact that $\mathbb{E}[\mathbf{x}_i\mathbf{x}_i^\top] = \mathbf{I}$ we get $\mathbb{E}_{\mathcal{E}'}[\mathbf{Q}_{-i}\mathbf{U}^\top\mathbf{x}_i\mathbf{x}_i^\top\mathbf{r}] = \mathbb{E}_{\mathcal{E}'}[\mathbf{Q}_{-i}\mathbf{U}^\top\mathbf{r}] = 0$, since $\mathbf{U}^\top\mathbf{r} = 0$. Therefore,

$$\|\mathbf{Z}_0\mathbf{r}\|^2 \leq 2\left\|\mathbb{E}_{\mathcal{E}'}[\mathbf{Q}_{-i}\mathbf{U}^\top\mathbf{x}_i\mathbf{x}_i^\top\mathbf{r}\cdot\mathbf{1}_{\neg\mathcal{E}_3}]\right\|^2$$
$$\leq 72\|\mathbb{E}_{\mathcal{E}'}[\mathbf{U}^\top\mathbf{x}_i\mathbf{x}_i^\top\mathbf{r}\cdot\mathbf{1}_{\neg\mathcal{E}_3}]\|^2.$$

The last inequality holds because conditioned on $\mathcal{E}'$, we know that $\|\mathbf{Q}_{-i}\| \leq 6$. Using Cauchy-Schwarz inequality we have,

$$\|\mathbf{Z}_0\mathbf{r}\|^2 \leq 72\left(\sqrt{\mathbb{E}_{\mathcal{E}'}[\mathbf{x}_i^\top\mathbf{U}\mathbf{U}^\top\mathbf{x}_i\cdot\mathbf{1}_{\neg\mathcal{E}_3}]}\cdot\sqrt{\mathbb{E}_{\mathcal{E}'}[\mathbf{r}^\top\mathbf{x}_i\mathbf{x}_i^\top\mathbf{r}]}\right)^2.$$

Note that $\mathbb{E}_{\mathcal{E}'}[\mathbf{r}^\top\mathbf{x}_i\mathbf{x}_i^\top\mathbf{r}] = \|\mathbf{r}\|^2$. Also, we can bound $\mathbb{E}_{\mathcal{E}'}[\mathbf{x}_i^\top\mathbf{U}\mathbf{U}^\top\mathbf{x}_i\cdot\mathbf{1}_{\neg\mathcal{E}_3}]$ as following:

$$\mathbb{E}_{\mathcal{E}'}[\mathbf{x}_i^\top\mathbf{U}\mathbf{U}^\top\mathbf{x}_i\cdot\mathbf{1}_{\neg\mathcal{E}_3}] = \int_0^\infty \Pr(\mathbf{x}_i^\top\mathbf{U}\mathbf{U}^\top\mathbf{x}_i\cdot\mathbf{1}_{\neg\mathcal{E}_3} > x)dx$$
$$= \int_0^y \Pr(\mathbf{x}_i^\top\mathbf{U}\mathbf{U}^\top\mathbf{x}_i\cdot\mathbf{1}_{\neg\mathcal{E}_3} > x)dx + \int_y^\infty \Pr(\mathbf{x}_i^\top\mathbf{U}\mathbf{U}^\top\mathbf{x}_i\cdot\mathbf{1}_{\neg\mathcal{E}_3} > x)dx$$
$$\leq y\delta' + \int_y^\infty \Pr(\mathbf{x}_i^\top\mathbf{U}\mathbf{U}^\top\mathbf{x}_i > x)dx.$$

Using Chebyshev's inequality, $\Pr(\mathbf{x}_i^\top\mathbf{U}\mathbf{U}^\top\mathbf{x}_i \geq x) \leq \frac{\mathrm{Var}(\mathbf{x}_i^\top\mathbf{U}\mathbf{U}^\top\mathbf{x}_i)}{x^2}$. By Restricted Bai-Silverstein, Lemma 3 with $p = 2$, we have $\mathrm{Var}(\mathbf{x}_i^\top\mathbf{U}\mathbf{U}^\top\mathbf{x}_i) \leq c\cdot\left(d + \frac{d^2\log(d/\delta)}{s}\right)$ for some absolute constant $c$. Let $y = m^2$ and $\delta' < \frac{1}{m^4}$ we get,

$$\mathbb{E}[\mathbf{x}_i^\top\mathbf{U}\mathbf{U}^\top\mathbf{x}_i\cdot\mathbf{1}_{\neg\mathcal{E}_3}] = \mathcal{O}\left(\frac{1}{m^2} + \frac{d}{m^2} + \frac{d^2\log(d/\delta)}{sm^2}\right) = \mathcal{O}\left(\frac{d}{m^2} + \frac{d^2\log(d/\delta)}{sm^2}\right).$$

This finishes upper bounding $\|\mathbf{Z}_0\mathbf{r}\|^2$ as:

$$\|\mathbf{Z}_0\mathbf{r}\|^2 = \mathcal{O}\left(\frac{d}{m^2} + \frac{d^2\log(d/\delta)}{sm^2}\right)\cdot\|\mathbf{r}\|^2. \tag{13}$$

Now we proceed with $\|\mathbf{Z}_2\mathbf{r}\|^2$,

$$\|\mathbf{Z}_2\mathbf{r}\| = \left\|\mathbb{E}_{\mathcal{E}}\left[\left(\frac{\gamma}{\gamma_i} - 1\right)\mathbf{Q}_{-i}\mathbf{U}^\top\mathbf{x}_i\mathbf{x}_i^\top\mathbf{r}\right]\right\|.$$

Applying Hölder's inequality with $p = \mathcal{O}(\log(n))$ and $q = 1 + \Delta$ where $\Delta = \frac{1}{\mathcal{O}(\log(n))}$, we get,

$$\|\mathbf{Z}_2\mathbf{r}\| \leq \left(\mathbb{E}_{\mathcal{E}}\left|\frac{\gamma}{\gamma_i} - 1\right|^p\right)^{1/p}\cdot\left(\sup_{\|\mathbf{v}\|=1}\mathbb{E}_{\mathcal{E}}[\mathbf{v}^\top\mathbf{Q}_{-i}\mathbf{U}^\top\mathbf{x}_i\mathbf{x}_i^\top\mathbf{r}]^q\right)^{1/q}$$
$$\leq \left(\mathbb{E}_{\mathcal{E}}\left|\frac{\gamma}{\gamma_i} - 1\right|^p\right)^{1/p}\cdot\left(\mathbb{E}_{\mathcal{E}}\|\mathbf{Q}_{-i}\mathbf{U}^\top\mathbf{x}_i\|^{2q}\right)^{1/2q}\cdot\left(\mathbb{E}_{\mathcal{E}}\|\mathbf{x}_i^\top\mathbf{r}\|^{2q}\right)^{1/2q}$$
$$\leq \left(\mathbb{E}_{\mathcal{E}}\left|\frac{\gamma}{\gamma_i} - 1\right|^p\right)^{1/p}\cdot\left(2\cdot\mathbb{E}_{\mathcal{E}'}[\|\mathbf{Q}_{-i}\mathbf{U}^\top\mathbf{x}_i\|^2\cdot\|\mathbf{Q}_{-i}\mathbf{U}^\top\mathbf{x}_i\|^{2\Delta}]\right)^{1/2q}\cdot\left(2\cdot\mathbb{E}_{\mathcal{E}'}[\|\mathbf{x}_i^\top\mathbf{r}\|^2\cdot\|\mathbf{x}_i^\top\mathbf{r}\|^{2\Delta}]\right)^{1/2q}.$$

Since $\|\mathbf{x}_i\| = \mathrm{poly}(n)$, we have $\|\mathbf{x}_i\|^{2\Delta} = \mathcal{O}(1)$ and therefore we have $\|\mathbf{Q}_{-i}\mathbf{U}^\top\mathbf{x}_i\|^{2\Delta} = \mathcal{O}(1)\|\mathbf{Q}_{-i}\mathbf{U}^\top\|^{2\Delta}$. Also $\mathbb{E}_{\mathcal{E}'}\|\mathbf{Q}_{-i}\mathbf{U}^\top\mathbf{x}_i\|^2 = \mathbb{E}_{\mathcal{E}'}\|\mathbf{Q}_{-i}\|^2$. Using that conditioned on $\mathcal{E}'$ we have $\|\mathbf{Q}_{-i}\| \leq 6$, we get $\left(\mathbb{E}_{\mathcal{E}}\|\mathbf{Q}_{-i}\mathbf{U}^\top\mathbf{x}_i\|^{2q}\right)^{1/2q} = \mathcal{O}(1)$. Similarly using $\|\mathbf{x}_i\|^{2\Delta} = \mathcal{O}(1)$ we get $\left(\mathbb{E}_{\mathcal{E}}\|\mathbf{x}_i^\top\mathbf{r}\|^{2q}\right)^{1/2q} = \mathcal{O}(1)\|\mathbf{r}\|$. This gives us:

$$\|\mathbf{Z}_2\mathbf{r}\| \leq \mathcal{O}(1)\cdot\left(\mathbb{E}_{\mathcal{E}}\left|\frac{\gamma}{\gamma_i} - 1\right|^p\right)^{1/p}\cdot\|\mathbf{r}\|.$$

Now using (8) from the proof of Theorem 5, we get,

$$\left(\mathbb{E}_{\mathcal{E}}\left|\frac{\gamma}{\gamma_i}-1\right|^p\right)^{1/p} = \mathcal{O}\left(\frac{\sqrt{d}\log^{4.5}(n/\delta)}{m}\cdot\left(1+\sqrt{\frac{d}{s}}\right)\right).$$

Finally the bound for $\|\mathbf{Z}_2\mathbf{r}\|^2$ follows as:

$$\|\mathbf{Z}_2\mathbf{r}\|^2 = \mathcal{O}\left(\frac{d\log^9(n/\delta)}{m^2}\cdot\left(1+\frac{d}{s}\right)\right)\cdot\|\mathbf{r}\|^2. \tag{14}$$

Combining (13) and (14) we conclude our proof.

## D  Higher-Moment Restricted Bai-Silvestein (Lemma 3)

In this section, we prove Lemma 3. We need the following two auxiliary lemmas to derive the main theorem of this section. The first lemma uses Matrix-Chernoff concentration inequality to upper bound the spectral norm of $\mathbf{U}_{\boldsymbol{\xi}}^{\top}\mathbf{U}_{\boldsymbol{\xi}}$ where $\boldsymbol{\xi}$ is a $(s,\beta_1,\beta_2)$-approximate leverage score sparsifier.

**Lemma 14** (Spectral norm bound with leverage score sparsifier). *Let* $\mathbf{U}\in\mathbb{R}^{n\times d}$ *has orthonormal columns. Let* $\boldsymbol{\xi}$ *be a* $(s,\beta_1,\beta_2)$-*approximate leverage score sparsifier for* $\mathbf{U}$*, and denote* $\mathbf{U}_{\boldsymbol{\xi}} = \mathrm{diag}(\boldsymbol{\xi})\mathbf{U}$*. Then for any* $\delta > 0$ *we have,*

$$\Pr\left(\left\|\mathbf{U}_{\boldsymbol{\xi}}^{\top}\mathbf{U}_{\boldsymbol{\xi}}\right\| \geq \left(1+\frac{3d\log(d/\delta)}{s}\right)\right) \leq \delta \quad \text{if } s < d,$$
$$\Pr\left(\left\|\mathbf{U}_{\boldsymbol{\xi}}^{\top}\mathbf{U}_{\boldsymbol{\xi}}\right\| \geq (1+3\log(d/\delta))\right) \leq \delta \quad \text{if } s \geq d.$$

*Proof.* Writing $\mathbf{U}_{\boldsymbol{\xi}}^{\top}\mathbf{U}_{\boldsymbol{\xi}}$ as a sum of matrices we have,

$$\mathbf{U}_{\boldsymbol{\xi}}^{\top}\mathbf{U}_{\boldsymbol{\xi}} = \sum_{i=1}^{n}\xi_i^2\mathbf{u}_i\mathbf{u}_i^{\top} = \sum_{i=1}^{n}\frac{b_i}{p_i}\mathbf{u}_i\mathbf{u}_i^{\top} = \sum_{i=1}^{n}\mathbf{Z}_i$$

where $p_i = \min\{1,\frac{s\beta_1\tilde{l}_i}{d}\}$ and $\mathbf{Z}_i = \frac{b_i}{p_i}\mathbf{u}_i\mathbf{u}_i^{\top}$. Note that $\mathbf{Z}_i's$ are independent random variables and $\mathbb{E}[\mathbf{Z}_i] = \mathbf{u}_i\mathbf{u}_i^{\top}$. Also $\sum_{i=1}^{n}\mathbb{E}\mathbf{Z}_i = \mathbf{U}^{\top}\mathbf{U} = \mathbf{I}_d$. If $p_i = 1$ then $\mathbf{Z}_i = \mathbf{u}_i\mathbf{u}_i^{\top}$ and therefore $\|\mathbf{Z}_i\| = \|\mathbf{u}_i\|^2 \leq 1$. If $p_i < 1$, we have $\|\mathbf{Z}_i\| \leq \frac{1}{p_i}\|\mathbf{u}_i\|^2 = \frac{d}{s\beta_1\tilde{l}_i}\cdot l_i$. As $l_i \leq \beta_1\tilde{l}_i$, we get $\|\mathbf{Z}_i\| \leq \frac{d}{s}$. Therefore $\|\mathbf{Z}_i\| \leq \max\{1,\frac{d}{s}\}$ for all $i$. Denote $R = \max\{1,\frac{d}{s}\}$. We use Matrix Chernoff (Lemma 7) to upper bound the largest eigenvalue of $\mathbf{U}_{\boldsymbol{\xi}}^{\top}\mathbf{U}_{\boldsymbol{\xi}}$. For any $\epsilon > 0$, we have,

$$\Pr\left(\lambda_{\max}\left(\sum_{i=1}^{n}\mathbf{Z}_i\right) \geq (1+\epsilon)\right) \leq d\cdot\exp\left(-\frac{\epsilon^2}{(2+\epsilon)R}\right).$$

With $R = \max\{1,\frac{d}{s}\}$ and depending on the case whether $s \leq d$ or $s > d$ we get

$$\Pr\left(\lambda_{\max}\left(\sum_{i=1}^{n}\mathbf{Z}_i\right) \geq \left(1+\frac{3d\log(d/\delta)}{s}\right)\right) \leq \delta \quad \text{if } s \leq d,$$
$$\Pr\left(\lambda_{\max}\left(\sum_{i=1}^{n}\mathbf{Z}_i\right) \geq (1+3\log(d/\delta))\right) \leq \delta \text{ if } s > d.$$

$\square$

Let $\mathcal{A}_{\boldsymbol{\xi}}$ denote the event $\left\|\mathbf{U}_{\boldsymbol{\xi}}^{\top}\mathbf{U}_{\boldsymbol{\xi}}\right\| \leq 1+\frac{3d\log(d/\delta)}{s}$, holding with probability at least $1-\delta$, for small $\delta > 0$. In the next result we upper bound the higher moments of the trace of $\mathbf{U}_{\boldsymbol{\xi}}\mathbf{C}\mathbf{U}_{\boldsymbol{\xi}}^{\top}$ for any matrix $\mathbf{C} \preceq \mathcal{O}(1)\cdot\mathbf{I}$. We first prove the upper bound in the case when the high probability event $\mathcal{A}_{\boldsymbol{\xi}}$ does not occur.

**Lemma 15** (Trace moment bound over small probability event)**.** *Let $k \in \mathbb{N}$ be fixed. Let $\mathbf{U} \in \mathbb{R}^{n \times d}$ have orthonormal columns. Let $\boldsymbol{\xi}$ be a $(s, \beta_1, \beta_2)$-approximate leverage score sparsifier for $\mathbf{U}$. Let $\mathbf{U}_{\boldsymbol{\xi}} = \mathrm{diag}(\boldsymbol{\xi})\mathbf{U}$. Also let $\Pr(\mathcal{A}_{\boldsymbol{\xi}}) \geq 1 - \frac{1}{(12d)^{4k}}$ and event $\mathcal{E}'$ be independent of the sparsifier $\boldsymbol{\xi}$. Then we have,*

$$\mathbb{E}_{\mathcal{E}'}\left[(\mathrm{tr}(\mathbf{U}_{\boldsymbol{\xi}}\mathbf{C}\mathbf{U}_{\boldsymbol{\xi}}^\top\mathbf{U}_{\boldsymbol{\xi}}\mathbf{C}\mathbf{U}_{\boldsymbol{\xi}}^\top))^k | \neg\mathcal{A}_{\boldsymbol{\xi}}\right] \leq (4k)^{4k}$$

*for any fixed matrix $\mathbf{C}$ such that $0 \preceq \mathbf{C} \preceq 6\mathbf{I}$.*

*Proof.*

$$\mathbb{E}_{\mathcal{E}'}\left[(\mathrm{tr}(\mathbf{U}_{\boldsymbol{\xi}}\mathbf{C}\mathbf{U}_{\boldsymbol{\xi}}^\top\mathbf{U}_{\boldsymbol{\xi}}\mathbf{C}\mathbf{U}_{\boldsymbol{\xi}}^\top))^k | \neg\mathcal{A}_{\boldsymbol{\xi}}\right] = \int_0^\infty \Pr((\mathrm{tr}(\mathbf{U}_{\boldsymbol{\xi}}\mathbf{C}\mathbf{U}_{\boldsymbol{\xi}}^\top\mathbf{U}_{\boldsymbol{\xi}}\mathbf{C}\mathbf{U}_{\boldsymbol{\xi}}^\top))^k \cdot \mathbf{1}_{\neg\mathcal{A}_{\boldsymbol{\xi}}} \geq x | \mathcal{E}')dx$$

$$= \int_0^\infty \Pr((\mathrm{tr}(\mathbf{U}_{\boldsymbol{\xi}}\mathbf{C}\mathbf{U}_{\boldsymbol{\xi}}^\top\mathbf{U}_{\boldsymbol{\xi}}\mathbf{C}\mathbf{U}_{\boldsymbol{\xi}}^\top))^k \cdot \mathbf{1}_{\neg\mathcal{A}_{\boldsymbol{\xi}}} \geq x)dx.$$

The last equality holds because $\mathcal{E}'$ is independent of $\boldsymbol{\xi}$. Consider some fixed $y > 0$.

$$\mathbb{E}_{\mathcal{E}'}\left[(\mathrm{tr}(\mathbf{U}_{\boldsymbol{\xi}}\mathbf{C}\mathbf{U}_{\boldsymbol{\xi}}^\top\mathbf{U}_{\boldsymbol{\xi}}\mathbf{C}\mathbf{U}_{\boldsymbol{\xi}}^\top))^k | \neg\mathcal{A}_{\boldsymbol{\xi}}\right] \leq \int_0^y \Pr((\mathrm{tr}(\mathbf{U}_{\boldsymbol{\xi}}\mathbf{C}\mathbf{U}_{\boldsymbol{\xi}}^\top))^{2k} \cdot \mathbf{1}_{\neg\mathcal{A}_{\boldsymbol{\xi}}} \geq x)dx$$

$$+ \int_y^\infty \Pr((\mathrm{tr}(\mathbf{U}_{\boldsymbol{\xi}}\mathbf{C}\mathbf{U}_{\boldsymbol{\xi}}^\top))^{2k} \cdot \mathbf{1}_{\neg\mathcal{A}_{\boldsymbol{\xi}}} \geq x)dx$$

$$\leq y \cdot \delta + \mathbb{E}[(\mathrm{tr}(\mathbf{U}_{\boldsymbol{\xi}}\mathbf{C}\mathbf{U}_{\boldsymbol{\xi}}^\top))^{4k}] \cdot \int_y^\infty \frac{1}{x^2}dx. \qquad (15)$$

The last inequality holds because by Chebyshev's inequality $\Pr((\mathrm{tr}(\mathbf{U}_{\boldsymbol{\xi}}\mathbf{C}\mathbf{U}_{\boldsymbol{\xi}}^\top))^{2k} \geq x) \leq \frac{\mathbb{E}[(\mathrm{tr}(\mathbf{U}_{\boldsymbol{\xi}}\mathbf{C}\mathbf{U}_{\boldsymbol{\xi}}^\top))^{4k}]}{x^2}$. Also note that,

$$(\mathrm{tr}(\mathbf{U}_{\boldsymbol{\xi}}\mathbf{C}\mathbf{U}_{\boldsymbol{\xi}}^\top))^{4k} = \left(\sum_{i=1}^n \xi_i^2 \mathbf{u}_i^\top\mathbf{C}\mathbf{u}_i\right)^{4k} = \left(\sum_{i=1}^n \frac{b_i}{p_i}\mathbf{u}_i^\top\mathbf{C}\mathbf{u}_i\right)^{4k}.$$

For $1 \leq i \leq n$, let $R_i$ be random variables denoting $\frac{b_i}{p_i}\mathbf{u}_i^\top\mathbf{C}\mathbf{u}_i$. Then note that $R_i$ are independent random variables with $\mathbb{E}[R_i] = \mathbf{u}_i^\top\mathbf{C}\mathbf{u}_i$. Also $R_i$ are non-negative random variables with finite $(4k)^{th}$ moment. Using Rosenthal's inequality (Lemma 9) we get,

$$\mathbb{E}\left[\left(\sum_{i=1}^n \frac{b_i}{p_i}\mathbf{u}_i^\top\mathbf{C}\mathbf{u}_i\right)^{4k}\right] \leq 2^{4k} \cdot (4k)^{4k} \cdot \left[\sum_{i=1}^n \mathbb{E}[R_i^{4k}] + \left(\sum_{i=1}^n \mathbb{E}[R_i]\right)^{4k}\right].$$

Now $\sum_{i=1}^n \mathbb{E}[R_i] = \mathrm{tr}(\mathbf{U}\mathbf{C}\mathbf{U}^\top)$ and $\mathbb{E}[R_i^{4k}]$ can be found as follows: if $p_i = 1$ then $R_i = \mathbf{u}_i^\top\mathbf{C}\mathbf{u}_i$ and therefore $R_i^{4k} = (\mathbf{u}_i^\top\mathbf{C}\mathbf{u}_i)^{4k} \leq \|\mathbf{u}_i\|^{2(4k-1)}\mathbf{u}_i^\top\mathbf{C}^{4k}\mathbf{u}_i \leq \mathbf{u}_i^\top\mathbf{C}^{4k}\mathbf{u}_i$, if $p_i < 1$, we have,

$$\mathbb{E}[R_i^{4k}] = p_i \cdot \frac{1}{p_i^{4k}}(\mathbf{u}_i^\top\mathbf{C}\mathbf{u}_i)^{4k} = \frac{d^{4k-1}}{s^{4k-1}} \cdot \frac{1}{(\beta_1\tilde{l}_i)^{4k-1}}(\mathbf{u}_i^\top\mathbf{C}\mathbf{u}_i)^{4k}$$

$$\leq \frac{d^{4k-1}}{s^{4k-1}} \cdot \frac{1}{(\beta_1\tilde{l}_i)^{4k-1}} \cdot \|\mathbf{u}_i\|^{2(4k-1)}\mathbf{u}_i^\top\mathbf{C}^{4k}\mathbf{u}_i$$

$$= \frac{d^{4k-1}}{s^{4k-1}} \cdot \frac{1}{(\beta_1\tilde{l}_i)^{4k-1}} \cdot l_i^{4k-1}\mathbf{u}_i^\top\mathbf{C}^{4k}\mathbf{u}_i.$$

Now using $l_i \leq \beta_1\tilde{l}_i$ we get,

$$\mathbb{E}[R_i^{4k}] \leq \frac{d^{4k-1}}{s^{4k-1}} \cdot \mathbf{u}_i^\top\mathbf{C}^{4k}\mathbf{u}_i.$$

Therefore,

$$\mathbb{E}\left[\left(\sum_{i=1}^n \frac{b_i}{p_i}\mathbf{u}_i^\top\mathbf{C}\mathbf{u}_i\right)^{4k}\right] \leq 2^{4k} \cdot (4k)^{4k} \cdot \left[\max\left(1, \frac{d^{4k-1}}{s^{4k-1}}\right) \cdot \mathrm{tr}(\mathbf{C}^{4k}) + (\mathrm{tr}(\mathbf{C}))^{4k}\right].$$

Using the above inequality along with using $\mathbf{C} \preceq 6\mathbf{I}$ we get,

$$\mathbb{E}\left[\left(\sum_{t=1}^{s} \frac{1}{sp_{i_t}} \mathbf{u}_{i_t}^\top \mathbf{C} \mathbf{u}_{i_t}\right)^{4k}\right] \leq 2^{4k} \cdot (4k)^{4k} \cdot 6^{4k} \cdot d^{4k} = (12)^{4k} \cdot (4k)^{4k} \cdot d^{4k}.$$

Substituting the above bound in (15), it follows that:

$$\mathbb{E}_{\mathcal{E}'}\left[(\operatorname{tr}(\mathbf{U}_{\boldsymbol{\xi}} \mathbf{C} \mathbf{U}_{\boldsymbol{\xi}}^\top \mathbf{U}_{\boldsymbol{\xi}} \mathbf{C} \mathbf{U}_{\boldsymbol{\xi}}^\top))^k | \neg \mathcal{A}_{\boldsymbol{\xi}}\right] \leq y \cdot \delta + (12)^{4k} \cdot (4k)^{4k} \cdot d^{4k} \cdot \frac{1}{y}.$$

For $y > (12d)^{4k}$ and $\delta < \frac{1}{y}$, we get the desired result. $\qquad\square$

We are now ready to prove the main result of this section. The following result, which is central to our analysis, upper bounds the high moments of a deviation of a quadratic form from its mean.

**Lemma 16** (Restricted Bai-Silverstein for $(s, \beta_1, \beta_2)$-LESS embedding)**.** *Let $p \in \mathbb{N}$ be fixed and $\mathbf{U} \in \mathbb{R}^{n \times d}$ have orthonormal columns. Let $\mathbf{x}_i = \operatorname{diag}(\boldsymbol{\xi})\mathbf{y}_i$ where $\mathbf{y}_i \in \mathbb{R}^n$ has independent $\pm 1$ entries and $\boldsymbol{\xi}$ is a $(s, \beta_1, \beta_2)$-approximate leverage score sparsifier for $\mathbf{U}$. Let $\mathbf{U}_{\boldsymbol{\xi}} = \operatorname{diag}(\boldsymbol{\xi})\mathbf{U}$. Then for any matrix $0 \preceq \mathbf{C} \preceq 6\mathbf{I}$ and any $\delta > 0$ we have,*

$$\left(\mathbb{E}\left[\operatorname{tr}(\mathbf{C}) - \mathbf{x}_i^\top \mathbf{U} \mathbf{C} \mathbf{U}^\top \mathbf{x}_i\right]^p\right)^{1/p} < c \cdot p^3 \sqrt{d} \cdot \left(1 + \sqrt{\frac{dp \log(d/\delta)}{s}}\right)$$

*for an absolute constant $c > 0$.*

*Proof.* Let $\mathbf{x}_i = \operatorname{diag}(\boldsymbol{\xi})\mathbf{y}_i$ where $\mathbf{y}_i$ is vector of Rademacher $\pm 1$ entries. Denote $\mathbf{U}_{\boldsymbol{\xi}} = \operatorname{diag}(\boldsymbol{\xi})\mathbf{U}$.

$$\mathbb{E}\left[(\operatorname{tr}(\mathbf{C}) - \mathbf{x}_i^\top \mathbf{U} \mathbf{C} \mathbf{U}^\top \mathbf{x}_i)^p\right] = \mathbb{E}\left[(\operatorname{tr}(\mathbf{C}) - \mathbf{y}_i^\top \mathbf{U}_{\boldsymbol{\xi}} \mathbf{C} \mathbf{U}_{\boldsymbol{\xi}}^\top \mathbf{y}_i)^p\right]$$

$$= \mathbb{E}\left[(\operatorname{tr}(\mathbf{C}) - \operatorname{tr}(\mathbf{U}_{\boldsymbol{\xi}} \mathbf{C} \mathbf{U}_{\boldsymbol{\xi}}^\top) + \operatorname{tr}(\mathbf{U}_{\boldsymbol{\xi}} \mathbf{C} \mathbf{U}_{\boldsymbol{\xi}}^\top) - \mathbf{y}_i^\top \mathbf{U}_{\boldsymbol{\xi}} \mathbf{C} \mathbf{U}_{\boldsymbol{\xi}}^\top \mathbf{y}_i)^p\right]$$

$$\leq 2^{p-1}\left(\underbrace{\mathbb{E}\left[\operatorname{tr}(\mathbf{C}) - \operatorname{tr}(\mathbf{U}_{\boldsymbol{\xi}} \mathbf{C} \mathbf{U}_{\boldsymbol{\xi}}^\top)\right]^p}_{T_1} + \underbrace{\mathbb{E}\left[\operatorname{tr}(\mathbf{U}_{\boldsymbol{\xi}} \mathbf{C} \mathbf{U}_{\boldsymbol{\xi}}^\top) - \mathbf{y}_i^\top \mathbf{U}_{\boldsymbol{\xi}} \mathbf{C} \mathbf{U}_{\boldsymbol{\xi}}^\top \mathbf{y}_i\right]^p}_{T_2}\right).$$

(16)

First, consider $T_1$, substitute $\xi_i^2 = \frac{b_i}{p_i}$, and assume exponent $p$ to be even.

$$\mathbb{E}\left[\operatorname{tr}(\mathbf{C}) - \operatorname{tr}(\mathbf{U}_{\boldsymbol{\xi}} \mathbf{C} \mathbf{U}_{\boldsymbol{\xi}}^\top)\right]^p = \mathbb{E}\left[\operatorname{tr}(\mathbf{U} \mathbf{C} \mathbf{U}^\top) - \operatorname{tr}(\mathbf{U}_{\boldsymbol{\xi}} \mathbf{C} \mathbf{U}_{\boldsymbol{\xi}}^\top)\right]^p$$

$$= \mathbb{E}\left[\left(\sum_{i=1}^{n} (\xi_i^2 - 1)\mathbf{u}_i^\top \mathbf{C} \mathbf{u}_i\right)^p\right]$$

$$= \mathbb{E}\left[\left(\sum_{i=1}^{n} (\frac{b_i}{p_i} - 1)\mathbf{u}_i^\top \mathbf{C} \mathbf{u}_i\right)^p\right].$$

For $1 \leq i \leq n$ consider random variables $R_i$ where $R_i = \frac{b_i}{p_i} \mathbf{u}_i^\top \mathbf{C} \mathbf{u}_i$. Furthermore let $Y_i = R_i - \mathbf{u}_i^\top \mathbf{C} \mathbf{u}_i$. Then $\sum_{i=1}^{n} Y_i = \sum_{i=1}^{n} (\frac{b_i}{p_i} - 1)\mathbf{u}_i^\top \mathbf{C} \mathbf{u}_i$ and $\mathbb{E}[Y_i] = 0$. $Y_i$ are independent mean zero random variables with finite $p^{th}$ moments and therefore we can use Rosenthal's inequality (for symmetric random variables, Lemma 9) to get,

$$\mathbb{E}\left[\sum_{i=1}^{n} Y_i\right]^p < A(p)\left(\underbrace{\sum_{i=1}^{n} \mathbb{E}[Y_i]^p}_{T_1^1} + \underbrace{\left(\sum_{i=1}^{n} \mathbb{E}[Y_i]^2\right)^{p/2}}_{T_1^2}\right)$$

(17)

where $A(p)$ is a constant depending on $p$. We bound $T_1^1$ and $T_1^2$ separately, starting with $T_1^1$ as,

$$T_1^1 = \sum_{i=1}^{n} \mathbb{E}[Y_i]^p.$$

Recall that $Y_i = R_i - \mathbf{u}_i^\top \mathbf{C} \mathbf{u}_i$. $R_i$ is always non-negative because $\mathbf{C}$ is a positive semi-definite matrix. Therefore we have,

$$\mathbb{E}(Y_i)^p \leq \mathbb{E}(R_i)^p.$$

We find the $p^{th}$ moment of $R_i$, $\mathbb{E}(R_i)^p = (\mathbf{u}_i^\top \mathbf{C} \mathbf{u}_i)^p$ if $p_i = 1$. If $p_i < 1$ then,

$$\mathbb{E}(R_i)^p = p_i^{p-1}(\mathbf{u}_i^\top \mathbf{C} \mathbf{u}_i)^p = \frac{d^{p-1}}{s^{p-1}} \cdot \frac{1}{(\beta_1 \tilde{l}_i)^{p-1}}(\mathbf{u}_i^\top \mathbf{C} \mathbf{u}_i)^p \leq \frac{d^{p-1}}{s^{p-1}} \cdot \frac{1}{(\beta_1 \tilde{l}_i)^{p-1}} \cdot \|\mathbf{u}_i\|^{2(p-1)} \mathbf{u}_i^\top \mathbf{C}^p \mathbf{u}_i$$

$$\leq \frac{d^{p-1}}{s^{p-1}} \mathbf{u}_i^\top \mathbf{C}^p \mathbf{u}_i$$

where in the last inequality we use $\|\mathbf{u}_i\|^{2(p-1)} = l_i^{p-1} \leq (\beta_1 \tilde{l}_i)^{p-1}$. Summing over $i$ from 1 to $n$ we get an upper bound for $T_1^1$ as the following:

$$\sum_{i=1}^n \mathbb{E}(R_i)^p \leq \sum_{i=1}^n \mathbb{E}(R_i)^p \leq \max\left(1, \frac{d^{p-1}}{s^{p-1}}\right) \cdot \operatorname{tr}(\mathbf{U}\mathbf{C}^p\mathbf{U}^\top). \tag{18}$$

Now we upper bound $T_1^2$. Note that $\mathbb{E}(Y_i)^2 = \mathbb{E}[R_i]^2 = (\mathbf{u}_i^\top \mathbf{C} \mathbf{u}_i)^2$ if $p_i = 1$ and if $p_i < 1$ we have $\mathbb{E}[R_i]^2 = \frac{1}{p_i} \cdot (\mathbf{u}_i^\top \mathbf{C} \mathbf{u}_i)^2 \leq \frac{d}{s} \cdot \frac{1}{\beta_1 \tilde{l}_i} \|\mathbf{u}_i\|^2 \mathbf{u}_i^\top \mathbf{C}^2 \mathbf{u}_i \leq \frac{d}{s} \mathbf{u}_i^\top \mathbf{C}^2 \mathbf{u}_i$. Summing over $i$ from 1 to $n$ we get an upper bound for $T_1^2$ as,

$$\left(\sum_{i=1}^n \mathbb{E}(Y_i)^2\right)^{p/2} \leq \left(\max\left(1, \frac{d}{s}\right) \operatorname{tr}(\mathbf{U}\mathbf{C}^2\mathbf{U}^\top)\right)^{p/2}. \tag{19}$$

Substituting (18) and (19) in (17) we have,

$$\mathbb{E}\left[\operatorname{tr}(\mathbf{C}) - \operatorname{tr}(\mathbf{U}_{\boldsymbol{\xi}}\mathbf{C}\mathbf{U}_{\boldsymbol{\xi}}^\top)\right]^p \leq A(p) \cdot \left(\max\left(1, \frac{d^{p-1}}{s^{p-1}}\right) \cdot \operatorname{tr}(\mathbf{U}\mathbf{C}^p\mathbf{U}^\top) + \left(\max\left(1, \frac{d}{s}\right) \operatorname{tr}(\mathbf{U}\mathbf{C}^2\mathbf{U}^\top)\right)^{p/2}\right). \tag{20}$$

Now we aim to upper bound the term $T_2$ in (16) i.e., $\mathbb{E}\left[\operatorname{tr}(\mathbf{U}_{\boldsymbol{\xi}}\mathbf{C}\mathbf{U}_{\boldsymbol{\xi}}^\top) - \mathbf{y}_i^\top \mathbf{U}_{\boldsymbol{\xi}}\mathbf{C}\mathbf{U}_{\boldsymbol{\xi}}^\top \mathbf{y}_i\right]^p$. First, we condition over $\boldsymbol{\xi}$ and take expectation over $\mathbf{y}_i$. This requires using standard Bai-Silverstein inequality (Lemma 10), we get,

$$\mathbb{E}\left[\operatorname{tr}(\mathbf{U}_{\boldsymbol{\xi}}\mathbf{C}\mathbf{U}_{\boldsymbol{\xi}}^\top) - \mathbf{y}_i^\top \mathbf{U}_{\boldsymbol{\xi}}\mathbf{C}\mathbf{U}_{\boldsymbol{\xi}}^\top \mathbf{y}_i\right]^p \leq B(p) \cdot \mathbb{E}\left[\left(\nu_4 \operatorname{tr}(\mathbf{U}_{\boldsymbol{\xi}}\mathbf{C}\mathbf{U}_{\boldsymbol{\xi}}^\top \mathbf{U}_{\boldsymbol{\xi}}\mathbf{C}\mathbf{U}_{\boldsymbol{\xi}}^\top)\right)^{p/2} + \nu_{2p} \operatorname{tr}(\mathbf{U}_{\boldsymbol{\xi}}\mathbf{C}\mathbf{U}_{\boldsymbol{\xi}}^\top \mathbf{U}_{\boldsymbol{\xi}}\mathbf{C}\mathbf{U}_{\boldsymbol{\xi}}^\top)^{p/2}\right]$$

where $B(p)$ is a constant depending on $p$. Since $\mathbf{y}_i$ consists of $\pm 1$ entries we have $\nu_4, \nu_{2p} \leq 1$. Also using $\operatorname{tr}(\mathbf{A}\mathbf{B}) \leq \operatorname{tr}(\mathbf{A})\operatorname{tr}(\mathbf{B})$ and considering the high probability event $\mathcal{A}_{\boldsymbol{\xi}}$ capturing $\left\|\mathbf{U}_{\boldsymbol{\xi}}^\top \mathbf{U}_{\boldsymbol{\xi}}\right\| \leq \left(1 + \max\left(\frac{3d\log(d/\delta)}{s}, 3\log(d/\delta)\right)\right)$. We get the following,

$$\mathbb{E}\left[\operatorname{tr}(\mathbf{U}_{\boldsymbol{\xi}}\mathbf{C}\mathbf{U}_{\boldsymbol{\xi}}^\top) - \mathbf{y}_i^\top \mathbf{U}_{\boldsymbol{\xi}}\mathbf{C}\mathbf{U}_{\boldsymbol{\xi}}^\top \mathbf{y}_i\right]^p \leq 2B(p) \cdot \mathbb{E}\left(\operatorname{tr}(\mathbf{U}_{\boldsymbol{\xi}}\mathbf{C}\mathbf{U}_{\boldsymbol{\xi}}^\top \mathbf{U}_{\boldsymbol{\xi}}\mathbf{C}\mathbf{U}_{\boldsymbol{\xi}}^\top)\right)^{p/2}$$

$$= 2B(p) \cdot \left[\mathbb{E}\left(\operatorname{tr}(\mathbf{U}_{\boldsymbol{\xi}}\mathbf{C}\mathbf{U}_{\boldsymbol{\xi}}^\top \mathbf{U}_{\boldsymbol{\xi}}\mathbf{C}\mathbf{U}_{\boldsymbol{\xi}}^\top) \cdot \mathbf{1}_{\mathcal{A}_{\boldsymbol{\xi}}}\right)^{p/2}\right] + 2B(p) \cdot \left[\mathbb{E}\left(\operatorname{tr}(\mathbf{U}_{\boldsymbol{\xi}}\mathbf{C}\mathbf{U}_{\boldsymbol{\xi}}^\top \mathbf{U}_{\boldsymbol{\xi}}\mathbf{C}\mathbf{U}_{\boldsymbol{\xi}}^\top) \cdot \mathbf{1}_{\neg\mathcal{A}_{\boldsymbol{\xi}}}\right)^{p/2}\right]$$

$$\leq 2B(p) \cdot \left(1 + \max\left(\frac{3d\log(d/\delta)}{s}, 3\log(d/\delta)\right)\right)^{p/2} \mathbb{E}\left(\operatorname{tr}(\mathbf{U}_{\boldsymbol{\xi}}\mathbf{C}^2\mathbf{U}_{\boldsymbol{\xi}}^\top)\right)^{p/2} + 2B(p) \cdot (2p)^{2p}. \tag{21}$$

The first term in the last inequality follows from the Matrix-Chernoff (Lemma 7) and the second term follows from Lemma 15 by considering $k = p/2$ (assuming that $\delta$ is small enough so that Lemma 15 is satisfied). We now upper-bound $\mathbb{E}\left(\operatorname{tr}(\mathbf{U}_{\boldsymbol{\xi}}\mathbf{C}^2\mathbf{U}_{\boldsymbol{\xi}}^\top)\right)^{p/2}$ using Rosenthal's inequality for uncentered (non-symmetric) random variables,

$$\operatorname{tr}(\mathbf{U}_{\boldsymbol{\xi}}\mathbf{C}^2\mathbf{U}_{\boldsymbol{\xi}}^\top) = \sum_{i=1}^n \xi_i^2 \mathbf{u}_i^\top \mathbf{C}^2 \mathbf{u}_i = \sum_{i=1}^n \frac{b_i}{p_i} \mathbf{u}_i^\top \mathbf{C}^2 \mathbf{u}_i.$$

For $1 \leq i \leq n$ consider independent random variables $R'_i = \frac{b_i}{p_i} \mathbf{u}_i^\top \mathbf{C}^2 \mathbf{u}_i$. We have,

$$\sum_{i=1}^{n} R'_i = \operatorname{tr}(\mathbf{U}_{\boldsymbol{\xi}} \mathbf{C}^2 \mathbf{U}_{\boldsymbol{\xi}}^\top).$$

Here $R'_i$ are positive random variables with finite $(p/2)^{th}$ moment. Using Rosenthal's inequality we get,

$$\mathbb{E} \left( \sum_{i=1}^{n} R'_i \right)^{p/2} \leq C(p/2) \cdot \left[ \underbrace{\sum_{i=1}^{n} \mathbb{E}(R'_i)^{p/2}}_{T_2^1} + \underbrace{\left( \sum_{i=1}^{n} \mathbb{E} R'_i \right)^{p/2}}_{T_2^2} \right]. \tag{22}$$

It is straightforward to upper bound $\mathbb{E}(R'_i)^{p/2}$ as,

$$\mathbb{E}(R'_i)^{p/2} = (\mathbf{u}_i^\top \mathbf{C}^2 \mathbf{u}_i)^{p/2} \ \text{if} \ p_i = 1,$$
$$\leq \frac{d^{\frac{p}{2}-1}}{s^{\frac{p}{2}-1}} \mathbf{u}_i^\top \mathbf{C}^p \mathbf{u}_i \ \text{if} \ p_i < 1.$$

Summing over $i$ from 1 to $n$ we get upper bound for $T_2^1$ as,

$$\sum_{i=1}^{n} \mathbb{E}(R'_n)^{p/2} \leq \max\left(1, \frac{d^{\frac{p}{2}-1}}{s^{\frac{p}{2}-1}}\right) \operatorname{tr}(\mathbf{U}\mathbf{C}^p\mathbf{U}^\top). \tag{23}$$

Now we consider $T_2^2$. It is simply given as,

$$\left( \sum_{i=1}^{n} \mathbb{E} R'_j \right)^{p/2} = \left( \operatorname{tr}(\mathbf{U}\mathbf{C}^2\mathbf{U}^\top) \right)^{p/2}. \tag{24}$$

Combining (23) and (24) and substituting in (22) we get,

$$\mathbb{E} \left( \operatorname{tr}(\mathbf{U}_{\boldsymbol{\xi}} \mathbf{C}^2 \mathbf{U}_{\boldsymbol{\xi}}^\top) \right)^{p/2} \leq C(p/2) \cdot \left( \max\left(1, \frac{d^{\frac{p}{2}-1}}{s^{\frac{p}{2}-1}}\right) \operatorname{tr}(\mathbf{U}\mathbf{C}^p\mathbf{U}^\top) + \left( \operatorname{tr}(\mathbf{U}\mathbf{C}^2\mathbf{U}^\top) \right)^{p/2} \right). \tag{25}$$

Substituting (25) in (21) and let $w = \max(1, d/s)$,

$$\mathbb{E}\left[ \operatorname{tr}(\mathbf{U}_{\boldsymbol{\xi}} \mathbf{C} \mathbf{U}_{\boldsymbol{\xi}}^\top) - \mathbf{y}_i^\top \mathbf{U}_{\boldsymbol{\xi}} \mathbf{C} \mathbf{U}_{\boldsymbol{\xi}}^\top \mathbf{y}_i \right]^p \leq 2B(p) \cdot (2p)^{2p}$$
$$+ 2B(p)C(p/2) \cdot (1 + 3w \log(d/\delta))^{p/2} \cdot \left[ w^{\frac{p}{2}-1} \operatorname{tr}(\mathbf{U}\mathbf{C}^p\mathbf{U}^\top) + \left( \operatorname{tr}(\mathbf{U}\mathbf{C}^2\mathbf{U}^\top) \right)^{p/2} \right]. \tag{26}$$

Combining the bounds for $T_1$ (20) and $T_2$ (26) substituting in (16), and noting that $\operatorname{tr}(\mathbf{U}\mathbf{C}^k\mathbf{U}^\top) = \operatorname{tr}(\mathbf{C}^k)$ for any $k$, we get,

$$\mathbb{E}\left[ (\operatorname{tr}(\mathbf{C}) - \mathbf{x}_i^\top \mathbf{U}\mathbf{C}\mathbf{U}^\top \mathbf{x}_i)^p \right] \leq 2^{p-1} A(p) \cdot \left( w^{p-1} \cdot \operatorname{tr}(\mathbf{C}^p) + \left( w \cdot \operatorname{tr}(\mathbf{C}^2) \right)^{p/2} \right)$$
$$+ 2^p B(p)C(p/2) \cdot (1 + 3w \log(d/\delta))^{p/2} \cdot \left[ w^{\frac{p}{2}-1} \operatorname{tr}(\mathbf{C}^p) + \left( \operatorname{tr}(\mathbf{C}^2) \right)^{p/2} \right]$$
$$+ 2^p B(p)(2p)^{2p}.$$

Now we specify the various constants depending on $p$. We have $A(p) \leq (2p)^p, B(p) \leq (2p)^p, C(p/2) \leq p^{p/2}$. Also we use $\operatorname{tr}(\mathbf{C}^k) \leq 6^k \cdot d$ since $\mathbf{C} \preceq 6\mathbf{I}$. This implies for an absolute constant $c$ we have,

$$\left( \mathbb{E}\left[ \operatorname{tr}(\mathbf{C}) - \mathbf{x}_i^\top \mathbf{U}\mathbf{C}\mathbf{U}^\top \mathbf{x}_i \right]^p \right)^{1/p} \leq c \cdot p^3 \sqrt{d} \cdot \left( 1 + \sqrt{\frac{dp \log(d/\delta)}{s}} \right)$$

where $\delta > 0$ is now arbitrary. $\qquad \square$

# E  Application to distributed settings with partitioned data

In this section, we extend our main result to settings where the dataset is partitioned across multiple machines. We illustrate this in the distributed setting considered in [43]. Let $\mathbf{A} \in \mathbb{R}^{N \times d}$ and $\mathbf{b} \in \mathbb{R}^N$ with $\text{rank}(\mathbf{A}) = d$. In practice, we randomly shuffle the rows of $\mathbf{A}$ and $\mathbf{b}$, and partition the data uniformly across $q$ machines, with every machine getting chunks of size $n$ denoted by $(\mathbf{A}_i, \mathbf{b}_i)$. In this distributed setup, $\mathbf{A}_i$ can be considered a uniformly subsampled sketch of $\mathbf{A}$ with sketch size $n$. However, similar to the analysis in [43], in the following analysis we assume that each machine constructs its sketch $(\mathbf{A}_i, \mathbf{b}_i)$ by uniformly sampling rows from $\mathbf{A}$ and $\mathbf{b}$ with replacement and independently from other machines. This setting is not ideally a partition but we believe similar results can also be shown for the setting where we partition the dataset after reshuffling the rows. Let $\mu = N \max_{i=1}^n \ell_i(\mathbf{A})$ denote the matrix coherence of $\mathbf{A}$. Let $\mathbf{S} \in \mathbb{R}^{n \times N}$ be a uniform sampling matrix with size $n$, where we will assume that $\mathbf{S}$ also scales the samples by $\sqrt{N/n}$, which implies that $\mathbb{E}[\mathbf{S}^\top \mathbf{S}] = \mathbf{I}$. Then the following holds:

**Lemma 17** (**Subspace embedding using uniform subsampling; based on Theorem 12 in [43]**). *Let $\mathbf{S} \in \mathbb{R}^{n \times N}$ be a uniform sampling matrix of size $n$ where $n \geq O(\mu \log(d/\delta))$. Then with probability $1 - \delta$ we have,*

$$\frac{1}{1+\eta} \cdot \mathbf{A}^\top \mathbf{A} \preceq \mathbf{A}^\top \mathbf{S}^\top \mathbf{S} \mathbf{A} \preceq (1+\eta) \cdot \mathbf{A}^\top \mathbf{A}$$

*for $\eta = O(\sqrt{\mu \log(d/\delta)/n})$.*

Let $\mathbf{x}^* = \mathbf{A}^\dagger \mathbf{b}$ denote the minimizer of least squares loss $\|\mathbf{A}\mathbf{x} - \mathbf{b}\|^2$ and $\mathbf{x}_i^* = \mathbf{A}_i^\dagger \mathbf{b}_i$ denote the solution to the sketched least squares problem at $i^{th}$ machine. Here $\mathbf{A}_i = \mathbf{S}_i \mathbf{A}$ and $\mathbf{b}_i = \mathbf{S}_i \mathbf{b}$, where $\mathbf{S}_i$ denotes the uniform subsampling matrix for $i^{th}$ machine. We can use the model averaging result from [43] to prove the following result.

**Lemma 18** (**Model averaging result; adapted from Theorem 20 in [43]**). *Let there be $q$ machines and each machine constructs its sketch $(\mathbf{A}_i, \mathbf{b}_i)$ independently by uniformly sampling $n$ rows from $\mathbf{A}$ and $\mathbf{b}$, where $n = O(\mu \log(dq/\delta))$. Then with probability $0.99$,*

$$\left\| \mathbf{A}\bar{\mathbf{x}}^* - \mathbf{b} \right\|^2 \leq \left( 1 + c\Big( \frac{\mu \log(d/\delta)}{qn} + \frac{\mu^2 \log^2(d/\delta)}{n^2} \Big) \right) \cdot \|\mathbf{A}\mathbf{x}^* - \mathbf{b}\|^2. \tag{27}$$

*where $\bar{\mathbf{x}}^* = \frac{1}{q} \sum_{i=1}^q \mathbf{x}_i^*$ and $c > 0$ is an absolute constant[2].*

Since computing $\mathbf{x}_i^*$ at any machine could be potentially expensive due to large $n$, we instead find $\tilde{\mathbf{x}}_i$ at $i^{th}$ machine using the sketching procedure from Theorem 2. Note that this requires applying our debiasing techniques to the least squares problem $\min \|\mathbf{A}_i \mathbf{x} - \mathbf{b}_i\|^2$ at every machine. To that end, let $\bar{\mathbf{x}} = \frac{1}{q} \sum_{i=1}^q \tilde{\mathbf{x}}_i$ and consider $\mathbb{E}_{\mathcal{P}} \|\mathbf{A}\bar{\mathbf{x}} - \mathbf{b}\|^2$, where $\mathbb{E}_{\mathcal{P}}$ means conditional expectation given the sketches $(\mathbf{A}_i, \mathbf{b}_i)$ at every machine. The following theorem states that if the bias from partitioning is much smaller than the sketching error in each machine, then our approach can be successfully used to reduce the bias of the sketching estimate so that averaging will work as desired.

**Theorem 6.** *Let there be $q$ machines and each machine constructs its sketch $(\mathbf{A}_i, \mathbf{b}_i)$ independently by uniformly sampling $n$ rows from $\mathbf{A}$ and $\mathbf{b}$, where $n \geq O(\mu \log(dq/\delta))$. Also, let $i^{th}$ machine construct $\tilde{\mathbf{x}}_i$ via locally sketching $\mathbf{A}_i$ and $\mathbf{b}_i$ such that $\tilde{\mathbf{x}}_i$ satisfies Theorem 1 (with $\mathbf{A}_i$ and $\mathbf{b}_i$) at $i^{th}$ machine. Then, the estimator $\bar{\mathbf{x}} = \frac{1}{q} \sum_{i=1}^q \tilde{\mathbf{x}}_i$ averaged across the machines satisfies*

$$\mathbb{E}_{\mathcal{P}} \|\mathbf{A}\bar{\mathbf{x}} - \mathbf{b}\|^2 \leq \left( 1 + c'\Big( \epsilon + \frac{1}{q} + \frac{\mu \log(d/\delta)}{qn} + \frac{\mu^2 \log^2(d/\delta)}{n^2} \Big) \right) \|\mathbf{A}\mathbf{x}^* - \mathbf{b}\|^2.$$

*with probability at least $0.9$ and an absolute constant $c' > 0$.*

*Proof.* By the Pythagorean theorem we have:

$$\mathbb{E}_{\mathcal{P}} \|\mathbf{A}\bar{\mathbf{x}} - \mathbf{b}\|^2 = \|\mathbf{A}\mathbf{x}^* - \mathbf{b}\|^2 + \mathbb{E}_{\mathcal{P}} \|\mathbf{A}(\bar{\mathbf{x}} - \mathbf{x}^*)\|^2. \tag{28}$$

---

[2]The result Theorem 20 [43] also requires that $\|\mathbf{S}_i^\top \mathbf{S}_i\|^2 \leq \frac{N}{n}$ for all $i$, which holds trivially if $\mathbf{S}_i$ is a sampling matrix without replacement. However, in our setting, we are implicitly assuming that with high probability $\|\mathbf{S}_i^\top \mathbf{S}_i\|^2 = O(\frac{N}{n})$ and absorb the extra resulting constant in $c$.

We proceed to upper bound $\mathbb{E}_{\mathcal{P}}\|\mathbf{A}(\bar{\mathbf{x}} - \mathbf{x}^*)\|^2$ as,

$$\mathbb{E}_{\mathcal{P}}\|\mathbf{A}(\bar{\mathbf{x}} - \mathbf{x}^*)\|^2 \leq 2\left(\mathbb{E}_{\mathcal{P}}\|\mathbf{A}(\bar{\mathbf{x}} - \bar{\mathbf{x}}^*)\|^2 + \|\mathbf{A}(\bar{\mathbf{x}}^* - \mathbf{x}^*)\|^2\right).$$

We use Lemma 18 to upper bound the second term and get,

$$\mathbb{E}_{\mathcal{P}}\|\mathbf{A}(\bar{\mathbf{x}} - \mathbf{x}^*)\|^2 \leq 2\left(\mathbb{E}_{\mathcal{P}}\|\mathbf{A}(\bar{\mathbf{x}} - \bar{\mathbf{x}}^*)\|^2 + c\Big(\frac{\mu\log(d/\delta_1)}{qn} + \frac{\mu^2\log^2(d/\delta_1)}{n^2}\Big)\|\mathbf{A}\mathbf{x}^* - \mathbf{b}\|^2\right). \tag{29}$$

On the other hand,

$$\mathbb{E}_{\mathcal{P}}\|\mathbf{A}(\bar{\mathbf{x}} - \bar{\mathbf{x}}^*)\|^2 = \mathbb{E}_{\mathcal{P}}\|\mathbf{A}(\bar{\mathbf{x}} - \mathbb{E}_{\mathcal{P}}[\bar{\mathbf{x}}] + \mathbb{E}_{\mathcal{P}}[\bar{\mathbf{x}}] - \bar{\mathbf{x}}^*\|^2$$
$$= \mathbb{E}_{\mathcal{P}}\|\mathbf{A}(\bar{\mathbf{x}} - \mathbb{E}_{\mathcal{P}}[\bar{\mathbf{x}}])\|^2 + \|\mathbf{A}(\mathbb{E}_{\mathcal{P}}[\bar{\mathbf{x}}] - \bar{\mathbf{x}}^*)\|^2. \tag{30}$$

We upper bound both terms in (30) separately. Proceeding with the first term we have,

$$\mathbb{E}_{\mathcal{P}}\|\mathbf{A}(\bar{\mathbf{x}} - \mathbb{E}_{\mathcal{P}}[\bar{\mathbf{x}}])\|^2 = \mathbb{E}_{\mathcal{P}}\Big\|\mathbf{A}\Big(\frac{1}{q}\sum_{i=1}^{q}(\tilde{\mathbf{x}}_i - \mathbb{E}_{\mathcal{P}}[\tilde{\mathbf{x}}_i])\Big)\Big\|^2$$

$$= \frac{1}{q^2} \cdot \mathbb{E}_{\mathcal{P}}\Big\|\mathbf{A}\Big(\sum_{i=1}^{q}(\tilde{\mathbf{x}}_i - \mathbb{E}_{\mathcal{P}}[\tilde{\mathbf{x}}_i])\Big)\Big\|^2$$

$$= \frac{1}{q^2} \cdot \sum_{i=1}^{q}\mathbb{E}_{\mathcal{P}}\|\mathbf{A}(\tilde{\mathbf{x}}_i - \mathbb{E}_{\mathcal{P}}[\tilde{\mathbf{x}}_i])\|^2 \tag{31}$$

$$\leq \frac{1+\eta}{q^2} \cdot \sum_{i=1}^{q}\mathbb{E}_{\mathcal{P}}\|\mathbf{A}_i(\tilde{\mathbf{x}}_i - \mathbb{E}_{\mathcal{P}}[\tilde{\mathbf{x}}_i])\|^2 \tag{32}$$

$$\leq \frac{2(1+\eta)}{q^2} \cdot \sum_{i=1}^{q}\mathbb{E}_{\mathcal{P}}\Big(\|\mathbf{A}_i(\tilde{\mathbf{x}}_i - \mathbf{x}_i^*)\|^2 + \|\mathbf{A}_i(\mathbb{E}_{\mathcal{P}}[\tilde{\mathbf{x}}_i] - \mathbf{x}_i^*)\|^2\Big)$$

$$\leq \frac{2(1+\eta)}{q^2} \cdot \sum_{i=1}^{q}\|\mathbf{A}_i\mathbf{x}_i^* - \mathbf{b}_i\|^2 \tag{33}$$

$$\leq \frac{2(1+\eta)}{q^2} \cdot \sum_{i=1}^{q}\|\mathbf{A}_i\mathbf{x}^* - \mathbf{b}_i\|^2. \tag{34}$$

The equality (31) holds as all $\tilde{\mathbf{x}}_i's$ are independent. The inequality (32) holds due to the subspace embedding property (Lemma 17). The inequality (33) holds due to the variance bound in Theorem 1. The last inequality holds because $\mathbf{x}_i^*$ minimizes the least squares loss $\|\mathbf{A}_i\mathbf{x} - \mathbf{b}_i\|^2$ at $i^{th}$ machine. We now upper bound the second term in (30) as

$$\|\mathbf{A}(\mathbb{E}_{\mathcal{P}}[\bar{\mathbf{x}}] - \bar{\mathbf{x}}^*)\| = \Big\|\mathbf{A}\Big(\frac{1}{q}\sum_{i=1}^{q}(\mathbb{E}_{\mathcal{P}}[\tilde{\mathbf{x}}_i] - \mathbf{x}_i^*)\Big)\Big\|$$

$$\leq \frac{1}{q} \cdot \sum_{i=1}^{q}\|\mathbf{A}(\mathbb{E}_{\mathcal{P}}[\tilde{\mathbf{x}}_i] - \mathbf{x}_i^*)\|$$

$$\leq \frac{\sqrt{1+\eta}}{q} \cdot \sum_{i=1}^{q}\|\mathbf{A}_i(\mathbb{E}_{\mathcal{P}}[\tilde{\mathbf{x}}_i] - \mathbf{x}_i^*)\| \tag{35}$$

$$\leq \frac{\sqrt{\epsilon(1+\eta)}}{q} \cdot \sum_{i=1}^{q}\|\mathbf{A}_i\mathbf{x}_i^* - \mathbf{b}_i\| \tag{36}$$

$$\leq \frac{\sqrt{\epsilon(1+\eta)}}{q} \cdot \sum_{i=1}^{q}\|\mathbf{A}_i\mathbf{x}^* - \mathbf{b}_i\|$$

The inequality (35) holds due to the subspace embedding property (Lemma 17) and inequality (36) holds due to the bias bound in Theorem 1. Therefore we get,

$$\|\mathbf{A}(\mathbb{E}_{\mathcal{P}}[\bar{\mathbf{x}}] - \bar{\mathbf{x}}^*)\|^2 \leq \frac{\epsilon(1+\eta)}{q^2}\Big(\sum_{i=1}^{q}\|\mathbf{A}_i\mathbf{x}^* - \mathbf{b}_i\|\Big)^2. \tag{37}$$

Combining (28, 29, 34, 37) and assuming $\eta < 1$ we get,

$$\mathbb{E}_{\mathcal{P}} \|\mathbf{A}\bar{\mathbf{x}} - \mathbf{b}\|^2 \leq \Big(1 + 2c\Big(\frac{\mu \log(d/\delta_1)}{qn} + \frac{\mu^2 \log^2(d/\delta_1)}{n^2}\Big)\Big) \cdot \|\mathbf{A}\mathbf{x}^* - \mathbf{b}\|^2 + \frac{8}{q^2} \cdot \sum_{i=1}^{q} \|\mathbf{A}_i\mathbf{x}^* - \mathbf{b}_i\|^2$$

$$+ \frac{4\epsilon}{q^2}\Big(\sum_{i=1}^{q} \|\mathbf{A}_i\mathbf{x}^* - \mathbf{b}_i\|\Big)^2. \tag{38}$$

As $\mathbb{E}\big(\frac{1}{q}\sum_{i=1}^{q}\|\mathbf{A}_i\mathbf{x}^* - \mathbf{b}_i\|^2\big) = \|\mathbf{A}\mathbf{x}^* - \mathbf{b}\|^2$, we just use Markov's inequality to upperbound the second term in (38) by $\frac{160}{q}\|\mathbf{A}\mathbf{x}^* - \mathbf{b}\|^2$ with probability 0.95. For the last term, we have

$$\Big(\sum_{i=1}^{q} \|\mathbf{A}_i\mathbf{x}^* - \mathbf{b}_i\|\Big)^2 = q \cdot \frac{1}{q}\sum_{i=1}^{q}\|\mathbf{A}_i\mathbf{x}^* - \mathbf{b}_i\|^2 + (q^2 - q) \cdot \frac{1}{q^2 - q}\sum_{i,j,i \neq j}\|\mathbf{A}_i\mathbf{x}^* - \mathbf{b}_i\| \cdot \|\mathbf{A}_j\mathbf{x}^* - \mathbf{b}_j\|$$

Again $\frac{1}{q}\sum_{i=1}^{q}\|\mathbf{A}_i\mathbf{x}^* - \mathbf{b}_i\|^2 \leq 20\|\mathbf{A}\mathbf{x}^* - \mathbf{b}\|^2$ with probability 0.95. The last term is an average of $q^2 - q$ random variables and we can provide uniform bound for the expectation of all of them using the independence among different machines. We have $\mathbb{E}[\|\mathbf{A}_i\mathbf{x}^* - \mathbf{b}_i\| \cdot \|\mathbf{A}_j\mathbf{x}^* - \mathbf{b}_j\|] \leq \sqrt{\mathbb{E}[\|\mathbf{A}_i\mathbf{x}^* - \mathbf{b}_i\|^2]} \cdot \sqrt{\mathbb{E}[\|\mathbf{A}_j\mathbf{x}^* - \mathbf{b}_j\|^2]} = \|\mathbf{A}\mathbf{x}^* - \mathbf{b}\|^2$. Again using Markov's inequality we can upper bound the average of these $q^2 - q$ random variables with 20 times their expectation with probability 0.95, finishing the proof of the theorem. $\qquad\square$

# F   Further experimental details

Here, we provide a small set of numerical experiments to empirically examine the relative error of distributed averaging estimates from individual machines to return an estimator $\hat{\mathbf{x}} = \frac{1}{q}\sum_{i=1}^{q}\tilde{\mathbf{x}}_i$. First, we show that when the sketching-based estimates $\tilde{\mathbf{x}}_i$ do have a non-negligible bias, so that the distributed averaging estimator $\hat{\mathbf{x}}$ remains inconsistent in the number of machines – even with an unlimited number of machines, as long as the space on each machine is limited, the averaged estimator's performance will be limited by the bias of the individual estimates. Second, we show that one can use sketching to compress a data subsample at no extra computational cost, without increasing its bias, which we refer to as the free lunch in distributed averaging via sketching.

We examine three benchmark regression datasets, `Abalone` (4177 rows, 8 features), `Boston` (506 rows, 13 features), and `YearPredictionMSD` (truncated to the first 2500 rows, with 90 features), from the `libsvm` repository from [8]. All the experiments were run on an i9-13900k processor with 128GB of RAM and an NVIDIA RTX 3090 GPU. We repeat the experiment 100 times, with the shaded region representing the standard error. We visualize the relative error of the averaged sketch-and-solve estimator $\frac{L(\hat{\mathbf{x}}) - L(\mathbf{x}^*)}{L(\mathbf{x}^*)}$, against the number of machines $q$ used to generate the estimate $\hat{\mathbf{x}} = \frac{1}{q}\sum_{i=1}^{q}\tilde{\mathbf{x}}_i$.

Each estimate $\tilde{\mathbf{x}}_i$ was constructed with the same sparsification strategy used by LESS, except that instead of sparsifying the sketch with leverage scores, we instead sparsify them with uniform probabilities. Following [16], we call the resulting method LESSUniform. Within each dataset, we perform four simulations, each with different sketch sizes and different numbers of nonzero entries per row. We vary these so that the product (sketch size × nnz per row) stays the same, so as to ensure that the total cost of sketching is fixed in each plot.

As expected, decreasing the sketch size while increasing the number of nonzeros per row (effectively increasing the amount of "compression" occurring here by sparse sketching) increases the error in all three datasets. However, remarkably, increasing the amount of "compression" does not seem to increase the amount of bias. We can therefore conclude that sparse sketches preserve near-unbiasedness, while enabling us to reduce the sketch size from subsampling without incurring any additional computational cost. The increase in error can be mitigated in a distributed setting by increasing the number of estimates/machines.

## F.1   Comparison with other sketching estimators

To further illustrate the phenomenon that suitable sketched least squares estimators enjoy small bias, we provide a further set of numerical experiments below in Figure 4 for the Boston and Abalone

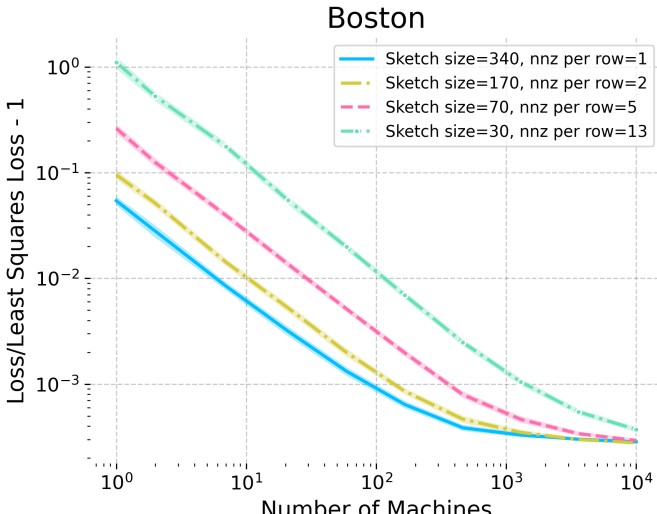

Figure 3: Comparison of the relative error of the distributed averaging estimator of sketch-and-solve least squares estimates where the sketches are constructed with sparse sketching matrices with uniform probabilities (LESSUniform) on `libsvm` dataset `Boston` (see Figure 2 for results on `YearPredictionMSD` and `Abalone`). For each dataset, the computational cost of sketching is the same in all four parameter settings. Remarkably, sketching to a smaller size appears to preserve near-unbiasedness without incurring any additional computational cost.

datasets. Recall that in Figures 2 and 3, we show that increasing the amount of "compression" of the sample, by reducing the sketch size when using sparse sketching matrices with uniform probabilities, does not increase the bias of the sketched estimator. Figure 4 further demonstrates that Gaussian and Subgaussian sketches, which are computationally much more expensive than our proposed sparse sketches, exhibit virtually no bias in sketched least squares (as expected). A similar conclusion can be made for the proper LESS method, which uses leverage score estimates, and is therefore more expensive than the LESSUniform method we use in other experiments, but still potentially much cheaper than Subgaussian sketches.

However, the unbiasedness does not hold for all sparse sketches, as we demonstrate below in Figure 4. At a high level, we show that sketches constructed using leverage score subsampling and the SRHT can exhibit a non-negligible level of least squares bias.

Gaussian and Subgaussian sketches, as we see in the figure, enjoy near-unbiasedness while not introducing additional bias as the sketch size decreases. This does not hold for leverage score subsampling. When we decrease the number of subsamples within leverage score subsampling, the bias introduced by subsampling increases as suggested by the lower bound in Theorem 10 of [18]. Intuitively, this happens as the number of subsamples is reduced without increasing the amount of "compression" as one would with LESS or LESSUniform.

We also show that the subsampled randomized Hadamard transform (SRHT) also introduces increased bias as the sketch size decreases. This complements the lower bound established for the failure of SRHT and other data-oblivious sparse embeddings like CountSketch to satisfy the restricted Bai-Silverstein property that is a structural condition for near-unbiasedness sketches [18].

For completeness, we also demonstrate the desirable performance of LESS proper. Figure 4 demonstrates that the desirable phenomenon that LESSUniform enjoys also extends to LESS proper. In fact, we observe that LESS enjoys similar desirable performance as the Gaussian and Subgaussian sketches and does not increase (in fact, it has minimal) least squares bias, while enjoying the computational speedups that sparse sketches also enjoy, suggesting a best-of-both-worlds property.

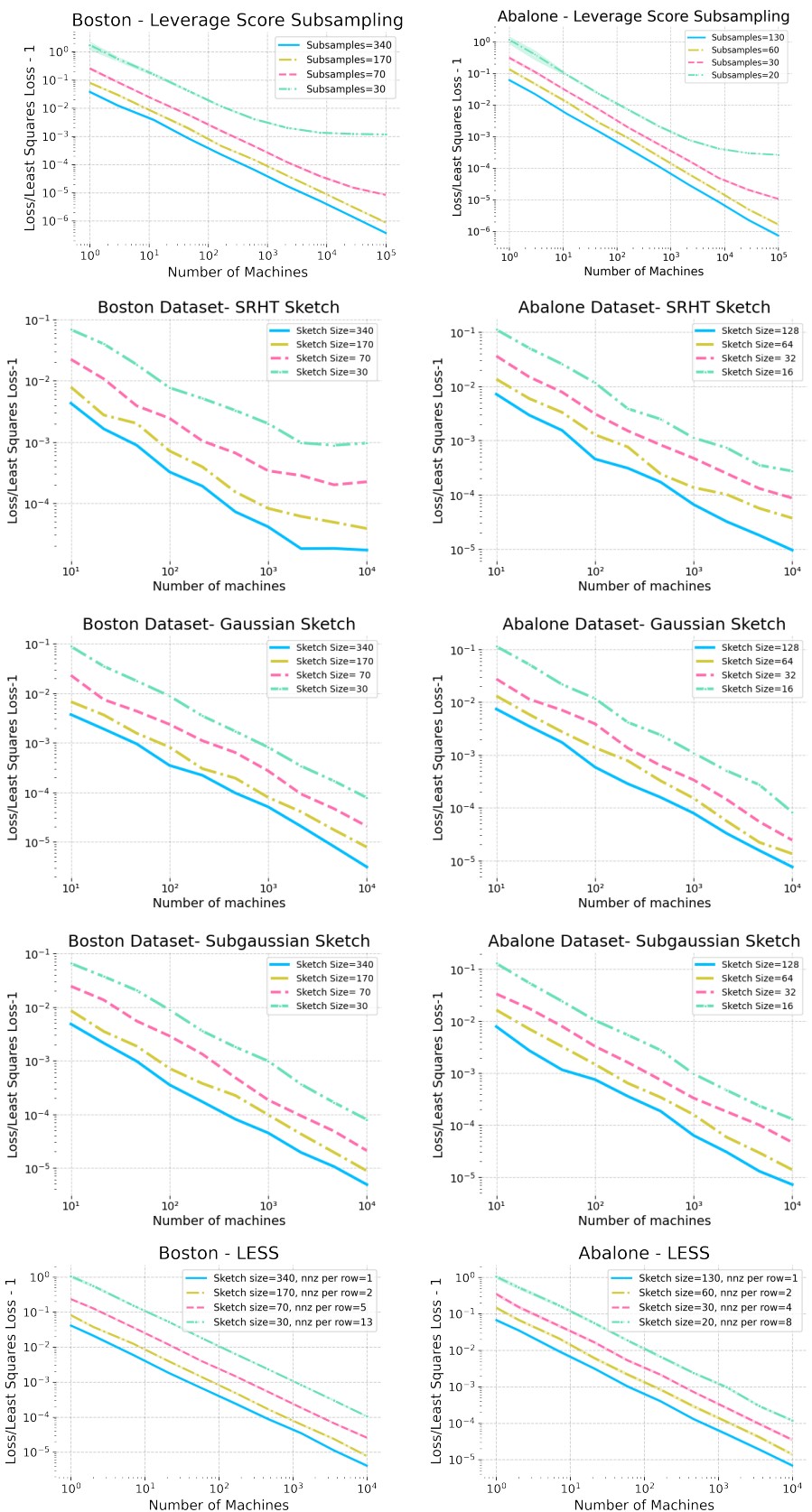

Figure 4: Distributed averaging experiment for the Boston and Abalone datasets using five different sketching techniques: Leverage Score Subsampling, Subsampled Randomized Hadamard Transform (SRHT), Gaussian sketches, Subgaussian sketches, and LESS. Both Gaussian and Subgaussian sketches exhibit no observable least squares bias, whereas Leverage Score Subsampling and SRHT exhibit a small but measurable level of bias. LESS enjoys the similar high performance as the Gaussian and Subgaussian sketches and does not increase (in fact, it has minimal) least squares bias, but as a sparse sketch also allows for the significant improvements in runtime that sparse sketches enjoy, demonstrating that it enjoys the best-of-both-worlds.

