# OpenReview forum: "Distributed Least Squares in Small Space via Sketching and Bias Reduction"
_NeurIPS.cc/2024/Conference — NeurIPS 2024 poster_

### Official Review · Reviewer_h7iG · 2024-07-01

**Soundness:** 3
**Presentation:** 3
**Contribution:** 3
**Rating:** 6
**Confidence:** 3

**Summary:**

Sketching is a technique from randomized numerical linear algebra which compresses an input matrix $A$ by multiplication with a random matrix $S$. Recent works [18] have shown how to characterize the bias in the sketched inverse covariance $(\tilde A^\top\tilde A)^{-1}$, where $\tilde A = SA$ for an input matrix $A$ and a sketching matrix $S$. This work extends this machinery to construct approximately unbiased estimators $\tilde x$ for the least squares regression problem $\min_x \|Ax-b\|$. As the authors show in Theorem 2, this gives an efficient distributed algorithm by averaging the unbiased estimator over $q = O(1/\epsilon)$ servers. The techniques introduced by the authors also improve prior results on bias-free estimation of the inverse covariance.

**Strengths:**

The authors introduce a new way to exploit unbiased estimators, which appears to be different from prior works, which incorporate debiasing in a second order optimization framework (e.g. https://arxiv.org/abs/2007.01327). The new technique simply averages local solutions, and conceptually seems simpler than prior techniques.

Technically, the authors improve prior results on bias-free estimation of the inverse covariance by showing an improved moment bound on a certain random variable by carefully exploiting better bounds for smaller moments via a Holder’s inequality. Other new tools from random matrix theory are additionally needed to handle this change. These ideas, as well as their improvements to bias-free estimation of the inverse covariance, are interesting and may be useful for future results in this area.

**Weaknesses:**

The result and techniques seem to be a bit niche and incremental. The notion of distributed computation is different from ones that I see in other works in the sketching literature (e.g. https://arxiv.org/abs/1408.5823, https://arxiv.org/abs/1504.06729) where the input is partitioned across multiple servers. Instead, this work considers a setting where there is just one stream which contains the entire input, and the central server has access to $q$ machines that can each access this stream (see the Computational model section), which seems nonstandard. The number of servers needed is also rather large (e.g. $q = O(1/\epsilon)$), which also seems restrictive. I encourage the authors to include references that prove results in this setting if there are any others. The main technical novelty (line 275) also seems to be an improvement to a bound in the analysis of LESS embeddings from prior work, while the overall sketching framework is largely unchanged.

**Questions:**

- The authors may consider discussion the related work https://arxiv.org/abs/2203.09755 on distributed least squares and https://arxiv.org/abs/2007.01327 on using debiasing approaches

Other comments

- Lemma 2 should have a probability statement (e.g., with probability at least $1-\delta$)

**Limitations:**

The authors claim that the work does not have limitations in the checklist. Some of the points I raised in the weaknesses section could be discussed in the work, such as restrictions on the computational model and the number of servers.

---

> ### Author Rebuttal · Authors · 2024-08-06
>
> We thank the reviewer for the comments and questions. We are glad that the reviewer appreciates the simplicity of our algorithmic approach, and that the reviewer finds our technical contributions interesting and useful. Below, we provide clarifications on the reviewer's remarks, including on the distributed computation framework and the number of machines, as well as providing a comparison with the four references provided by the reviewer. We will be sure to include all of those details in the final version. If you think the responses adequately address your concerns, we encourage you to consider increasing your score.
> - **Novelty in our techniques.** Our main contribution is in the theoretical analysis, which includes several new ideas (including the higher-moments version of Restricted Bai-Silverstein inequality, as well as the careful use of H\"older's inequality in the analysis of the dominant term). In fact, these ideas are not niche, since they are relevant to many instances of RMT-style analysis (i.e., analysis relying on the Stieltjes transform of the resolvent matrix) for sparse sketching operators, which has been used for Newton Sketch [16], Randomized SVD [17], and Sketch-and-Project [21]. We chose to focus on least squares, as this gives the clearest computational improvements.
> - **Number of machines required.** The use of $O(1/\epsilon)$ machines in our main results comes from the small space complexity, namely $O(d^2\log nd)$ bits per machine, that we impose in our setting. However, if we relax this constraint, then there is a natural trade-off between the space complexity and number of machines required by our methods: If we allow $((1/\theta)d^2\log nd)$ bits per machine for some $\theta\in [\epsilon,1]$, then we only need $O(\theta/\epsilon)$ machines. Setting $\theta=\epsilon$, we recover the standard sketched least squares on a single machine, while $\theta=1$ recovers our main results, but of course we can freely interpolate between those two extremes.
> - **Comparison with [BP22, arXiv:2203.09755].** This work indeed considers essentially the same problem setting as we do. In fact their Algorithm 1 is the same basic distributed averaging procedure that we use, and their computational model described in Remark 2.3 (Option 2) matches our single-server multiple-machines setup (except that they allow random access instead of streaming access to the data). Our results can be viewed as a direct improvement over their guarantees for Gaussian sketches (e.g., their Theorem 2.2), since we obtain analogous guarantees for extremely sparse sketches which are far more computationally efficient. In fact, we already mention these Gaussian sketch guarantees as a baseline for our work (see Table 1 for a comparison), and we will certainly add the reference to this work in that context.
> -  **Comparison with [DBPM20, arXiv:2007.01327].** This work also considers a very similar problem setting to us, except they focus on regularized least squares, and how the regularization parameter affects the bias of the sketched least squares estimator, with distributed averaging as a main motivation (their main results also assume that all workers have access to the centralized data for the purpose of sketching). Similarly as for the earlier reference, their theoretical results require expensive sketching methods (called "surrogate sketches"), which are based on Determinantal Point Processes. We use these sketches as one of our theoretical baselines (see Table 1 for a comparison), and we will certainly add this work as a reference. It is worth noting that our random matrix theory techniques can likely be used to extend the regularization-based debiasing techniques developed in that work to fast sparse sketching. We leave this as a promising direction for future work.
> - **Comparison with [BKLW14, arXiv:1408.5823] and [BWZ16, arXiv:1504.06729].** The distributed computation framework considered in those works partitions the data across multiple servers. Importantly, note that these works solve the task of Principal Component Analysis, which is different from our task of least squares regression, so these works are not directly comparable. Nevertheless, our methods and results can be naturally extended to the multiple-server setting, as we discuss below. The main reason we focused the paper on a more centralized computational model, with one server and multiple machines, is because this allowed us to obtain worst-case results which are independent of condition-number type quantities (such as the data coherence defined below).
> - **Results for data partitioned into multiple servers.** As mentioned above, our results can be extended to the setting where a dataset is partitioned into multiple servers, so that each one of the $q$ machines is accessing a separate chunk of the data. Suppose that a dataset $A,b$ of size $N\times d$ is partitioned uniformly at random into smaller chunks of size $n=N/q$, and each machine constructs an estimate $\tilde x_i$ based on a sketch of its own chunk. Then with the same computational guarantees on each machine as in Theorem 2, the averaged estimator $\tilde x=\frac1q\sum_i\tilde x_i$ will with high probability enjoy a guarantee of $||A\tilde x-b||\leq (1+\epsilon + \tilde O(\mu/n))||Ax^*-b||$, where $\mu=N\max l_i(A)$ is the coherence of the dataset (here, $l_i$ denotes the $i$th leverage score). The additional error term $\tilde O(\mu/n)$, which arises from the partitioning (e.g., see [43]), is often negligible compared to the sketching error $\epsilon$ when the chunk size $n$ is sufficiently larger than the sketch size. An analogous result can be obtained for Distributed Newton Sketch. We will add these claims to the final version.

---

> > ### Comment · Reviewer_h7iG · 2024-08-09
> >
> > Thank you for the rebuttal. I appreciate the additional comments which show how the limitations on the number of servers can be addressed with a smooth trade-off, as well as allowing the data to be partitioned if the chunks are partitioned uniformly (although perhaps it still doesn't work if the data is partitioned adversarially, which is still a limitation in my mind). Thank you also for the explanation of connections with the prior work. I have increased my score, in recognition that there is a growing body of work on RMT-style arguments in the sketching literature, and thus this work may have a broad impact in many future works.

---

### Official Review · Reviewer_X2Ks · 2024-07-02

**Soundness:** 3
**Presentation:** 3
**Contribution:** 4
**Rating:** 7
**Confidence:** 3

**Summary:**

Sketched least squares involve estimating the term $(X^TX)^{-1}$ which has a high bias when the sketch matrix $S$ is not sub-Gaussian. This paper gives a sparse sketching method using a LESS embedding which runs in optimal space and current matrix multiplication time, where $S$ is sparse, and constructed based on the leverage scores of the data matrix $A$ (Definition 2 $(s,\beta_1,\beta_2)$-LESS embedding). The paper also improves the sharpness of the probability bounds which is applicable to similar problems in RMT using LESS embeddings. For $s=1$, nothing is different, but when $s > 1$, the bias bound is reduced.

**Strengths:**

Inversion bias for estimates of $(X^TX)^{-1}$ is a challenging problem in least-squares sketching. Sub-gaussian sketches have high computational cost but low bias ; other sketches have low computational cost but high bias. The sketch proposed in this paper minimizes the bias of the estimator only requires 1 parallel pass over the data, and has runtime of nnz($A$) + $\tilde{O}(d^2/\epsilon)$ which is much faster. Moreover, computing the estimate can be done in parallel, with only the final result needing to be averaged.

The authors rigorously prove that their sketch fulfills the above criteria by using techniques from random matrix theory, and this is not trivial at all. I find the key contribution in the paper is proving these statements, and the authors did an excellent job here. The paper is extremely well written and easy to follow. Of particular note is the bias analysis for the least squares estimator (Section 4) which is concisely written, where the main ideas (proof sketch) given in the main paper, and technical details in the appendix. The ideas given in the proof sketch can be used for similar problems, and are straightforward to understand. Moreover, the technical details come with sufficient exposition such that it is straightforward for a reader to understand the direction the proof is going (which is certainly much appreciated).

**Weaknesses:**

1. It would be nice to have experiments with other sketching methods to (empirically) justify some statements on the bias, variance (although not stated) and computational time, e.g. estimators mentioned in Table 1. For example, are there sketching estimators with higher bias, but less variance?

There are some minor typos, e.g. line 212 reference missing, lines 282 to Equation (2) at bottom of the page is missing a bracket for the numerous expectations (in contrast to lines 542 onwards in the appendix).

In Appendix A, notation for concentration inequalities should be looked at and made consistent, e.g. Lemma 6 / H{\"o}lder's inequality should have a $\frac{1}{q}$, Lemma 7 ($\lambda$ max isn't defined), Lemma 8 / Azuma's inequality ($\lambda$, $m$ should be consistent), Lemma 10 is missing a bracket for $\mathbb E[x_i^2]$.

The presentation of the proof for Theorem 5 was slightly jarring (due to Lemma 11, Lemma 12 appearing in the proof), but there also doesn't seem to be a good way to include them (since referring to the two lemmas requires the upper bounds, and flipping back a page is also inconvenient). Maybe a solution is to indent the Lemmas, or box them up?

The dot before line 626 (after 72) should be removed.

**Questions:**

1. It would be nice to have experiments with other sketching methods to (empirically) justify some statements on the bias, variance (although not stated) and computational time, e.g. estimators mentioned in Table 1. For example, are there sketching estimators with higher bias, but less variance?

2. Despite being clear to read, I had to go back and forth a bit to find out what the novelty is. I appreciate the clarity and thoroughness of explaining the bounds on the bias and variance, runtime, but I would like it if Definition 1 & 2 came much earlier (or at least maybe an informal Definition 2 after line 85?)

---

> ### Author Rebuttal · Authors · 2024-08-06
>
> We thank the reviewer for a careful read and detailed comments. We will address all of the typos and presentation suggestions in the final version.
> - **Additional experiments.** We include additional experiments with other sketching methods (all of which are more computationally expensive than the fast sparse LESSUniform method we used in the paper): Leverage Score Sampling, Gaussian and Subgaussian sketches, as well as Subsampled Randomized Hadamard Transform (see our general response and the PDF). We note that the fact that sketching methods tend to yield smaller least squares bias than uniform subsampling has been empirically observed in prior works [42], which is why we did not focus on this here. Our main contribution is to provide the first sharp theoretical characterization of this phenomenon for extremely sparse sketches.
> - **Are there sketching estimators with higher bias, but less variance?** By definition, the bias has to be no larger than the variance, but there can be cases where the two quantities are comparable in size (which implies that distributed averaging will not be effective). The main example here is i.i.d. sub-sampling, including leverage score sampling. There are theoretical lower bounds [18] which show that (in some cases) the bias of leverage score sampling may not be much smaller than the variance.

---

> > ### Comment · Reviewer_X2Ks · 2024-08-07
> >
> > Thank you for the rebuttal, and for providing the experiments. I was thinking more of biased estimators with lower variance though. My score remains unchanged.

---

### Official Review · Reviewer_RJVb · 2024-07-12

**Soundness:** 3
**Presentation:** 3
**Contribution:** 3
**Rating:** 7
**Confidence:** 3

**Summary:**

The paper studies the least squares regression task and improves the space and communication amount in a distributed setting to be independent of $\epsilon$. The key to achieve this is by sketching the data in blocks that have an $\epsilon$ dependence which can be reduced to only d-dependencies by aggregating them. For the covariance A^TA this becomes only a d x d matrix and for the other required term A^Tb it can be handled by communicating and aggregating just the solutions which have a dimension of d. Although the idea seems very simple described this way, the analysis seems highly non-trivial and has very interesting aspects and novel techniques (or at least new to me). It analyzes in this setting not only the standard least squares error but also a bias term, which allows for smaller sketch dimensions, and only 1/sqrt(eps) dependencies.

**Strengths:**

* gives least squares sketching results with lower time, space, and communication complexities
* interesting techniques
* some further applications are given, though details are fully in the appendix
* very good writing

**Weaknesses:**

* motivation of the model, seems to be a niche
* very low improvements but there seems no big gap that can be leveraged

**Questions:**

* I am slightly confused by the definition of the bias. Is $||AE(\tilde x)-b||$ standard in some literature? I would say this is the variance of the expected sketched estimator. Why would I be interested in this expected estimator, instead of the actual outcome after sketching?
* I would rather define the bias as $||\tilde x -x^*||$ which I think does not allow for any improvements over standard sketching results, right?

**Limitations:**

f

---

> ### Author Rebuttal · Authors · 2024-08-06
>
> We thank the reviewer for the positive feedback, as well as the comments and questions. We will address them in the final version.
> - **Definition of bias.** At a high level, we rely on the statistical definition of the bias of an estimator, which is (informally) the difference between the expectation of that estimator and its target quantity (the estimand). In the case of a least squares estimator $\tilde x$, the most standard statistical notion of bias would be $E[\tilde x]-x^*$. Since the estimator is multivariate, one will typically measure the amount of bias by taking the norm, i.e., $||E[\tilde x]-x^*||$ (as opposed to $||\tilde x-x^*||$ which is typically referred to as the estimation error). In our case, since we are interested in the least squares prediction vector, i.e. $A\tilde x$, as an estimator of the vector $b$, we compute the bias as $||E[A\tilde x]-b||=||A E[\tilde x]-b||$. This turns out to be the exact right notion for the purpose of distributed averaging and bounding the least squares loss.
> - **Motivation of the model.** Our model is motivated by the general distributed averaging framework (also known as model averaging, or bagging), which is widely used in many settings beyond least squares. In fact, our methods and results are applicable much more generally than the single-server multiple-machine computation model used in the paper. For example, they can be naturally extended to the multiple-server model [43], where the data is randomly partitioned into multiple chunks stored on separate servers, which is common in the literature (see response to Reviewer h7iG for details). The main reason we focused the paper on the single-server multiple-machine model is because this allowed us to obtain worst-case results that are independent of condition-number type quantities (those are unavoidable in the multiple-server model, given our other computational constraints).

---

> > ### Comment · Reviewer_RJVb · 2024-08-12
> >
> > Thank you for the clarifications. I will keep my score.

---

### Official Review · Reviewer_8xrQ · 2024-07-12

**Soundness:** 3
**Presentation:** 3
**Contribution:** 3
**Rating:** 5
**Confidence:** 3

**Summary:**

This paper presents new techniques for distributed least squares regression using matrix sketching. The key contributions are:

1. A sparse sketching method that produces a nearly-unbiased least squares estimator in two passes over the data, using optimal space and current matrix multiplication time.
2. Improved communication-efficient distributed averaging algorithms for least squares and related tasks.
3. A novel bias analysis for sketched least squares, characterizing its dependence on sketch sparsity. This includes new higher-moment restricted Bai-Silverstein inequalities.

The theoretical results are backed by experiments on real datasets showing the practical benefits of the approach.

**Strengths:**

1. Provides a sparse sketching method that achieves near-unbiased least squares estimation in optimal space and current matrix multiplication time.
2. Achieves O(d^2 log(nd)) bits of space, which is optimal. Matches current matrix multiplication time O(d^ω), improving over previous approaches.
3. Introduces a new bias analysis for sketched least squares that sharply characterizes dependence on sketch sparsity. The techniques developed may be applicable to other sketching and randomized linear algebra problems.

**Weaknesses:**

1. Experiments are conducted on only a few datasets. And it does not explore a wide range of problem sizes or distributed computing scenarios.
2. Primarily focused on least squares regression, with limited discussion of extensions to other problems.
3. Some of the theoretical results rely on assumptions (e.g., about leverage score approximation) that may not always hold in practice.

**Questions:**

1. Are there natural extensions of this work to other loss functions beyond least squares?
2. Do you expect the techniques developed here to be applicable to other sketching problems beyond least squares? If so, which ones?

Minor Comments:

1. The abstract could more clearly state the key theoretical results/bounds achieved
2. Some additional discussion of practical implications and potential applications would be valuable
3. A few typos noted (e.g. line 330)

**Limitations:**

yes

---

> ### Author Rebuttal · Authors · 2024-08-06
>
> We thank the reviewer for comments and feedback. We will revise the abstract, and also expand our discussion of applications beyond least squares as outlined below.
> - **Extensions to other loss functions beyond least squares.** In addition to least squares, we provide a more broadly applicable result in Theorem 3, which in particular applies to minimizing general convex loss functions via the framework of Distributed Newton Sketch (see Corollary 1). Thus, our sketching methods and proof techniques are of broader interest to sketching-based optimization algorithms for a variety of loss functions, including the logistic loss, other generalized linear model losses such as the hinge loss, etc.
> - **Applications to other sketching problems.** Our theoretical analysis, which includes several new ideas (including the higher-moment version of Restricted Bai-Silverstein inequality, as well as the careful use of H"older's inequality in the analysis) is relevant to many instances of RMT-style analysis (i.e., analysis relying on the Stieltjes transform of the resolvent matrix) for sparse sketching operators. This RMT-style analysis has been used in Newton Sketch [16], Randomized SVD [17], and Sketch-and-Project [21]. We chose to focus on distributed least squares, as this gives the clearest worst-case computational improvements.
> - **Distributed computing scenarios.**  In fact, our methods and results are applicable much more generally than the single-server multiple-machine computation model used in the paper. For example, they can be naturally extended to the multiple-server model [43], where the data is randomly partitioned into multiple chunks stored on separate servers, which is common in the literature (see response to Reviewer h7iG for details). The main reason we focused the paper on the single-server multiple-machine model is because this allowed us to obtain worst-case results that are independent of condition-number type quantities (those are unavoidable in the multiple-server model, given our other computational constraints).
> - **Experiments.** Our main contribution is to provide the first sharp theoretical characterization of the least squares bias for extremely sparse sketches, and this is also where we focused our experiments. Nevertheless, we include additional experiments (see the general response and PDF) with other sketching methods (all of which are more computationally expensive than the fast sparse LESSUniform method we used in the paper): Leverage Score Sampling, Gaussian and Subgaussian sketches, as well as Subsampled Randomized Hadamard Transform.
> - **Assumptions about leverage score approximation.** Our main results, Theorems 2 and 3, do not require any assumptions related to leverage score approximation, as they include leverage score approximation as part of the algorithmic procedure.

---

> > ### Comment · Reviewer_8xrQ · 2024-08-14
> >
> > Thank you for your response on the questions! After reading the other reviews, I would like to keep my current score of 5.

---

### Author Rebuttal · Authors · 2024-08-06

Thanks to all reviewers for the positive feedback and comments. We responded to those comments in the individual responses to each reviewer. We also provided additional experimental results on four different sketching methods (included in the PDF), and discussed the implications of our theoretical results beyond least squares and in other distributed models. All of this will be included in the final version of the paper, alongside other reviewer suggestions. Here, we summarize the main takeaways.
- **Theoretical implications beyond least squares.** Our main contribution is a set of new theoretical techniques (e.g., higher-moment Restricted Bai-Silverstein, Lemma 3) for analyzing sparse sketching methods, which goes far beyond least squares, as we showed in Theorem 3 and Corollary 1, with an application to general optimization over convex losses (e.g., logistic loss, GLMs, etc) via a variant of the Distributed Newton Sketch. These techniques have wide implications for the analysis of other sketching problems where Restricted Bai-Silverstein-type inequalities have been used, including Randomized SVD [17] and Sketch-and-Project [21].
- **Extensions to other computation models.**  In fact, our methods and results are applicable much more generally than the single-server multiple-machine computation model used in the paper. For example, they can be naturally extended to the multiple-server model [43], where the data is randomly partitioned into multiple chunks stored on separate servers, which is common in the literature (see response to Reviewer h7iG for details). The main reason we focused the paper on the single-server multiple-machine model is because this allowed us to obtain worst-case results that are independent of condition-number type quantities (those are unavoidable in the multiple-server model, given our other computational constraints).
- **Additional experiments.** The fact that sketching methods tend to yield smaller least squares bias than uniform subsampling has been empirically observed in prior works [42]. Our main contribution is to provide the first sharp theoretical characterization of this phenomenon for extremely sparse sketches, and this is also where we focused our experiments. Nevertheless, we include additional experiments (see the PDF) with other sketching methods (all of which are more computationally expensive than the fast sparse LESSUniform method we used in the paper): Leverage Score Sampling, Gaussian and Subgaussian sketches, as well as Subsampled Randomized Hadamard Transform (SRHT). These numerical results further support the claim that sketching enjoys small least squares bias.

---

### Decision · Program_Chairs · 2024-09-25

**Decision:**

Accept (poster)

**Comment:**

This paper proves an interesting headline theoretical result on faster small space distribution regression, requires non-trivial analysis, and also provides nice initial experiments on the method. There are some limitations — for example, that the distributed regression setup the authors consider is fairly niche. Nevertheless, we recommend accepting this paper to NeurIPS.